# ORCHILEAK (revision 3875): A new model branch to simulate carbon transfers along the terrestrial-aquatic continuum of the Amazon basin

Ronny Lauerwald[1,2,3], Pierre Regnier[1], Marta Camino-Serrano[4], Bertrand Guenet[2], Matthieu Guimberteau[2], Agnès Ducharne[5], Jan Polcher[6], Philippe Ciais[2]

[1]Université Libre de Bruxelles, Belgium
[2]IPSL-LSCE, Gif-sur-Yvette, France
[3]University of Exeter, United Kingdom
[4]CREAF, Barcelona, Catalonia, Spain
[5]UPMC, UMR Metis, Paris, France
[6]IPSL-LMD, Paris, France

*Correspondence to*: Ronny Lauerwald (R.Lauerwald@exeter.ac.uk)

**Abstract.** Lateral transfer of carbon (C) from terrestrial ecosystems into the inland water network is an important component of the global C cycle, which sustains a large aquatic $CO_2$ evasion flux fuelled by the decomposition of allochthonous C inputs. Globally, estimates of the total C exports through the terrestrial-aquatic interface range from 1.5 to 2.7 Pg C $yr^{-1}$ (Cole et al. 2007; Battin et al. 2009; Tranvik et al. 2009), i.e. in the order of 2-5% of the terrestrial NPP. Earth System Models (ESM) of the climate system ignore these lateral transfers of C, and thus likely overestimate the terrestrial C sink.

In this study, we present the implementation of fluvial transport of dissolved organic carbon (DOC) and $CO_2$ into ORCHIDEE, the land surface scheme of the Institut Pierre-Simon Laplace ESM. This new model branch, called ORCHILEAK, represents DOC production from canopy and soils, DOC and $CO_2$ leaching from soils to streams, DOC decomposition and $CO_2$ evasion to the atmosphere during its lateral transport in rivers, as well as exchange with the soil carbon and litter stocks on floodplains and in swamps. We parameterized and validated ORCHILEAK for the Amazon basin, the world's largest river system with regard to discharge and one of the most productive ecosystems of the world.

With ORCHILEAK, we are able to reproduce observed terrestrial and aquatic fluxes of DOC and $CO_2$ in the Amazon basin, both in terms of mean values and seasonality. In addition, we are able to resolve the spatio-temporal variability in C fluxes along the canopy-soil-aquatic continuum at high resolution (1°, daily) and to quantify the different terrestrial contributions to the aquatic C fluxes. We simulate that more than 2/3 of the Amazon's fluvial DOC export is contributed by the decomposition of submerged litter. Throughfall DOC fluxes from canopy to ground are about as high as the total DOC inputs to inland waters. The latter, however, are mainly sustained by litter decomposition. Decomposition of DOC and submerged plant litter contributes slightly more than half of the $CO_2$ evasion from the water surface, while the remainder is contributed by soil respiration. Total $CO_2$ evasion from the water surface equals about 5% of the terrestrial NPP. Our results highlight that ORCHILEAK is well suited to simulate carbon transfers along the terrestrial-aquatic continuum of tropical forests. It also

opens the perspective that provided parameterization, calibration and validation is performed for other biomes, the new model branch could improve the quantification of the global terrestrial C sink and help better constrain carbon cycle-climate feedbacks in future projections.

## 1 Introduction

The 5th Assessment Report (AR5) of the Intergovernmental Panel on Climate Change (IPCC) acknowledges the transport of carbon (C) across the inland water network as a key component of the global C cycle (Ciais et al., 2013), involving a significant lateral C transfer along the flow path and stimulating vertical C fluxes in the form of greenhouse gases. However, Earth System models (ESMs) of the climate system and biogeochemical cycles used for the IPCC 5th Assessment currently omit lateral C transfers and simulate only local vertical exchange of C between atmosphere, vegetation and soils from photosynthesis, respiration and fires (Regnier et al., 2013). This is a major knowledge gap because recent evidence, from multiple disciplines, has highlighted that anthropogenic disturbances likely increase the lateral C transfers along hillslopes of upland catchments and through streams and rivers (Battin et al., 2009; Cole et al., 2007; Regnier et al., 2013). This perturbation may reduce significantly the estimated carbon stored in terrestrial vegetation and soils (Regnier et al., 2013) and increase the C evasion from inland waters to the atmosphere. Thus, it is suggested that lateral carbon transfers induce a positive feedback on the coupled carbon cycle-climate system, enhancing atmospheric $CO_2$ levels and global temperature.

Despite this important paradigm shift in carbon cycle science, it must be recognized that the quantitative significance of inland waters for the global C budget is entailed with large uncertainties. In particular, the horizontal flux of organic C through the terrestrial-aquatic interface is poorly constrained (Regnier et al., 2013). Global first-order estimates of this flux, calculated as the sum of estimates of fluvial total organic C (TOC) exports to the coastal ocean, particulate organic C (POC) burial in aquatic sediments and net-$CO_2$ evasion through the air-inland water interface of the Land-Ocean Aquatic Continuum (LOAC, Fig. 1), range from 1.5 to 2.7 Pg C yr$^{-1}$ (Battin et al., 2009; Cole et al., 2007; Tranvik et al., 2009), i.e. in the order of 2-5% of the terrestrial NPP. It is now broadly accepted that the $CO_2$ outgassing from inland waters is the major export path in the LOAC C budget (Battin et al., 2009; Ciais et al., 2013; Le Quéré et al., 2014; Regnier et al., 2013; Tranvik et al., 2009), highlighting the highly reactive character of continental aquatic systems. However, it remains challenging to attribute and quantify the sources of the $CO_2$ evasion, as it is generally not known how much of the evading $CO_2$ originates from terrestrial soil respiration, from in-stream respiration of terrestrially derived organic C, or from other sources such as root respiration of wetland plants (Abril et al., 2014). This is not only true at the global scale, but also at the regional scale of large river catchments like the Amazon basin. Budget calculations from observations alone have limited capabilities to constrain such C exports from terrestrial ecosystems, in particular with regard to temporal and spatial variability.

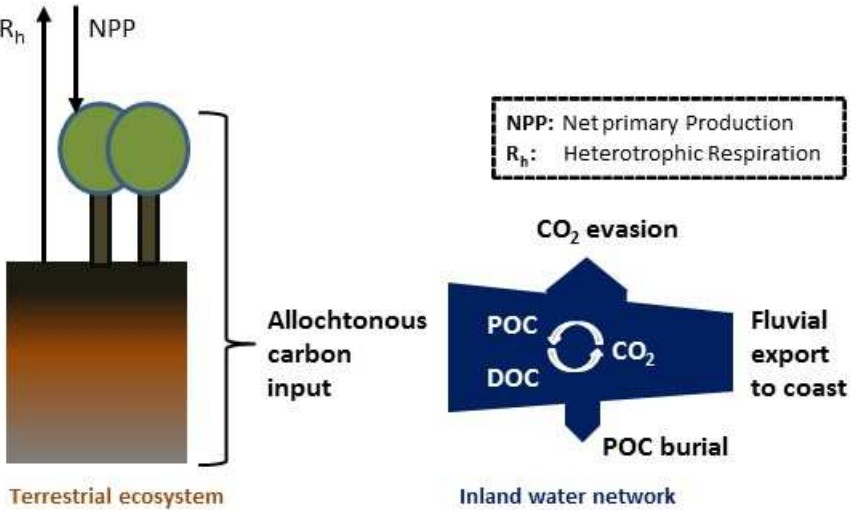

**Figure 1:** Representation of C exports from terrestrial ecosystems through the Land-Ocean Aquatic continuum (LOAC).

In this study, we present an integrated, physical-based modelling approach, which incorporates the various allochthonous
sources of dissolved organic carbon (DOC) and $CO_2$ to the inland water network, the lateral transfers of C along the inland
water network, as well as transformation of C in transit and $CO_2$ exchange with the atmosphere in a temporally resolved and
spatially explicit manner. We parameterize and develop the model for the Amazon basin, although it is intended to be
generalized in future works to be applied at global scale. We consider the Amazon basin as an appropriate while challenging
benchmark test, as it is the world's largest river system with regard to discharge (206 000 $m^3$/s, Callede et al. 2010)  and one
of the most productive ecosystem of the world (Grace et al., 2001). Richey et al. (2002) estimated the $CO_2$ evasion from the
Amazon River system and its connected floodplains at 0.47 Pg C $yr^{-1}$, about 13 times the fluvial TOC exports to the Atlantic
Ocean from this catchment. Such evasion flux corresponds to about 6% of the average terrestrial NPP within the Amazon
basin. In the Amazon River and its major tributaries, in-stream respiration of allochthonous OC is likely the dominant source
of $CO_2$. The study by Mayorga et al. (2005) further revealed that a small pool of labile organic carbon maintains high $CO_2$
levels in the water column, likely linked to inputs from the riparian zone, while the bulk of TOC transported in the river channel
is older and more refractory. Richey et al. (2002) also showed that the intense seasonal flooding in the central Amazon basin
is a major control of river $CO_2$ dynamics, suggesting submerged leaf-litter in flooded forests and root respiration of floating
and emergent plants to be important sources of $CO_2$. In a more recent study, Abril et al. (2014) estimated that riparian wetlands
in the Amazon river system export about half of their gross primary production (GPP) to rivers as TOC and dissolved $CO_2$
produced by autotrophic root respiration in wetland plants, while terrestrial ecosystems export only a few percent of their GPP.
Vascular wetland plants, including flooded forests and floating grasses clearly dominate primary production in the flooded
areas, the autochthonous contribution from phytoplankton and periphyton being negligible (Melack et al., 2009). Another

specific challenge is the reproduction of the different DOC loadings from the different sub-basins of the Amazon. While most of the major tributaries are white or clear water rivers with low to moderate average DOC concentrations of up to 6 mg C L$^{-1}$, the Rio Negro, which after the Rio Madeira is the second largest tributary of the Amazon, is a black water river with twice the concentrations of DOC (Moreira-Turcq et al., 2003).

Recently, a first step in modelling the Amazon river C dynamics was performed using a river carbon model (RivCM) coupled to the land surface scheme LPJmL (Lund-Potsdam-Jena managed Land, Bondeau et al., 2007) to simulate fluvial C transfers in the Amazon basin (Langerwisch et al., 2016). While the model was able to roughly reproduce the annual DOC export to the coast, it still largely underestimated the $CO_2$ evasion from the inland water network to the atmosphere, indicating that C inputs into the river network and their subsequent transformation would need to be reassessed. In our study, we go a step further with

the direct implementation of the non-conservative transport of C through the inland water network into the ORCHIDEE land surface model (ORganising Carbon and Hydrology In Dynamic EcosystEms, Krinner et al., 2005). This approach has the advantage to account for the effects of the lateral exports on the carbon budgets of terrestrial ecosystems, and could thus help refining the assessment of the terrestrial C sink and its feedback on the climate system. The newly developed model branch, called ORCHILEAK, represents DOC production from soils and canopy, DOC and $CO_2$ leaching from soils to river

headstreams, DOC decomposition and $CO_2$ evasion to the atmosphere during its lateral transport in rivers, as well as exchange with the soil carbon and litter stocks in riparian wetlands. The production and leaching of DOC relies on a new soil carbon module ORCHIDEE-SOM (Camino Serrano, 2015) with a vertically resolved soil column. We simulate all C fluxes and stocks at half-hourly to daily time steps, which allows representing seasonal and inter-annual variations. We focus on the lateral transfer of dissolved $CO_2$ and dissolved organic C (DOC), which represents the major and more reactive proportion of TOC

exports to the coasts in the Amazon basin (Moreira-Turcq et al., 2003). Although we neglect the lateral transport of POC, we simulate decomposition of submerged litter in floodplains and rivers as an important source of DOC and $CO_2$ to the water column. While being of importance for the GHG exchange, $CH_4$ evasion is assumed to be negligible with regard to C exports (Wilson et al., 2016). Further, we ignore the fluxes of carbonate alkalinity as at average pH values of 6.5 to 7.2 typical of the Amazon basin (Richey et al., 1990)  the concentrations of $CO_3^{2-}$ are negligible  and, thus, the carbonate-buffering of $CO_2$ is

limited.2 Model developments

ORCHILEAK is based on the recent model branch ORCHIDEE-SOM (Camino Serrano, 2015) which relies on a novel module representing the vertical distribution of soil organic carbon (SOC) and associated transport and reaction processes. These processes include the production, consumption, adsorption/desorption and transport of DOC within the soil column as well as DOC exports from the soil column by drainage and surface runoff. In this study, the module is upgraded to represent DOC

cycling in tropical rain forests, in particular by adding fluxes of DOC from the atmosphere and canopy with throughfall and by distinguishing soil carbon processes on non-flooded and flooded soils, including the direct input of DOC and $CO_2$ from the decomposition of submerged litter and soil carbon to the water column. The trunk-version of ORCHIDEE, as well as the branch ORCHIDEE-SOM, includes a river routing module (Guimberteau et al., 2012; Polcher, 2003) that simulates the lateral

transfer of water from one grid to another, representing the river channel as well as connected wetlands. Here, this routing module has been upgraded with a tracer transport equation to simulate the fluxes of DOC and $CO_2$ along the fluvial network, distinguishing two pools of DOC, labile and refractory DOC. In addition, the representation of the floodplain dynamics is improved in this study to better reproduce the seasonal flooding in the Amazon basin, which is a major controlling factor of

the water (Guimberteau et al., 2012) and carbon flow dynamics along the river network (Richey et al., 1990). ORCHIDEE can be run at different spatial and temporal resolutions. Here, in line with Guimberteau et al. (2012), the model runs for calibration and model testing were performed at 1° spatial resolution over the period 1980-2000, using the regional climate and wetland forcing for the Amazon from Guimberteau et al. (2012), forcing of land cover and land use change after Belward et al. (1999), Olson et al. (1983) and Hurtt et al. (2006), river flow directions from Vörösmarty et al. (2000), as well as soil parameters after

Reynolds et al. (1999) and the Harmonized World Soil Data base (FAO/IIASA/ISRIC/ISS-CAS/JRC, 2009) compiled by Guenet et al., in prep. The necessary forcing data are listed in table 1. As temporal resolution, we use the default 30-minute time-step for all vertical exchanges of water, carbon and energy between atmosphere, vegetation, and soils, and the default 1-day time step for the lateral routing of water. In the following, the model description will be based on these spatial and temporal resolutions. To obtain initial soil carbon pools which are in steady-state with the model set-up for the 1980-2000 period, the

model was first run for 5000 years, looping over the full set of climate forgings and using the land use and an atmospheric $pCO_2$ as representative for the year 1980. The terrestrial C pools simulated during this initialization phase were subsequently used for the simulation over the period 1980-2000 with changing land cover and increasing atmospheric $pCO_2$.This section starts with the representation of the soil hydrology and the river routing scheme in ORCHIDEE and ORCHILEAK (section 2.1). Here, we give an overview of the features that are shared between the original version of ORCHIDEE (the configuration

used by Guimberteau et al., 2012) and ORCHILEAK and we then highlight the improvements that have been implemented in ORCHILEAK. In the second part, the mathematical formulation of DOC production and leaching from the soil as well as transport and transformation of DOC and $CO_2$ along the fluvial network is described (section 2.2).

## 2.1 Hydrology

Like most land surface schemes of ESMs, ORCHIDEE distinguishes two kinds of surface hydrology processes: (i) the water

budget processes, which are mostly vertical and control the partitioning of precipitation into evapotranspiration, infiltration, production of surface runoff and drainage (section 2.1.1); (ii) the horizontal transfer, or routing, of grid-based simulated surface runoff and drainage along the river network (section 2.1.2, with improvements described in 2.1.3).

### 2.1.1 Water budget and soil hydrology

In the vegetation canopy, rainfall is partitioned between interception loss and throughfall according to the leaf area index

(LAI). The throughfall (possibly increased by snowmelt in cold climates and by return flow from the floodplains, cf. section 2.1.2) is then further subdivided into infiltration into the soil and surface runoff produced by infiltration excess. In ORCHIDEE,

the infiltration rate depends on precipitation rates, local slope, and vegetation and is limited by the hydraulic conductivity of the soil, which defines a Hortonian surface runoff (D'Orgeval et al. 2008). The corresponding parameterization is tightly linked to the soil moisture redistribution scheme, which is ruled by the Richards equation, solved here over a 2 m soil profile, using an 11-layer discretization, with layers of geometrically increasing depth (de Rosnay et al. 2002; Campoy et al. 2013). The redistribution of soil moisture is controlled by the soil hydraulic properties, transpiration and evaporation within the soil column, and a gravitational drainage at the soil bottom. All these processes are simulated at a 30 min time step and a 1° resolution. In addition, a bottom return flow feeding the soil is also accounted for in presence of swamps, simulated at the daily time-step of the routing scheme (section 2.1.2).

**Table 1**. List of forcing data needed to run ORCHILEAK. See text for explanations and Fig. 6 for an overview.

| Variable | Spatial resolution | Temporal resolution | Data source |
|---|---|---|---|
| | | *Forcing data* | |
| Rainfall | 1° | 6 hours | Guimberteau et al. (2012), replaced original NCC data |
| Snowfall | 1° | 6 hours | NCC (Ngo-Duc et al., 2005) |
| Air Temperature (close to surface) | 1° | 6 hours | NCC (Ngo-Duc et al., 2005) |
| Incoming shortwave radiation | 1° | 6 hours | NCC (Ngo-Duc et al., 2005) |
| Incoming longwave radiation | 1° | 6 hours | NCC (Ngo-Duc et al., 2005) |
| Air pressure (close to surface) | 1° | 6 hours | NCC (Ngo-Duc et al., 2005) |
| Wind speed (10 m above surface) | 1° | 6 hours | NCC (Ngo-Duc et al., 2005) |
| Relative humidity (close to surface) | 1° | 6 hours | NCC (Ngo-Duc et al., 2005) |
| Soil texture class | 0.5° | - | Reynolds et al. (1999) |
| Soil pH | 0.5° | - | after HWSD v 1.1 (FAO et al., 2009) |
| Soil bulk density | 0.5° | - | after HWSD v 1.1 (FAO et al., 2009) |
| Poor soils | 0.5° | - | This study after HWSD v 1.1 (FAO et al., 2009) |
| Land cover (and change) | 0.5° | annual | after Belward et al. (1999), Olson et al. (1983) and Hurtt et al. (2006) |
| Stream flow directions | 0.5° | - | STN-30p (Vorosmarty et al., 2000) |
| Topographic index ($Topo_{grid\,x}$) | 0.5° | - | STN-30p (Vorosmarty et al., 2000) |
| Floodplains ($\%flood_{max}$) | 0.5° | - | After Guimberteau et al. (2012) |
| Swamps ($\%swamp$) | 0.5° | - | After Guimberteau et al. (2012) |
| River surface areas ($A_{river}$) | 0.5° | - | Lauerwald et al. (2015) |
| 10th, 50th, 90th percentile of the stream reservoir | 1° | - | derived from pre-runs with ORCHIDEE (see text) |
| 95th percentile of water table height over flood plain | 1° | - | derived from pre-runs with ORCHIDEE (see text) |

## 2.1.2 Routing of water along the river network, floodplains and swamps

The river routing module simulates the water exports from the soil column as river discharge along a distributed routing scheme, and it is possible to simulate lateral flows at a higher spatial resolution than the rest of the model to better describe the borders of watersheds within each grid-box and the directions of incoming / outgoing water from distinct basins (Fig.2). For that, each ORCHIDEE grid cell x is divided into multiple subunits named "basins". As in our case, we run simulations at 1° resolution and use a routing scheme at 0.5° resolution (Vörösmarty et al., 2000), each grid cell is simply subdivided into four basins (Fig. 2). Note that all information derived from the forcing files or computed in the other modules has the resolution of the grid cell and is then downscaled to the basins within the routing module. In the following, variables at grid scale are

denoted by the index 'grid x', while information at basin scale are denoted by the index '$i$'. For a full overview of the variables and the system of indices used here, consult Table A.1 in the appendix.

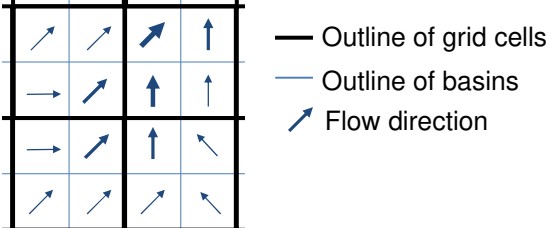

— Outline of grid cells
— Outline of basins
↗ Flow direction

**Figure 2.** Schematic representation of 4 ORCHIDEE grids $x$ at 1 degree spatial resolution for a simulation using a river routing scheme running at 0.5-degree resolution.

The river routing aggregates the 30' surface runoff and drainage computed by the soil hydrology module to the daily time step $t$ of this module. As shown in Fig. 3, surface runoff and drainage initially feed a 'fast' ($S_{fast,H2O}$) and a 'slow' ($S_{slow,H2O}$) water
reservoir, respectively (Eqs. 1,2). The proportions of runoff ($F_{RO,H2O,grid\,x,t}$) and drainage ($F_{DR,H2O,grid\,x,t}$) assigned to each basin $i$ within the grid $x$ are scaled to the area of the basin ($A_{total,i}$) relative to that of the grid cell ($A_{total,grid\,x.}$). $S_{fast}$ and $S_{slow}$ have distinct linear response time scales in each basin of the simulation domain, which are defined by a topographic index $Topo_{grid\,x}$ extracted from a forcing file (values range between 1 and 4 in our study area) and a factor $\tau$ which translates $Topo_{grid\,x}$ into a water residence time of each reservoir (Eqs. 3,4). Following the calibration of Guimberteau et al. (2012), both $\tau_{fast}$ and $\tau_{slow}$ are
set to a value of 3.0 days. The river reservoir ($S_{river}$) in each basin $i$ is mainly fed by the outflows of $S_{fast}$, $S_{slow}$, and $S_{river}$ of the basins $i$-$1$ lying immediately upstream (Eqs. 5,6,7), but can, in addition, interact with two kinds of hydraulic sub-systems, the floodplains and the swamps, the maximum extent of which are defined by forcing files. Swamps are intended to mimic ground water fed wetlands. Where swamps are present, a constant fraction of the upstream inflow $F_{up}$ (Eq. 7), which is scaled to the areal proportion of swamps (*%swamp*) in a given basin $i$, is diverted from the $S_{river}$ and added to the bottom of the soil column
of the *grid x* containing the basin $i$ ($F_{up2swamp}$, Eq. 8). Contrarily to the floodplains, the swamps are not represented by an explicit water body ($S_{flood}$). In the original version of ORCHIDEE, if floodplains are present, all the water coming from upstream not diverted to swamps is first directed to the floodplains ($F_{up2flood}$, Eqs. 9,10, see section 2.1.3 for an improved representation). $S_{flood}$ then sustains a delayed return flow ($F_{flood\,out,H2O}$) to the river reservoir of the same basin $i$ (Eqs. 11,12). The water balance of the $S_{flood}$ is in addition controlled by input from throughfall ($F_{WD,H2O}$), evaporation ($F_{flood2atm,H2O}$), or infiltration into the soil
($F_{flood2soil,H2O}$) in the floodplain (Eq. 11), depending on the temporarily changing areal extent of the inundation *%flood*. The values of $\tau_{river}$ and $\tau_{flood}$ used by *Guimberteau et al.* (2012) are 0.24 days and 2.5 days, respectively. Note that both $F_{flood\,out}$ (Eq. 12) and $F_{river\,out}$ (Eq. 6) are dependent on %flood as well. For further details see the publications of *d'Orgeval et al.* (2008) and *Guimberteau et al.* (2012).

$$S_{fast,H_2O,i,t+1} = S_{fast,H_2O,i,t} + F_{RO,H_2O,grid\ x,t} \cdot \frac{A_{total,i}}{A_{total,grid\ x}} - F_{fast\ out,H_2O,i,t} \tag{1}$$

$$S_{slow,H_2O,i,t+1} = S_{slow,H_2O,i,t} + F_{DR,H_2O,grid\ x,t} \cdot \frac{A_{total,i}}{A_{total,grid\ x}} - F_{slow\ out,H_2O,i,t} \tag{2}$$

$$F_{fast\ out,H_2O,i,t} = \frac{S_{fast,H_2O,i,t}}{\tau_{fast} \cdot Topo_{grid\ x}} \tag{3}$$

$$F_{slow\ out,H_2O,i,t} = \frac{S_{slow,H_2O,i,t}}{\tau_{slow} \cdot Topo_{grid\ x}} \tag{4}$$

$$S_{river,H_2O,i,t+1} = S_{river,H_2O,i,t} + F_{up2river,H_2O,i,t} + F_{flood\ out,H_2O,i,t} - F_{river\ out,H_2O,i,t} \tag{5}$$

$$F_{river\ out,H_2O,i,t} = \frac{S_{river,H_2O,i,t}}{\tau_{river} \cdot Topo_{grid\ x}} \cdot \left(1 - \sqrt{\%flood_{i,t}}\right) \tag{6}$$

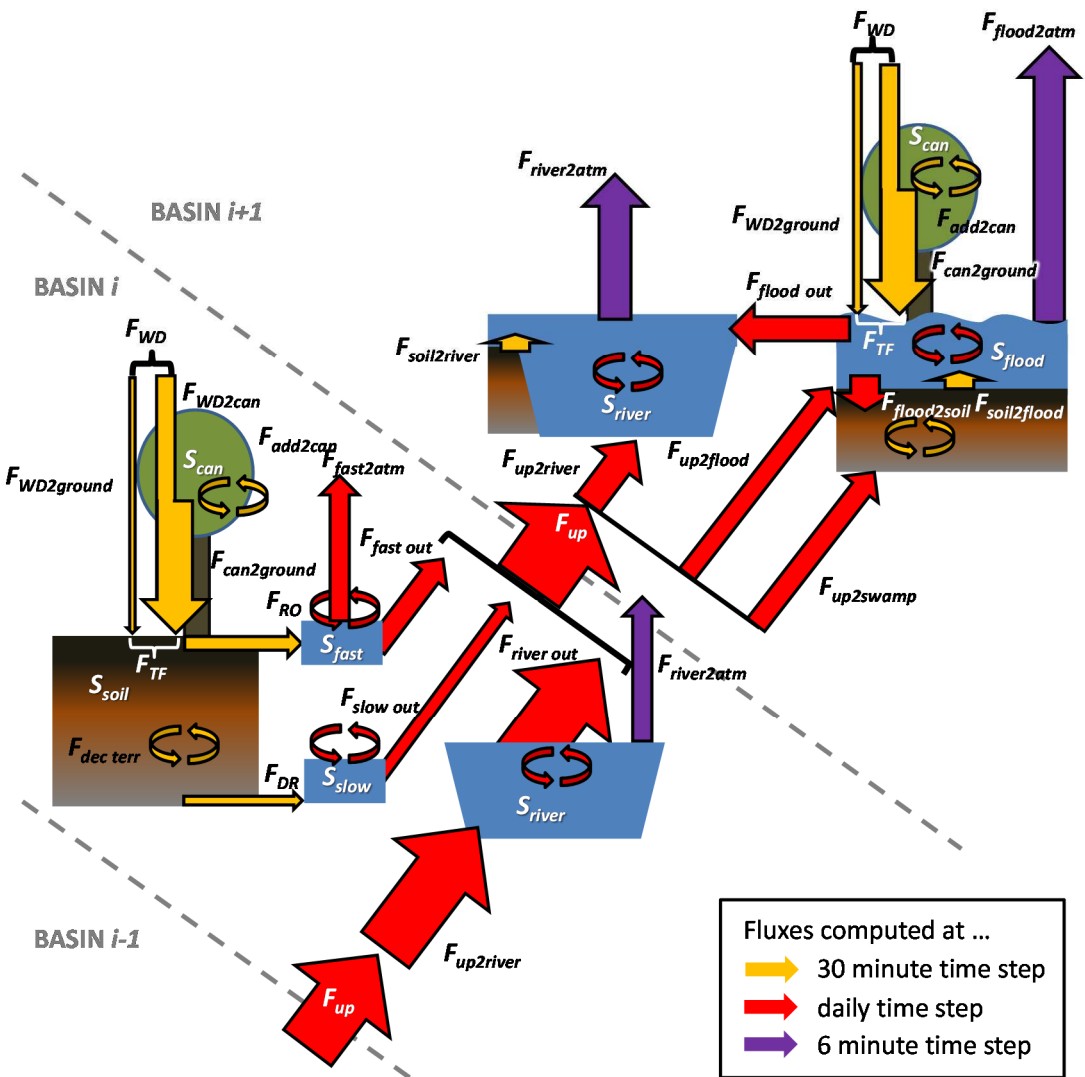

**Figure 3.** Simulated flows of water and C along the vegetation-soil-aquatic continuum. For reasons of simplicity, the fluxes ($F$) and storages ($S$) are characterized by subscripts indicating path or environmental compartment only (see Table A.1). Basin $i$-$1$ is the basin upstream of basin $i$, basin $i$+$1$ is the basin downstream of basin $i$. In this hypothetical example, swamps and floodplains are only present in basin $i$+$1$. The depiction of water and soil-river C fluxes in basins $i$+$1$ and $i$-$1$ were omitted for reasons of readability. Straight arrows represent water and C fluxes between the canopy ($S_{can}$), soil ($S_{soil}$), fast ($S_{fast}$), slow ($S_{slow}$), river ($S_{river}$) and flood ($S_{flood}$) reservoirs. Circular arrows represent carbon transformations within the reservoirs. See text for further details.

$$F_{up,H_2O,i,t} = \sum_{i-1}\left(F_{fast\ out,H_2O,i-1,t} + F_{slow\ out,H_2O,i-1,t} + F_{river\ out,H_2O,i-1,t}\right) \tag{7}$$

$$F_{up2swamp,H_2O,i,t} = f_{swamp} \cdot F_{up,H_2O,i,t} \cdot \%swamp_i \qquad (8)$$

$$F_{up2flood,H2O,i,t} = F_{up,H2O,i,t} - F_{up2swamp,H_2O,i,t} \qquad (9)$$

$$F_{up2river,H2O,i,t} = F_{up,H2O,i,t} - F_{up2swamp,H_2O,i,t} - F_{up2flood,H_2O,i,t} \qquad (10)$$

$$S_{flood,H_2O,i,t+1} = S_{flood,H_2O,i,t} + F_{up2flood,H_2O,i,t} - F_{flood2soil,H_2O,i,t} \qquad (11)$$
$$+ \left(F_{TF,H_2O,i,t} \cdot \%flood_{i,t} - F_{flood2atm,H_2O,i,t}\right) - F_{i,flood\ out,H_2O,t}$$

$$F_{flood\ out,H_2O,i,t} = \frac{S_{flood,H_2O,i,t}}{\tau_{flood} \cdot Topo_{grid\ x}} \cdot \%flood_{i,t} \qquad (12)$$

### 2.1.3 Improved floodplain dynamics

Seasonal flooding in the Amazon is a major control of the hydraulic and C dynamics of the river system (Abril et al., 2014; Melack et al., 2009; Rasera et al., 2013; Richey et al., 1990, 2002). This is particularly true in the central basin where the extent

5   of flooded areas can increase from 4 to 16% of the total area (Hamilton et al., 2002; Hess et al., 2003; Richey et al., 2002). In the following, we first present how flooding is simulated in the trunk-version of ORCHIDEE, summarizing mainly the work of D'Orgeval et al. (2008) and Guimberteau et al. (2012); next we describe improvements in simulated floodplain dynamics undertaken for ORCHILEAK in this study. Flooding is generally simulated in the temporal resolution of the routing module, in the default setting used in this study at the daily time-step.

*Original trunk version*

When floodplains are present in a given basin, all water inputs from upstream basins ($F_{up}$) which are not infiltrating in swamps ($F_{up2swamp}$) are routed to $S_{flood}$ instead of $S_{river}$ (Eq. 9). After floodplain and river reservoirs have been updated with in- and outflows for each basin (Eqs. 5,11), the inundated fraction *%flood* is calculated firstly for each grid-cell, and secondly for each

15  basin within the grid cell. This sequential procedure is necessary, because the maximum floodable proportion *(%flood$_{max}$)*, which is prescribed by the forcing file, is given at the resolution of the grid cells. *%flood* per grid *x* is calculated from the total water storage in the floodplain reservoirs ($S_{flood,H2O,grid\ x,t}$, Eq. 13) of all basins *i* contained in that grid cell, assuming a slightly convex slope of the floodable area (Eqs. 14,15), as this shape is typical of large lowland rivers like the Amazon (Hamilton et al., 2002; Huggett, 2016). In the original version of ORCHIDEE (Fig. 5), the computation is performed as follows: first, a

potential fraction of flooded area (*%flood*$_{pot}$) is calculated based on the total area of the grid cell (*A*$_{total,grid\ x}$) and a potential water level height on the floodplain (*floodcri*, set to 2m by default) for which it is assumed that the whole grid cell is inundated (Eq. 14, Fig 5). The maximum flooded proportion (*%flood*$_{max}$) of the grid cell is defined by values reported in the PRIMA forcing file (see below), that is, *%flood* cannot exceed *%flood*$_{max}$ (Eq. 15). Second, the actual water level over the floodplain area (*floodh*) is calculated from *%flood* and the water storage in the floodplain reservoir *S*$_{flood,H2O}$ (Eq. 16). Finally, the *%flood* of each basin *i* within the grid *x* is calculated based on the *S*$_{flood,H2O}$ of the basin compared to that of the grid box and *A*$_{total}$ of the basin *i* compared to *A*$_{total}$ of grid *x* (Eq. 17).

$$S_{flood,H2O,grid\ x,t} = \sum_i S_{flood,H_2O,i,t} \tag{13}$$

$$\%flood_{pot,grid\ x,t} = \left( \frac{S_{grid\ x,flood,H_2O,t} \cdot 3}{A_{total,grid\ x} \cdot floodcri} \right)^{\frac{2}{3}} \tag{14}$$

$$\%flood_{grid\ x,t} = min\left( \%flood_{pot,grid\ x,t}, \%flood_{max,grid\ x} \right) \tag{15}$$

$$floodh_{grid\ x,t} = \frac{2}{3} \cdot floodcri \cdot \sqrt{\%flood_{grid\ x,t}} + \frac{S_{flood,H_2O,grid\ x,t}}{\left( \%floodh_{grid\ x,t} \cdot A_{total,grid\ x} \right)} \tag{16}$$

$$\%flood_{i,t} = \%flood_{grid\ x,t} \cdot \frac{\left( \frac{S_{flood,H_2O,i,t}}{S_{flood,grid\ x,H2O,t}} \right)}{\left( \frac{A_{total,i}}{A_{total,grid\ x}} \right)} \tag{17}$$

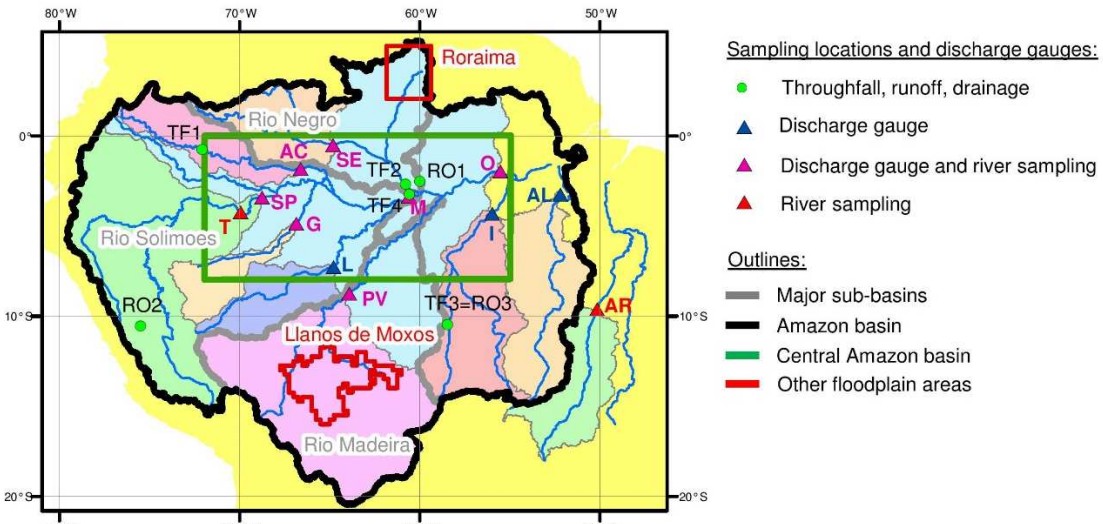

**Figure 4.** Overview of the Amazon Basin, with highlighted boundaries (thick grey) between the three major sub-basins (R. Solimoes, Madeira and Negro). The central Amazon basin (green box) and the sampling locations discussed in this study are also shown. River sampling locations and discharge gauges include: Rio Japura at Acanaui (AC), Rio Xingu at Altamira (AL), Rio Araguaia (AR), Rio Jurua at Gaviao (G), Rio Tapajos at Itaituba (I), Rio Purus at Labrea (L), Rio Solimoes at Manacapuru (M), Amazon River at Obidos (O), Rio Madeira at Porto Velho (PV), Rio Negro at Serrinha (SE), Rio Solimoes at Sao Paulo de Olivenca (SP) and Tabatinga (T). The contributing areas are shown by the different colour codes on the map, except for location T as it is very similar to location SP. The remaining ungauged terrestrial area is represented in yellow. Sampling locations for throughfall DOC are indicated by "TF" and report data from Tobon et al. (2004) (TF1), Filoso et al. (1999) (TF2), Johnson et al. (2006) (TF3) and Williams et al. (1997) (TF4). Sampling location for DOC concentration in surface runoff and/or head waters are indicated by "RO" and report data from Waterloo et al. (2006) (RO1), Saunders et al. (2006) (RO2) and Johnson et al. (2006) (RO3). The red box and red line represent large floodplain areas outside the central Amazon basin for which observations are available.

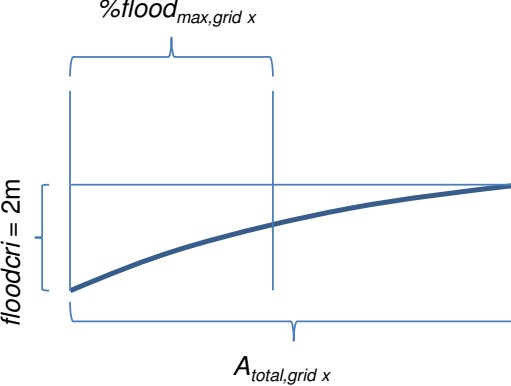

**Figure 5.** Schematic representation of the floodplain dynamics in the trunk version of ORCHIDEE. The bold line corresponds to the assumed shape of the floodplain. In ORCHILEAK, *floodcri* is replaced by *floodh95th*, which represent the 95th percentile of the water level above the floodplain (*floodh*) over the simulation period 1980-2000.

**Table 2.** Data sets used for model evaluation.

| Variable | Spatial resolution | Temporal resolution | Data source |
|---|---|---|---|
| Discharge | Multiple locations | bi-weekly | ORE-HYBAM (Cochonneau et al., 2006) |
| Discharge | Multiple locations | average monthly values | GRDC (Global Runoff Data Center) |
| Inundation in the Central Amazon basin | - | seasonality | Richey et al. (2002) after (Hess et al., 2003) |
| Inundation in Roraima and Llanos de Moxos wetland areas | - | multi-year time-series of monthly values | Hamilton et al. (2011) |
| Soil Organic Carbon stocks | 1:5,000,000 | - | HWSD v 1.1 (FAO et al., 2009) |
| Water temperature | Multiple locations | bi-weekly | ORE-HYBAM (Cochonneau et al., 2006) |
| Riverine DOC concentrations and fluxes | Multiple locations | Irregular time-series | CAMREX (Richey et al., 2008), ORE-HYBAM (Cochonneau et al., 2006), Moreira-Turcq et al. (2003) |
| Seasonality of $CO_2$ evasion from Central Amazon Basin | - | Seasonality with average monthly values | Richey et al. (2002) |
| $CO_2$ evasion rates from the river surface at different sampling locations | - | Multiple values during high and low flow periods | Rasera et al. (2013) |

The PRIMA forcing file was introduced by Guimberteau et al. (2012) to represent the maximum spatial extent of swamps and floodplains at the scale of the entire Amazon basin. The available global wetland (swamps and floodplains) forcings (Lehner and Döll, 2004) are underrepresenting swamp and floodplain areas in this region, and were thus not sufficient to simulate water

retention needed to reproduce the hydrograph of the Amazon River. The PRIMA dataset was obtained using the maximum floodable areas derived from satellite imagery (Prigent et al., 2007), after subtraction of the vegetated proportion reported by Martinez and le Toan (2007). The vegetated part of the maximum floodable area was assigned to 'swamp' areas, which, as stated above, does not include a specific water body in ORCHIDEE.

*Changes in ORCHILEAK*

Although water retention in floodplains was validated by reproducing the water height over the floodplains (Guimberteau et al., 2012), the seasonality in flooded areas extent is still not well captured in the trunk version. Furthermore, according to the PRIMA forcing, the maximum floodable area in the central Amazon basin is < 5%, while according to Richey et al. (2002) the

10 areal proportion of inundated area is comprised between 4 and 16%, leaving a temporarily flooded proportion of 12%. For the simulations with ORCHILEAK, we merged back the swamp and floodplain areas, thus relying directly on the maximum inundated area of Prigent et al. (2007), while, at the same time, keeping swamp areas as zone of return flow from the river to the bottom layer of the soil column (Fig. 6). With this modified forcing, *%flood$_{max}$* increases to 10% within the Central Amazon basin, in better agreement with observations.

To improve the representation of seasonal flooding using updated values of *%flood$_{max}$*, the original equations to calculate the inflow of water to the floodplains and the extent of flooded area in each grid cell were altered as follows. Firstly, floodplains are now only inundated when a threshold in river discharge is exceeded (*F$_{up\ lim}$*, Eq. 18), and it is only the excess part of the river discharge that contributes to the flooding while the remainder is directly entering the next river reservoir (Eq. 19). The threshold is defined for each grid by the median river reservoir water storage of each grid cell over the simulation period (1980-

2000), which is derived in a first simulation with flooding deactivated, and then used as a forcing file for the model (Fig. 6). The choice of the median as threshold provides the advantage of a robust statistical measure and is similar to threshold of 90% of long-term mean discharge used by Vörösmarty et al. (1989) for the Amazon. This modification assumes that a fraction of river water continues to be transported by the river instead of being entirely diverted to the floodplains.

$$F_{up2flood,H_2O,grid\ x,t} = max(F_{up,H_2O,grid\ x,t} - F_{up2swamp,H_2O,grid\ x,t} - F_{up\ lim,H_2O,grid\ x}, 0) \qquad (18)$$

$$F_{up2river,H_2O,grid\ x,t} = min(F_{up,H_2O,grid\ x,t} - F_{up2swamp,H_2O,grid\ x,t}, F_{up\ lim,H_2O,grid\ x}) \qquad (19)$$

$$\%flood_{pot,grid\ x,t} = \left(\frac{S_{flood,H_2O,grid\ x,t} \cdot 3}{Area_{total,grid\ x} \cdot floodh_{grid\ x,95th}}\right)^{\frac{2}{3}} \qquad (20)$$

$$floodh_{grid\ x,t} = \frac{2}{3} \cdot floodh_{grid\ x,95th} \cdot \sqrt{\%flood_{grid\ x,t}} + \frac{S_{flood,H_2O,grid\ x,t}}{(\%flood_{max,grid\ x} \cdot A_{total,grid\ x})} \qquad (21)$$

While the default value for *floodcri*, as to be used in global modelling, was set to 2 m in the trunk version, this value is not applicable to the Amazon, where water levels of up to 12 m have been reported in the Central Amazon floodplain (Trigg et al., 2009). Thus, instead of using a single value for *floodcri* as previously done, we now first compute for each grid cell the

95[th] percentile of all simulated water level heights over the floodplain area for the simulation period 1980-2000 (*floodh95th*, Eq. 21, cf. Fig. 5). We used the regional data set of monthly inundated areas from Hamilton et al. (2011) for validation in the Roraima and Llanos de Moxos wetland areas, which covers part of our simulation period. For inundation in the central Amazon basin, we used the data from Hess et al. (2003) as summarized in Richey et al. (2002) for validation.

$$F_{river\ out,H_2O,i,t} = \frac{S_{river,H_2O,i,t}}{\tau_{river} \cdot Topo_{grid\ x}} \qquad (22)$$

Following the changes in the flooding scheme, we recalibrated two parameters in order to reproduce the monthly discharges from the Amazon and its major tributaries: 1) We decrease the water residence time on the floodplains by changing $\tau_{flood}$ from 2.5 days as used by Guimberteau et al. (2012) to 1.4 days (Eq. 12); and 2) we halved the proportion of water diverted to swamps by setting *fswamp* from 0.2 to 0.1 (Eq. 8), while using the same forcing for %swamp as Guimberteau et al. (2012). In addition, because %flood can now take values close to 100% in some areas, we modified the equation to calculate the outflow

from the river reservoir, which is not decreased anymore depending of %flood (Eq. 22). The simulated river discharges were validated against gauging data from ORE-HYBAM (Cochonneau et al., 2006) and mean monthly discharges provided by the Global Runoff Data Centre (GRDC, n.d.).

In ORCHILEAK, for the purpose of calculating $CO_2$ evasion from the river network, the river reservoir is now assigned a surface area as well (*Ariver*). The base surface area $A_{river}$ (*Ariver basic*) per grid cell is extracted from a forcing file derived from

the global river surface maps of Lauerwald et al. (2015). Following the findings by Rasera et al. (2013), we assume that the surface area of small rivers (*Ariver small*, width < 100m) can increase by about 20% from low to high water stages, whereas the area of larger rivers (*Ariver large*, width ≥ 100m) increases by about 10%. Assuming the 10[th] and 90[th] percentile of $S_{river,H2O}$ over the simulation period 1980-2000 (*Sriver,H2O,grid x,10th*, *Sriver,H2O,grid x,90th*, Fig. 6) as representative for the low and high water stages, an actual $A_{river}$ (*Ariver act*) is calculated at each time-step depending on $S_{river,H20}$ (Eqs. 23-26). As the $A_{river}$ forcings likely

underestimate the total $A_{river}$ (Lauerwald et al., 2015), it is assumed that *Ariver basic* represent $A_{river}$ at low water stage. *Ariver act* per basin *i* is calculated from $A_{river}$ per grid *x* containing that basin, scaling to the square root of $S_{river,H2O}$, because $S_{river,H2O}$ is linearly related to discharge (Eq. 27) and it was empirically shown that stream width scales roughly with the square root of discharge

(Raymond et al., 2012, 2013). Assuming that stream length does not change significantly, the relative change in stream width equals the relative change in $A_{river\ act}$.

$$A_{river\ basic,grid\ x} = A_{river\ small,grid\ x} + A_{river\ large,grid\ x} \qquad (23)$$

If $S_{river,H_2O,grid\ x,t} \leq S_{river,H_2O,grid\ x,10th}$:

$$A_{river\ act,grid\ x,t} = A_{river\ basic,grid\ x} \qquad (24)$$

If $S_{river,H_2O,grid\ x,10th} < S_{river,H_2O,grid\ x,t} < S_{river,H_2O,grid\ x,90th}$:

$$A_{river\ act,grid\ x,t} = \left(1 + \frac{S_{river,H_2O,grid\ x,t} - S_{river,H_2O,grid\ x,10th}}{S_{river,H_2O,grid\ x,90th}} \cdot 0.2\right) \cdot A_{river\ small,grid\ x} \qquad (25)$$
$$+ \left(1 + \frac{S_{river,H_2O,grid\ x,t} - S_{river,H_2O,grid\ x,10th}}{S_{river,H_2O,grid\ x,90th}} \cdot 0.1\right) \cdot A_{river\ large,grid\ x}$$

If $S_{river,H_2O,grid\ x,t} \geq S_{river,H_2O,grid\ x,90th}$:

$$A_{river\ act,grid\ x,t} = 1.2 \cdot A_{river\ small,grid\ x} + 1.1 \cdot A_{river\ large,grid\ x} \qquad (26)$$

$$A_{river\ act,i,t} = A_{river\ act,grid\ x,t} \cdot \frac{\sqrt{S_{river,H_2O,i,t}}}{\sqrt{S_{river,H_2O,grid\ x,t}}} \qquad (27)$$

5   The difference between $A_{river\ act}$ and $A_{river\ basic}$ gives a seasonally flooded area directly adjacent to the river (%$flood_{river}$, Eqs. 28, 29). This flooded area induced by changes in water levels in the river was then added to the total flooded proportion of soils (%$flood_{total}$, Eqs. 30,31). Note, however, that for the calculation of C inputs from flooded soils to the water column (section 2.3), $S_{flood}$ and $S_{river}$ need again to be distinguished.

$$\%flood_{river,grid\ x,t} = \frac{A_{river\ act,grid\ x,t} - A_{river\ basic,grid\ x}}{A_{total,grid\ x}} \qquad (28)$$

$$\%flood_{river,i,t} = \frac{\left(A_{river\ act,grid\ x,t} - A_{river\ basic,grid\ x}\right) \cdot \sqrt{S_{river,H_2O,i,t}}}{A_{total,i} \cdot \sqrt{S_{river,H_2O,grid\ x,t}}} \qquad (29)$$

$$\%flood_{total,grid\ x,t} = \%flood_{grid\ x,t} + \%flood_{river,grid\ x,t} \qquad (30)$$

$$\%flood_{total,i,t} = \%flood_{i,t} + \%flood_{river,i,t} \qquad (31)$$

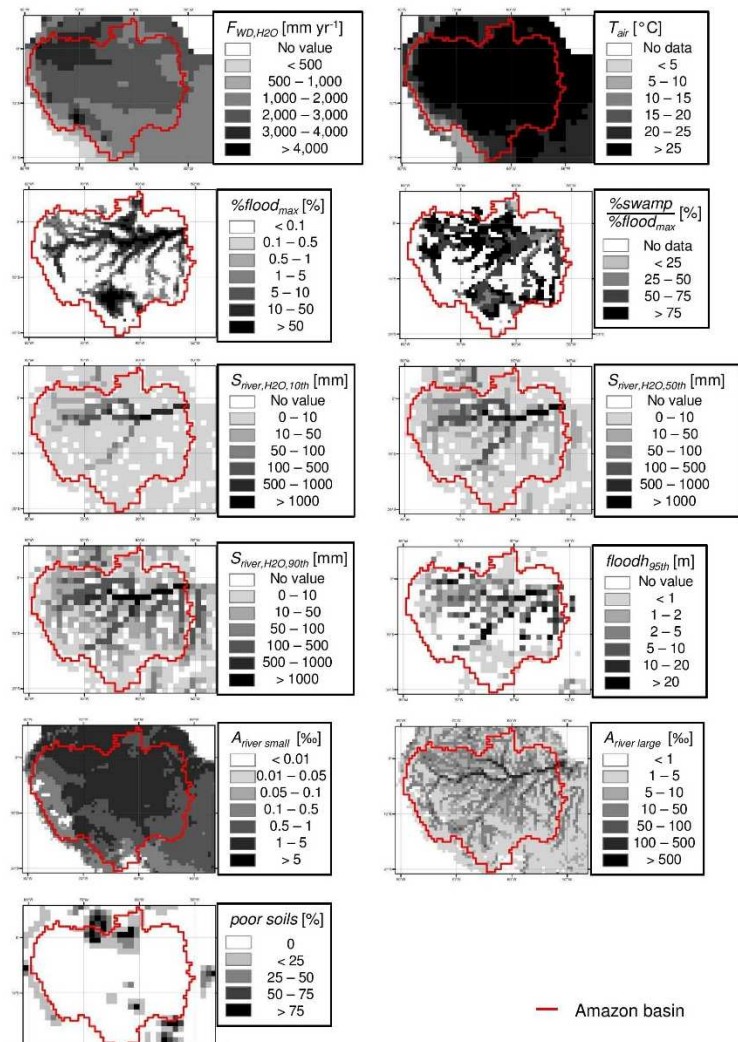

**Figure 6.** Overview of forcing files (cf. Table 2). Climatic forcings comprise, among others, variables like precipitation ($F_{WD,H2O}$) and air temperature ($T_{air}$). The climatic forcings used here are based on the NCC ((Ngo-Duc et al., 2005) data set, only $F_{WD,H2O}$ was replaced by a regional data set created by Guimberteau et al. (2012). The forcing of maximum floodable areas $\%flood_{max}$ was adopted from Guimberteau et al. (2012) after merging swamp areas ($\%swamp$) into $\%flood_{max}$. Simulations of inundation in ORCHILEAK are based on 10th, 50th and 90th percentile of water storage in the river reservoir $S_{river}$ ($S_{river,H2O,10th}$, $S_{river,H2O,50th}$, $S_{river,H2O,90th}$), here given in mm which equals kg H₂O m⁻² assuming a density of water of $10^{-3}$ kg m⁻³, and the 95th percentile of water table level over the floodplains $floodh$ ($floodh_{95th}$), all derived from simulation results over the period 1980 to 2000. Surface areas of small (width < 100 m) and large (width ≥ 100 m) rivers ($A_{river\ small}$, $A_{river\ large}$) are taken from Lauerwald et al. (2015). Of importance for representation of DOC cycling in watersheds of black water rivers is the identification of '*poor soils*' (Podzols, Arenosols and soils in black water swamps), which we derived from the Harmonized World Soil Database (HWSD, FAO/IIASA/ISRIC/ISS-CAS/JRC, 2009) and $\%swamp$.

## 2.2 Carbon dynamics along the vegetation-soil-aquatic continuum

### 2.2.1 Overview of the DOC transport scheme

DOC and $CO_2$ are exported through the terrestrial-water interface by runoff ($F_{RO}$) and drainage ($F_{DR}$), respectively (Fig. 3). Part of the terrestrial DOC stems from throughfall ($F_{TF} = F_{WD2ground} + F_{can2ground}$, see below), the other part stems from the decomposition of litter and soil organic carbon ($F_{dec\ terr}$). DOC exports from flooded areas to the river network are another important source, because $F_{TF}$ and the decomposition of submerged litter and soil carbon in the floodplains ($F_{soil2flood}$) add directly to the DOC storage in the overlying water column and, from there, a delayed flux ($F_{flood\ out}$) feeds $S_{river}$. In addition, streams and rivers extend laterally during high flow periods (see section 2.1.3) and there is thus a direct input of DOC from litter and SOC decomposition on/in seasonally inundated soils immediately adjacent to the stream bed into $S_{river}$ ($F_{soil2river}$).

DOC and $CO_2$ are transported as passive tracers with the fluxes of water through the different reservoirs of the routing scheme (see section 2.1) and can feed back into the soil system via two mechanisms: 1) re-infiltration from the floodplain reservoir into the first layer of the soil column ($F_{flood2soil}$); 2) infiltration of DOC into the bottom layer of the soil column entrained with water entering swamps ($F_{up2swamp}$) (Fig. 3). In addition, DOC is mineralized to $CO_2$ in transit and $CO_2$ is evading to the atmosphere from the water surface. Depending on the relative magnitude between inputs, outputs and in-situ transformations, the storage of DOC in canopy, soil, fast, slow, river and floodplain reservoirs ($S_{can}$, $S_{soil}$, $S_{fast}$, $S_{slow}$, $S_{river}$, and $S_{flood}$) can thus increase or decrease over different time periods. For the routing of DOC, we distinguish two pools, a labile and a refractory pool. Like the cycling of water and C in vegetation and soils, the allochthonous inputs of DOC from $S_{can}$ and $S_{soil}$ into the inland water network ($F_{RO}$, $F_{DR}$, $F_{soil2flood}$, $F_{soil2river}$, see Fig 3) are computed at a temporal resolution of 30 minutes and at the spatial resolution of the grid cell. The lateral transfer between the $S_{fast}$, $S_{slow}$, $S_{river}$ and $S_{flood}$ and the transformation of C within those storage reservoirs are only simulated at a daily time step and at the spatial resolution of the basin. Therefore, to simulate the lateral transfers, the allochthonous DOC and $CO_2$ inputs are first aggregated over 48 30-minute time steps until one full day is over. The fluxes from the water column back into the soil column ($F_{flood2soil}$, $F_{up2swamp}$ in Fig. 3) are simulated at the daily time-step of the routing module, but are used as inputs in the soil carbon module, which runs at a 30 minute temporal resolution. This is achieved by downscaling the daily fluxes uniformly over the 48 30 minute time-steps of the following day of simulation. The evasion of $CO_2$ from river and floodplain water surface ($F_{river2atm}$, $F_{flood2atm}$) is also simulated at the daily time-step of the routing module, but to approximate the continuous interplay of $CO_2$ inputs and $CO_2$ evasion controlling the water-air gradient in $CO_2$ partial pressures ($pCO_2$) a much shorter time-step of 6 minutes is used, and the $CO_2$ inputs to the water column are thus uniformly distributed over the 240 6-minutes time-step contained in each day. The following subsections describe in more detail the simulation of DOC in precipitation and throughfall (section 2.2.2), production of DOC and its export through the terrestrial-aquatic interface (section 2.2.3), $CO_2$ inputs through the terrestrial-aquatic interface (2.2.4), and in-transit DOC mineralisation and $CO_2$ evasion along the inland water network (section 2.2.5).

### 2.2.2 DOC in precipitation and throughfall

Reported average rain DOC concentrations in the Amazon basin are significant with 1.3 to 3.9 mg C L$^{-1}$ (Table 5, in most temperate regions average concentrations < 1 mg C L$^{-1}$ are common, see Michalzik et al. 2001), of the same magnitude as observed concentrations in white and clear water rivers of the region (Moreira-Turcq et al., 2003). The spatial variation in rain

DOC concentration is unknown and we thus assumed a constant value of 2.4 mg C L$^{-1}$ throughout the Amazon basin, from the average of reported literature values (Table 5). Observed average DOC concentrations in throughfall are higher than in precipitation because of the DOC enrichment of leaf-intercepted water due to evaporation losses and dissolution of organic carbon from leaf-leachates and dry deposition. Reported annual throughfall DOC flux ($F_{TF}$) in the Amazonian rain forest varies little, from 14.8 to 19.0 g C m$^{-2}$ yr$^{-1}$ (see Table 5). The temporal variability in throughfall DOC concentrations is mainly

controlled by the amount of throughfall, which acts as a dilution factor, and by the duration of preceding dry periods, which favours the accumulation of soluble organic C on the canopy (Johnson et al., 2006). Here, we used the time-series data on throughfall DOC fluxes in South Amazonia from Johnson et al. (2006) to set up and calibrate a simple model of throughfall DOC fluxes.

In ORCHILEAK, the wet deposition of DOC, $F_{WD}$ is calculated from precipitation and the prescribed constant concentration

of 2.4 mg C L$^{-1}$, which also equals the minimum throughfall concentration in the time-series by Johnson et al. (2006). For each of the 13 ORCHIDEE plant functional types (PFTs) which are potentially present in a grid cell, the wet deposition of DOC onto the canopy ($F_{WD2can}$) and the direct precipitation of DOC onto the ground ($F_{WD2ground}$) directly scales to the corresponding water fluxes simulated in the hydrology module. According to our simulation, $F_{WD}$ contributes to only about one third of the $F_{TF}$ at our calibration site (14.9 g C m$^{-2}$yr$^{-1}$ (Johnson et al., 2006)). Thus we assumed that the unaccounted flux of 10 g C m$^{-2}$

yr$^{-1}$ must originate from dry deposition onto the canopy or leaf leachates. We further assumed that this dry addition of soluble organic carbon ($F_{add2can}$) does not vary over time and scales to the leaf biomass (which, in the model, is directly related to leaf area). Based on the simulated leaf biomass of 457±1 g C m$^{-2}$ for tropical rain forests at the field-site location, we calibrated $F_{add2can}$ at 6*10$^{-5}$ g C per day and per g C in the leaf biomass (Eq. 32). For agricultural and grass lands, we set $F_{add2can}$ to zero. Whenever intercepted water from the canopy falls to the ground ($F_{can2ground}$), the related flux of DOC ($F_{can2ground}$) will empty

the storage of DOC in the canopy ($S_{can}$) at once unless a maximum concentration $DOC_{max}$ of 100 mg DOC kg$_{H2O}$$^{-1}$ (Eq. 33) in $F_{can2ground}$ is exceeded. This value corresponds to the maximum concentration observed by Johnson et al. (2006). Beyond this threshold, $F_{can2ground}$ is set as the product of the water flux and the maximum concentration, and the DOC in excess is assumed to remain in the canopy reservoir $S_{can}$. This threshold prevents unreasonably high DOC concentrations in the first throughfall events after dry periods and allows simulation of progressive depletion of the $S_{can}$ reservoir after a time of significant DOC

accumulation. At each 30 min time step, $F_{WD2can}$, $F_{add2can}$ and $F_{can2ground}$ are calculated and subsequently used to update the DOC storage in the canopy at each *grid x* and PFT *v* (Eq. 34).

$$F_{add2can,DOC,grid\ x,v,t} = leaf\ biomass_{grid\ x,v,t} \cdot 10^{-5} \frac{dt}{day} \tag{32}$$

$$F_{can2ground,DOC,grid\ x,v,t} = \max(F_{can2ground,H_2O,grid\ x,v,t} \cdot 0.1\ g\ kg^{-1} H_2O, S_{can,DOC,grid\ x,v,t}) \tag{33}$$

$$
\begin{aligned}
S_{can,DOC,grid\ x,v,t+1} \\
= S_{can,DOC,grid\ x,v,t} + F_{WD2can,DOC,grid\ x,v,t} + F_{add2can,DOC,grid\ x,v,t} \\
+ F_{can2ground,DOC,grid\ x,v,t}
\end{aligned}
\tag{34}
$$

$$F_{TF,DOC,grid\ x,v,t} = F_{WD2ground,DOC,grid\ x,v,t} + F_{can2ground,DOC,grid\ x,v,t} \tag{35}$$

$$F_{TF,DOC_{lab},grid\ x,v,t} = F_{TF,DOC_{ref},grid\ x,v,t} = 0.5 \cdot F_{TF,DOC,grid\ x,v,t} \tag{36}$$

$F_{TF}$ is calculated as the sum of the non-intercepted wet deposition $F_{WD2ground}$ and $F_{can2ground}$ (Eq 35). Based on the range of values reported in the literature (Aitkenhead-Peterson et al., 2003), we assume that half of the DOC reaching the ground is labile ($DOC_{lab}$) while the other half is refractory ($DOC_{ref}$) (Eq. 36). $F_{TF}$ then infiltrates into the topsoil or adds to $S_{flood}$ in areas where it falls on inundated land (see section 2.2.4).

## 2.2.3 Production and export of soil DOC through the terrestrial-aquatic interface

ORCHILEAK is largely based on ORCHIDEE-SOM, the new soil carbon module simulating microbial production and consumption of DOC, its adsorption and desorption onto/from mineral surfaces, the vertical advective and diffusive fluxes of DOC within the soil profile and the exports of DOC from the soil via surface runoff and drainage (Camino Serrano, 2015). Consistent with the soil hydrology module (Campoy et al., 2013; de Rosnay et al., 2002), the carbon dynamics are resolved using a discretization of a 2m-soil profile into 11 layers geometrically increasing in depth and running at a 30 minutes time-step (Camino Serrano, 2015).

DOC is produced from the decomposition of litter and soil organic carbon (SOC) (Eqs. 37-40), and consumed by further decomposition (Eqs. 41,42). Here, the soil carbon module has been modified to better represent the soil DOC dynamics in the Amazon. First, decomposition on non-flooded ($F_{dec\ terr}$) and flooded ($F_{dec\ flood}$) soils is distinguished, with decomposition rates of the litter, SOC and DOC pools 3 times slower when soils are flooded (Rueda-Delgado et al., 2006). Second, in 'poor soils' characterized by low pH and low nutrient levels such as Podzols, Arenosols or soils located in black water swamps (referred to as Igapo in the Amazon basin), decomposition rates are significantly reduced. Here, we assume a reduction by a factor of 2, following findings from the literature (Bardy et al., 2011; Vitousek and Hobbie, 2000; Vitousek and Sanford, 1986). This feature was implemented in the model by adding a layer defining the areal proportion of 'poor soils' in the soil-forcing file.

The spatial distribution of Podzols and Arenosols was derived from the Harmonized World Soil Data base (FAO/IIASA/ISRIC/ISS-CAS/JRC, 2009). To determine the spatial distribution of Igapo forest soils, we used the PRIMA forcing for swamps in combination with the boundaries of the Rio Negro catchment as derived from the 0.5° river network (Fig. 6).

$$F_{dec\ terr,SOC\ pool,grid\ x,v,l,t} = S_{soil,SOC\ pool,grid\ x,v,l,t} \cdot \frac{k_{SOC\ pool}}{1 + \%poorsoils_{grid\ x}} \cdot \left(1 - \%flood_{total,grid\ x,t}\right) \quad (37)$$

$$F_{dec\ flood,SOC\ pool,grid\ x,v,l,t} = S_{soil,SOC\ pool,grid\ x,v,l,t} \cdot \frac{k_{SOC\ pool}}{1 + \%poorsoils_{grid\ x}} \cdot \frac{\%flood_{total,grid\ x,t}}{3} \quad (38)$$

$$F_{dec\ terr,litter\ pool,grid\ x,v,l,t} = S_{soil,litter\ pool,grid\ x,v,l,t} \cdot \frac{k_{litter\ pool}}{1 + \%poorsoils_{grid\ x}} \cdot \left(1 - \%flood_{total,grid\ x,t}\right) \quad (39)$$

$$F_{dec\ flood,litter\ pool,grid\ x,v,l,t} = S_{soil,litter\ pool,grid\ x,v,l,t} \cdot \frac{k_{litter\ pool}}{1 + \%poorsoils_{grid\ x}} \cdot \frac{\%flood_{total,grid\ x,t}}{3} \quad (40)$$

$$F_{dec\ terr,DOC\ pool,grid\ x,v,l,t} = S_{soil,DOC\ pool,grid\ x,v,l,t} \cdot \frac{k_{DOC\ pool}}{1 + \%poorsoils_{grid\ x}} \cdot \left(1 - \%flood_{total,grid\ x,t}\right) \quad (41)$$

$$F_{dec\ flood,DOC\ pool,grid\ x,v,l,t} = S_{soil,DOC\ pool,grid\ x,v,l,t} \cdot \frac{k_{DOC\ pool}}{1 + \%poorsoils_{grid\ x}} \cdot \frac{\%flood_{total,grid\ x,t}}{3} \quad (42)$$

The soil carbon module distinguishes 3 different pools of DOC depending on the source material: active, slow and passive (Camino Serrano, 2015). The DOC derived from the active SOC pool and metabolic litter is assigned to the active DOC pool, while the DOC derived from the slow and passive SOC pools are assigned to the slow and passive DOC pools, respectively (Eqs. 43-45). A part of DOC derived from structural plant litter, which is related to the lignin structure of the litter pool (Krinner et al., 2005), is allocated to the slow DOC pool, while the remainder feeds the active DOC pool. The proportion of the

decomposed litter and SOC that is transformed into DOC instead of $CO_2$ depends on the carbon use efficiency (*CUE*), set here to a value of 0.5 (Manzoni et al., 2012). Taken that the same residence time for the slow and passive DOC pools is used in ORCHIDEE-SOM (Camino Serrano, 2015), we merge these two pools when computing throughfall and lateral transport of DOC. Thus, the labile pool is identical to the active pool of the soil carbon module, while the refractory pool combines the slow and passive pools. The labile ($F_{TF,DOClab}$) and refractory ($F_{TF,DOCref}$) proportions of throughfall DOC are added to the

active and slow DOC pools of the first soil layer, respectively.

$$\Delta S_{soil,DOC\ active,grid\ x,v,t} \quad (43)$$

$$= \sum_{l=1}^{11} \left( \left( F_{dec\ terr,litter\ str,grid\ x,v,l,t} + F_{dec\ flood,litter\ str,grid\ x,v,l,t} \right) \right.$$

$$\cdot \left(1 - \%lignin_{grid\ x,v,l,t}\right) + F_{dec\ terr,litter\ met,grid\ x,v,l,t} + F_{dec\ flood,litter\ met,grid\ x,v,l,t}$$

$$\left. + F_{dec\ terr,SOC\ active,grid\ x,v,l,t} + F_{dec\ flood,SOC\ active,grid\ x,v,l,t} \right) \cdot CUE$$

$$- \sum_{l=1}^{11} \left( F_{dec\ terr,DOC\ labile,grid\ x,v,l,t} + F_{dec\ flood,DOC\ labile,grid\ x,v,l,t} \right)$$

$$+ F_{TF,DOC_{lab},grid\ x,v,t} \cdot \left(1 - \%flood_{grid\ x,t}\right) - F_{RO,DOC\ active,grid\ x,v,t}$$

$$- F_{DR,DOC\ active,grid\ x,v,t} - F_{Flood\ inp,DOC\ active,grid\ x,v,t}$$

$$\Delta S_{soil,DOC\ slow,grid\ x,v,t} \quad (44)$$

$$= \sum_{l=1}^{11} \left( \left( F_{dec\ terr,litter\ str,grid\ x,v,l,t} + F_{dec\ flood,litter\ str,grid\ x,v,l,t} \right) \right.$$

$$\cdot \%lignin_{grid\ x,v,l,t} + F_{dec\ terr,SOC\ slow,grid\ x,v,l,t} + F_{dec\ flood,SOC\ slow,grid\ x,v,l,t}$$

$$\left. + F_{dec\ terr,SOC\ passive,grid\ x,v,l,t} + F_{dec\ flood,SOC\ passive,grid\ x,v,l,t} \right) \cdot CUE$$

$$- \sum_{l=1}^{11} \left( F_{dec\ terr,DOC\ slow,grid\ x,v,l,t} + F_{dec\ flood,DOC\ slow,grid\ x,v,l,t} \right) + F_{TF,DOC_{ref},grid\ x,v,t}$$

$$\cdot \left(1 - \%flood_{grid\ x,t}\right) - F_{RO,DOC\ slow,grid\ x,v,t} - F_{DR,DOC\ slow,grid\ x,v,t}$$

$$- F_{soil2flood,DOC\ slow,grid\ x,v,t}$$

$$\Delta S_{soil,DOC\ passive,grid\ x,v,t} \quad (45)$$

$$= \sum_{l=1}^{11} \left( F_{dec\ terr,SOC\ passive,grid\ x,v,l,t} + F_{dec\ flood,SOC\ passive,grid\ x,v,l,t} \right) \cdot CUE$$

$$- \sum_{l=1}^{11} \left( F_{dec\ terr,DOC\ passive,grid\ x,v,l,t} + F_{dec\ flood,DOC\ passive,grid\ x,v,l,t} \right)$$

$$- F_{RO,DOC\ passive,grid\ x,v,t} - F_{DR,DOC\ passive,grid\ x,v,t} - F_{soil2flood,DOC\ passive,grid\ x,v,t}$$

Alongside with decomposition, DOC is lost from the soil column through lateral exports with surface runoff and / or drainage, which occur at the top and bottom of the soil column, respectively. The DOC export by drainage at the bottom of the soil is proportional to the DOC concentration in the deepest (11[th]) soil layer (Eq. 46). Surface runoff occurs when the maximum infiltration rate is exceeded, beyond which the excess water does not enter the soil column anymore. Because the first soil layers are extremely thin, it is assumed here that surface runoff can entrain DOC from the first five layers of the soil column,

which together have a thickness of 4.5 cm (Eq. 47). In each basin, the DOC release is proportional to the mean DOC concentration in this zone of the soil column as well as to the areal extent of the saturated zone around headwaters, as detailed below. To simulate the DOC production in flooded areas, we assume that the DOC produced from the decomposition of litter and SOC within these same 5 topsoil layers adds directly to the DOC storage in the overlying surface water body $S_{flood}$ (see Fig. 3, Eqs. 48-50). Accordingly, the inputs of DOC to the non-flooded soils via $F_{dec\ terr}$ are estimated using the non-flooded proportion of the grid cell (1-%$flood_{i,t}$) (Eqs. 37, 39, 41).

$$F_{DR,DOC\ pool,grid\ x,v,t} = min(F_{DR,H2O,i,v,t} \cdot \frac{S_{soil,DOC\ pool,grid\ x,v,l=11,t}}{S_{soil,H_2O,grid\ x,v,l=11,t}}, \tag{46}$$

$$S_{soil,DOC\ pool,grid\ x,v,l=11,t})$$

$$F_{RO,DOC\ pool,grid\ x,v,t} = min(F_{RO,H2O,grid\ x,v,t} \cdot \frac{\Sigma_{l=1}^{5} S_{soil,DOC\ pool,grid\ x,v,l,t}}{\Sigma_{l=1}^{5} S_{soil,H_2O,grid\ x,v,l,t}} \cdot red_{RO,grid\ x,t}, \tag{47}$$

$$\Sigma_{l=1}^{5} S_{soil,DOC\ pool,grid\ x,v,l,t})$$

$$F_{soil2flood,DOC\ active,grid\ x,v,t} \tag{48}$$
$$= \sum_{l=1}^{5} \left( F_{dec\ flood,litter\ str,grid\ x,v,l,t} \cdot \left(1 - \%lignin_{grid\ x,v,l,t}\right) \right.$$
$$\left. + F_{dec\ flood,litter\ met,grid\ x,v,l,t} + F_{dec\ flood,SOC\ active,grid\ x,v,l,t}\right) \cdot CUE$$

$$F_{soil2flood,DOC\ slow,grid\ x,v,t} \tag{49}$$
$$= \sum_{l=1}^{5} \left( F_{dec\ flood,litter\ str,grid\ x,v,l,t} \cdot \%lignin_{grid\ x,v,l,t} \right.$$
$$\left. + F_{dec\ flood,SOC\ slow,grid\ x,v,l,t}\right) \cdot CUE$$

$$F_{soil2flood,DOC\ passive,grid\ x,v,t} = \sum_{l=1}^{5} F_{dec\ flood,SOC\ passive,grid\ x,v,l,t} \cdot CUE \tag{50}$$

The usually higher DOC concentration in the topsoil compared to the subsoils is mainly due to the higher inputs of plant litter into and onto the topsoil. However, DOC is efficiently transported between the soil layers along with the vertical flow of water through the soil matrix ($F_{soil\ adv}$, Eqs. 51-52). Therefore, a part of the DOC exported with the drainage is not produced in-situ but rather originates from percolation across the entire soil column. The vertical DOC transport within the soils, as well as for the export of DOC with surface runoff are not directly computed as the product of water flux and DOC concentration. Instead, a reduction factor ($red_{DOC}$) is applied to account for the effect of preferential vertical flow paths, e.g. along macrospores

produced by the root system (Karup et al., 2016), and zones of reduced flow rates which increase the DOC residence time in the remaining parts of the soil. Only in "poor soils" the flow of DOC is not reduced relative to the flow of water (no reduction, Eq. 54). This allows to account for their poor filtering capacity which is the cause of the very high DOC concentrations in groundwater below Podzols and black water swamps (Brinkmann, 1984; McClain et al., 1997). While the effect of preferential flow path should be envisioned as a general concept in ORCHILEAK, the introduction of 'poor soils' is specific to tropical black water systems. It remains to be shown in future work how their effects will have to be parametrized in other climate zones, for instance in the Boreal zone where Podzols are abundant.

$$F_{soil\ adv,DOC\ pool,grid\ x,v,l\rightarrow l+1,t}$$
$$= max\left(F_{soil\ adv,H_2O,grid\ x,v,l\rightarrow l+1,t} \cdot \frac{S_{soil,DOC\ pool,grid\ x,v,l,t}}{S_{soil,H_2O,grid\ x,v,l,t}} \cdot red_{DOC,grid\ x}, 0\right)$$
(51)

$$F_{soil\ adv,DOC\ pool,grid\ x,v,l\rightarrow l-1,t}$$
$$= max\left(F_{soil\ adv,H_2O,grid\ x,v,l\rightarrow l-1,t} \cdot \frac{S_{soil,DOC\ pool,grid\ x,v,l,t}}{S_{soil,H_2O,grid\ x,v,l,t}} \cdot red_{DOC,grid\ x}, 0\right)$$
(53)

$$red_{RO,grid\ x,v,t} = red_{DOC,grid\ x} \cdot red_{connect,grid\ x,t}$$
(53)

$$red_{DOC,grid\ x} = (1 - \%poorsoils) \cdot red_{DOC,base} + \%poorsoils$$
(54)

$$red_{connect,grid\ x,t} = min\left(\frac{(S_{fast,H2O,grid\ x,t} + S_{slow,H2O,grid\ x,t})^{0.5}}{S_{fast+slow,H_2O,ref}^{0.5}}, 1\right)$$
(55)

DOC exports with surface runoff is even further reduced, because the riverine DOC mostly derives from saturated soils in direct vicinity to surface waters (Idir et al., 1999). As we do not have direct information on the density of headwater streams at small scale and the extent of the saturated, riparian zone, the reduction in DOC exports with surface runoff ($red_{connect}$) was scaled to the storage of water in $S_{fast}$ and $S_{slow}$ (Eq. 55). We assumed these reservoirs to represent the water stored in groundwater and headwater streams ($S_{river}$ being attributed to wider water bodies due to the coarse resolution (0.5°) of the river network). Next, based on model calibration, we set a threshold value for the sum of $S_{fast,H2O}$ and $S_{slow,H2O}$ ($S_{fast+slow,H_2O,ref}$) at which a 100% connection between top soils and headwaters is achieved. When $S_{fast+slow,H_2O,ref}$ does not reach the threshold, a lower proportion of topsoil is in connection with the headwaters. Consistent with our approach in section 2.1.3, we assumed here that the extent of saturated soils around headwaters (i.e. the connected topsoils) increases linearly with the square root of

the sum of $S_{fast,H2O}$ and $S_{slow,H2O}$. Finally, the maximum amount of DOC that can be exported through surface runoff and drainage is limited by the storage of DOC in the top and bottom soil layers (Eqs. 46-47).

### 2.2.4 Export of dissolved $CO_2$ through the soil-aquatic network interface

Although mineralization of litter, SOC, DOC in the soil are simulated in ORCHIDEE, the $CO_2$ partial pressure in the soil air and soil solution of the different layers is not represented. Thus, we implemented simple estimates of these soil-derived $CO_2$ inputs in order to reproduce the observed $CO_2$ evasion fluxes from the water surface of the fluvial network. For simulating the export of $CO_2$ with surface runoff and drainage, we use fixed concentrations of 20 mg C $L^{-1}$ (pCO2 of 50,000 μatm at 25°C) and 2 mg C $L^{-1}$ (pCO2 of 5,000 μatm at 25°C), respectively, derived from reported literature values (Davidson et al., 2010; Johnson et al., 2008; Saunders et al., 2006). The lateral exports of $CO_2$ dissolved in soil water are then calculated by multiplying these $CO_2$ concentrations with the water fluxes from surface runoff and drainage simulated at half-hourly time-step in the soil hydrology module (Eqs. 56,57). Next, the computed lateral fluxes of $CO_2$ exported out of soils are subtracted from the total soil respiration and the remainder, by far the dominant fraction (Davidson et al., 2010), is assumed to evade directly to the atmosphere through the topsoil (Eq. 58). Carbonate chemistry and export of alkalinity are neglected.

$$F_{RO,CO_2,grid\ x,t} = F_{RO,H_2O,grid\ x,t} \cdot w_{RO,CO_2} \tag{56}$$

$$F_{DR,CO_2,grid\ x,t} = F_{DR,H_2O,grid\ x,t} \cdot w_{DR,CO_2} \tag{57}$$

$$F_{soil2atm,CO_2,grid\ x,t} = F_{soil\ resp,CO_2,grid\ x,t} - F_{RO,CO_2,grid\ x,t} - F_{DR,CO_2,grid\ x,t} \tag{58}$$

$$
\begin{aligned}
F_{soil2flood,CO_2,grid\ x,t} & \\
= \sum_{v=1}^{13} &\left( \sum_{l=1}^{11} \left( \left( \sum_{litter\ pool} F_{dec\ flood,litter\ pool,grid\ x,v,l,t} \right. \right. \right. \\
& + \sum_{SOC\ pool} F_{dec\ flood,SOC\ pool,grid\ x,v,l,t} + \sum_{DOC\ pool} F_{dec\ flood,DOC\ pool,grid\ x,v,l,t} \bigg) \\
& \left. \left. \cdot (1 - CUE) \right) + F_{root\ respiration,grid\ x,v,t} \cdot \%flood_{grid\ x,t} \right)
\end{aligned}
\tag{59}
$$

In floodplains, mineralisation of submerged litter and soil carbon are considered to be sources of $CO_2$ to $S_{flood}$ (Eq. 59). In addition, we allocated the root respiration in inundated areas to the "$CO_2$ inputs to $S_{flood}$" term. The lateral transfer of $CO_2$ by advection and the re-infiltration of dissolved $CO_2$ into swamps and on floodplains are simulated following the approach implemented for DOC (Fig. 3, and preceding subsections).

## 2.2.5 Carbon transport and transformation along the inland water network

Transport and transformation of terrestrially derived C in the river system are implemented into the river routing module. The lateral transport of DOC and $CO_2$ between reservoirs are assumed to be proportional to the water fluxes, that is, the exports from each reservoir to the next have the same concentration of DOC and $CO_2$ as in the reservoir from which they originate (Eq. 60). The same holds true for infiltration on the floodplains ($F_{flood2soil}$, Eq. 61). The inputs from upstream $F_{up}$ are the sum of $F_{fast\,out}$, $F_{slow\,out}$, $F_{river\,out}$ of all basins $i\text{-}1$ lying directly upstream (Eq. 62), and inputs into swamps ($F_{up2swamp}$, Eq. 63), $S_{flood}$ ($F_{up2flood}$, Eq. 64) and $S_{river}$ ($F_{up2river}$, Eq. 65) have all the same concentrations as $F_{up}$.

$$F_{*\,out,C\,spec,i,t} = F_{*\,out,H_2O,i,t} \cdot \frac{S_{*,C\,spec,i,t}}{S_{*,H2O,i,t}} \tag{60}$$

*: 'fast', 'slow', 'stream', or 'flood' reservoir; C spec: $DOC_{lab}$, $DOC_{ref}$, $CO_2$

$$F_{flood2soil,C\,spec,i,t} = F_{flood2soil,H_2O,t} \cdot \frac{S_{flood,C\,spec,i,t}}{S_{flood,H_2O,i,t}} \tag{61}$$

$$F_{up,C\,spec,i,t} = \sum_{i-1}(F_{fast\,out,C\,spec,i-1,t} + F_{slow\,out,C\,spec,i-1,t} + F_{river\,out,C\,spec,i-1,t} \tag{62}$$

$$F_{up2swamp,C\,spec,t} = F_{up2swamp,H_2O,t} \cdot \frac{F_{up,C\,spec,i,t}}{F_{up,H_2O,i,t}} \tag{63}$$

$$F_{up2flood,C\,spec,i,t} = F_{up2flood,H_2O,t} \cdot \frac{F_{up,C\,spec,i,t}}{F_{up,H_2O,i,t}} \tag{64}$$

$$F_{up2river,C\,spec,i,t} = F_{up,C\,spec,i,t} - F_{up2swamp,C\,spec,i,t} - F_{up2flood,C\,spec,i,t} \tag{65}$$

As discussed above, in the routing scheme, we distinguish two pools of DOC: the labile ($DOC_{lab}$), which corresponds to the active DOC pool of the soil carbon module, and the refractory pool ($DOC_{ref}$), which combines the slow and passive pool of the soil carbon module. For each pool, the DOC stocks in $S_{fast}$ and $S_{low}$ are then updated from the balance between the C inputs simulated in the soil carbon module at 30 minute time-step and aggregated to the one day time step of the routing module, and the outflows of C which are proportional to the water fluxes (Eqs. 66, 67). $S_{river}$ in basin $i$ is augmented by the sum of outflows from the fast, slow and river reservoirs of the basins located directly upstream ($i\text{-}1$), minus the flows diverted to the subsoil of

swamps and into floodplains (Eq. 68). The floodplains ($S_{flood}$) receive inputs from upstream ($F_{up2flood}$) and transfer C to the river reservoir ($F_{flood\ out}$) and via infiltration into the soil ($F_{flood2soil}$) (Eq. 69). The inputs of DOC from the decomposition of inundated SOC and litter are added to $S_{river}$ and $S_{flood}$ according to their contribution to the total fraction of inundated soil (%$flood_{total}$).

$$S_{fast,C\ spec,i,t+1} = S_{fast,C\ spec,i,t} + F_{RO,C\ spec,i,t} - F_{fast\ out,C\ spec,i,t} \tag{66}$$

$$S_{slow,C\ spec,i,t+1} = S_{slow,C\ spec,i,t} + F_{DR,C\ spec,i,t} - F_{slow\ out,C\ spec,i,t} \tag{67}$$

$$S_{river,C\ spec,i,t+1} = S_{river,C\ spec,i,t} + F_{up2river,C\ spec,i,t} + F_{flood\ out,C\ spec,i,t} - F_{river\ out,C\ spec,i,t} \tag{68}$$
$$+ \sum_{v=1}^{13} \left( F_{soil2flood,C\ spec,grid\ x,v,t} \right) \cdot \frac{dt}{day} \cdot \frac{\%flood_{river,i,t} \cdot A_{total,i}}{\%flood_{total,grid\ x,t} \cdot A_{total,grid\ x}}$$

$$S_{flood,C\ spec,i,t+1} = S_{flood,C\ spec,i,t} + F_{up2flood,C\ spec,i,t} - F_{flood2soil,C\ spec,i,t} + F_{TF,C\ spec,i,t} \cdot \%flood_{i,t} \tag{69}$$
$$- F_{flood\ out,C\ spec,i,t} + \sum_{v=1}^{13} \left( F_{soil2flood,C\ spec,grid\ x,v,t} \right) \cdot \frac{dt}{day}$$
$$\cdot \frac{\%flood_{i,t} \cdot A_{total,i}}{\%flood_{total,grid\ x,t} \cdot A_{total,grid\ x}}$$

For Eqs. 68, 69: $F_{soil2flood}$ only for DOC; for $CO_2$, see Eqs. 83, 84

At each daily time-step, after the lateral transfers along the flow path have been calculated, DOC decomposition and $CO_2$ evasion within the river and floodplain reservoirs are simulated. The continuous $CO_2$ production and $CO_2$ evasion from the aquatic network are computed using a much finer integration time step of 1/240 day (6 min) than the one of the river routing scheme to ensure precision of our numerical scheme. In addition, $CO_2$ inputs from the decomposition from flooded SOC and litter are also added at the same time-step to represent the continuous additions of $CO_2$ during the water-atmosphere gas
exchange.

For each 6-min time step, the $pCO_2$ in the water column is calculated from the concentration of dissolved $CO_2$ and the temperature dependent solubility of $CO_2$ ($K_{CO2}$) (Eq. 70). The water temperature ($T_{water}$) needed to calculate $K_{CO2}$ (Telmer and Veizer, 1999) (Eq. 71) is derived from the average air temperature close at the ground ($T_{ground}$) over the whole one-day-time-step of the routing scheme  (Eq. 72, $R^2$=0.56, $\sigma$=0.91°C). This equation was empirically derived using values from the ORE-
HYBAM dataset (Cochonneau et al., 2006) observed at a 10 day interval over the years 1999 and 2000 at 3 sampling locations (Fig. 7, see Fig. 4 for location). As the linear fits for each sampling location are quite similar (Fig. 6 a), we consider the prediction equation derived for the total of observed data as representative. Note that the slope is quite similar to that (0.82)

found by Lauerwald et al. (2015) for average monthly $T_{water}$ using a global data set. Furthermore, we investigated whether the correlations could be improved by introducing a time-lag between $T_{water}$ and $T_{ground}$, as suggested in the literature (Ducharne, 2008; Van Vliet et al., 2011). However, no significant improvement could be achieved (Fig. 7 b), and we thus maintained Eq. (72) as predictor of water temperature.

$$pCO_{2H_2O,*,i,t} = \frac{S_{*,CO_2,i,t}}{S_{*,H_2O,i,t} \cdot 12.011 \cdot K_{CO_2}} \tag{70}$$

\* stands for slow, fast, river, flood

$$K_{CO_2,i,t} = 10^{(2.22 \cdot 10^{-6} \cdot T_{water,grid\,x,t}^{3} + 1.91 \cdot 10^{-5} \cdot T_{water,grid\,x,t}^{2} + 1.63 \cdot 10^{-2} \cdot T_{water,grid\,x,t} - 1.11)} \tag{71}$$

$$T_{water,grid\,x,t} = 6.13°C + 0.80 \cdot T_{ground,grid\,x,t} \tag{72}$$

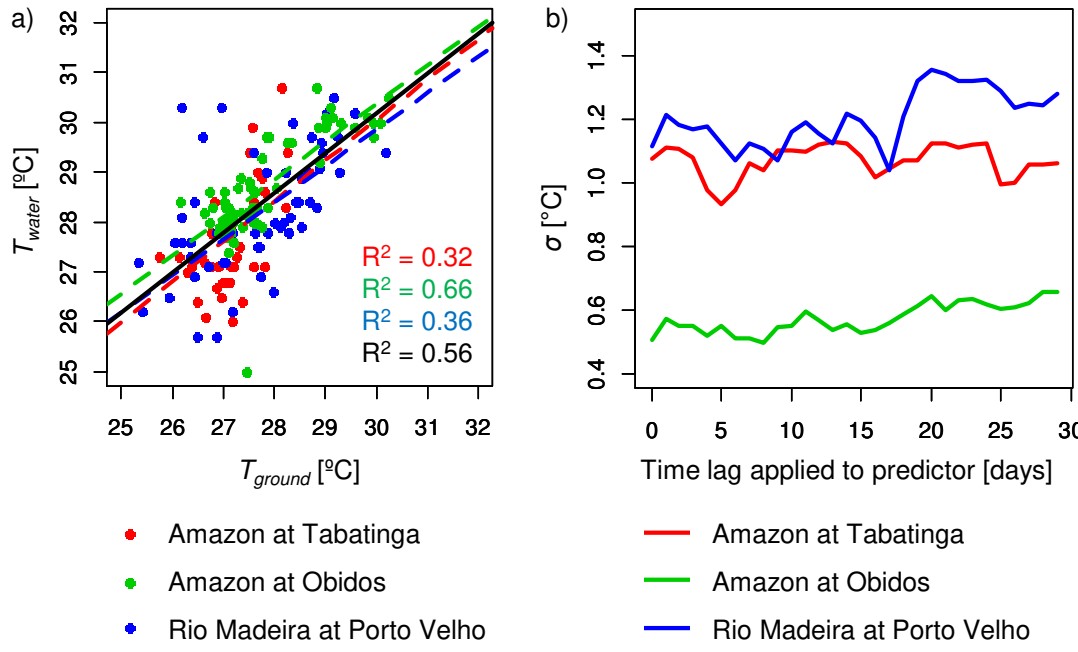

**Figure 7.** Predictability of water temperature ($T_{water}$) from simulated ground temperature ($T_{ground}$). a) Linear regressions between $T_{water}$ and $T_{ground}$ recorded on the same day. The black line represents the linear fit through all data combined, while the coloured dashed lines represent

the linear fits per sampling location. b) Changes in RMSE ($\sigma$) of the prediction equation per sampling location after applying different time lags to the predictor, $T_{ground}$.

The same water temperature is used for the calculation of the Schmidt number (*SC*) (Wanninkhof, 1992) (Eq. 73), which is needed to calculate the actual gas exchange velocity from the standard conditions $k_{600}$ (Eqs. 74, 75). We used distinct values

of $k_{600}$ for rivers ($k_{river,600}$), and for swamps ($k_{swamp,600}$) to account for the reduced effect of the wind in flooded forests. The value $k_{swamp,600} = 0.65$ m d$^{-1}$ is taken from Richey et al. (2002) while the value for $k_{river,600} = 3.5$ m d$^{-1}$ corresponds to the average of the values reported in Alin et al. (2011). For the calculation of $k_{flood,600}$ on the floodplains, we assumed that open floodplains have the same gas exchange velocity than the rivers, while within flooded forests (represented by *%swamp*), the gas exchange velocity is set to $k_{swamp,600}$. As the gas exchange is calculated for the whole floodplain, and is thus a combination of open-water

floodplain and swamps, the average $k_{flood}$ is calculated according to the vegetated and open proportions (Eq. 75). In rivers and floodplains, the CO$_2$ evasion is calculated based on the $pCO_2$, the gas exchange velocity, and the surface water area available for gas exchange, which changes at the daily time-step (Eqs. 76, 77). The maximum possible CO$_2$ evasion per time-step is constrained by the amount of dissolved CO$_2$ in excess to the hypothetical equilibrium with the atmospheric $pCO_2$. For $S_{fast}$, for which a surface area is not known, full equilibration with the atmosphere is assumed (Eq. 78). For $S_{slow}$, which we consider as

groundwater storage even though a ground water table itself is not simulated, no gas exchange is assumed.

$$SC_{i,t} = 1911 - 118.11 \cdot T_{water,grid\ x,t} + 3.453 \cdot T_{water,grid\ x,t}^2 - 0.0413 \cdot T_{water,grid\ x,t}^3 \tag{73}$$

$$k_{river,i,t} = k_{river,6oo} \cdot \sqrt{\frac{600}{SC_{i,t}}} \tag{74}$$

$$k_{flood,i,t} = \left(\left(1 - \frac{\%_{swamp}}{\%flood_{max}}\right) \cdot k_{river,6oo} + \left(\frac{\%_{swamp}}{\%flood_{max}}\right) \cdot k_{swamp,600}\right) \cdot \sqrt{\frac{600}{SC_{i,t}}} \tag{75}$$

$$F_{river2atm,CO_2,i,t} = \min(K_{CO_2,i,t} \cdot \left(pCO_{2\ river,i,t} - pCO_{2\ atm,t}\right) \cdot 12.011 \cdot A_{river\ act,i,t} \cdot k_{river,CO_2,i,t} \cdot \frac{dt}{day} \tag{76}$$
$$\cdot\ 10^3, K_{CO_2,i,t} \cdot \left(pCO_{2\ river,i,t} - pCO_{2\ atm,t}\right) \cdot 12.011 \cdot S_{river,H_2O,i,t} \cdot 10^3)$$

$$F_{flood2atm,CO_2,i,t} = \min(K_{CO_2,i,t} \cdot \left(pCO_{2\ flood,i,t} - pCO_{2\ atm,t}\right) \cdot 12.011 \cdot A_{flood,i,t} \cdot k_{flood,CO_2,i,t} \cdot \frac{dt}{day} \tag{77}$$
$$\cdot\ 10^3, K_{CO_2,i,t} \cdot \left(pCO_{2\ flood,i,t} - pCO_{2\ atm,t}\right) \cdot 12.011 \cdot S_{flood,H_2O,i,t} \cdot 10^3)$$

$$F_{fast2atm,CO_2,i,t} = K_{CO_2,i,t} \cdot \left( pCO_{2\,fast,i,t} - pCO_{2\,atm,t} \right) \cdot 12.011 \cdot S_{fast,H_2O,i,t} \cdot 10^3 \tag{78}$$

The instream decomposition of terrestrial DOC is calculated using base rate constants for labile and refractory DOC, $k_{DOClab}$ = 0.3 day$^{-1}$ and $k_{DOCref}$ = 0.01 day$^{-1}$, respectively (Eqs. 79, 80). These values correspond to half-live times of 2 days and 80 days respectively. The value for $k_{DOClab}$ is thus in agreement with Devol and Hedges (2001) who conclude that $DOC_{lab}$ in the Amazon

5    river must have a very short half-life of hours to a few days. $k_{DOCref}$ also corresponds to the lower range of respiration rates found for Rio Solimoes of 0.2 µM h$^{-1}$ (Amon and Benner, 1996) if an average concentration of about 5 mg C L$^{-1}$ is assumed (cf. Moreira-Turcq et al., 2003). We assumed that the values for the rate constants are valid for an average $T_{water}$ of 28°C (consistent with experiments of Amon and Benner, 1996 and the average temperature simulated here) and apply a temperature sensitivity factor on decomposition rates after Hanson et al. (2011) (Eqs. 79,80).

$$S_{*,\text{DOC}_{lab},i,t+1} = S_{*,\text{DOC}_{lab},i,t} - S_{*,\text{DOC}_{lab},i,t} \cdot \frac{k_{DOC_{lab}} \cdot dt}{day} \cdot 1.073^{(T_{water,i,t}-28)} \tag{79}$$

$$S_{*,\text{DOC}_{ref},i,t+1} = S_{*,\text{DOC}_{ref},i,t} - S_{*,\text{DOC}_{ref},i,t} \cdot \frac{k_{DOC_{ref}} \cdot dt}{day} \cdot 1.073^{(T_{water,i,t}-28)} \tag{80}$$

*: fast, slow, river, flood

$$S_{slow,\text{CO2},i,t+1} = S_{slow,\text{CO2},i,t+1} + S_{slow,\text{DOC}_{lab},i,t} \cdot \frac{k_{DOC_{lab}} \cdot dt}{day} + S_{slow,\text{DOC}_{ref},i,t} \cdot \frac{k_{DOC_{ref}} \cdot dt}{day} \tag{81}$$

$$S_{fast,\text{CO2},i,t+1} = S_{fast,\text{CO2},i,t+1} + S_{fast,\text{DOC}_{lab},i,t} \cdot \frac{k_{DOC_{lab}} \cdot dt}{day} + S_{fast,\text{DOC}_{ref},i,t} \cdot \frac{k_{DOC_{ref}} \cdot dt}{day} \tag{82}$$
$$- F_{fast2atm,CO_2,i,t}$$

$$S_{river,\text{CO2},i,t+1} = S_{river,\text{CO2},i,t+1} + S_{river,\text{DOC}_{lab},i,t} \cdot \frac{k_{DOC_{lab}} \cdot dt}{day} + S_{river,\text{DOC}_{ref},i,t} \cdot \frac{k_{DOC_{ref}} \cdot dt}{day} \tag{83}$$
$$- F_{river2atm,CO_2,i,t} + \sum_{v=1}^{13} \left( F_{soil2flood,CO_2,grid\,x,v,t} \right) \cdot \frac{dt}{day}$$
$$\cdot \frac{\%flood_{river,i,t} \cdot A_{total,i}}{\%flood_{total,grid\,x,t} \cdot A_{total,grid\,x}}$$

$$S_{flood,\text{CO2},i,t+1} = S_{floos,\text{CO2},i,t+1} + S_{flood,\text{DOC}_{lab},i,t} \cdot \frac{k_{DOC_{lab}} \cdot dt}{day} + S_{flood,\text{DOC}_{ref},i,t} \cdot \frac{k_{DOC_{ref}} \cdot dt}{day} \qquad (84)$$

$$- F_{flood2atm,CO_2,i,t} + \sum_{v=1}^{13} \left( F_{soil2flood,CO_2,grid\ x,v,t} \right) \cdot \frac{dt}{day}$$

$$\cdot \frac{\%flood_{i,t} \cdot A_{total,i}}{\%flood_{total,grid\ x,t} \cdot A_{total,grid\ x}}$$

At each 6-min time-step, the $CO_2$ produced from the decomposition of DOC is added to the relevant reservoirs (Eqs. 81-84). For $S_{fast}$, $S_{river}$, and $S_{flood}$, the amount of evading $CO_2$ is subtracted from the $CO_2$ stocks (Eqs. 82-84). For $S_{river}$ and $S_{flood}$, the inputs of $CO_2$ from the decomposition of inundated SOC and litter are added to these reservoirs, based on the relative

contribution of swollen rivers ($\%flood_{river}$) and floodplains ($\%flood$) on the total fraction of inundated soils ($\%flood_{total}$) (Eqs. 83-84).

## 2.3 Model calibration and evaluation

The main strategy was to start with the calibration of the hydrology, before calibrating the fluxes of carbon. We started from the forcing data and parametrization used by Guimberteau et al. (2012), and thus already had an initial calibration for that

model. As we changed the flooding scheme and increased the maximum floodable area, we had to recalibrate discharge, in particular the residence time of water in the floodplains $\tau_{flood}$. Due to the increased floodable area, more water is infiltrating into the topsoil on the floodplain and, thus, we had to reduce the water infiltrating into the subsoil ($f_{swamp}$) in order to reproduce the total amount of discharge. The recalibration of discharge focused mainly on reproducing the river discharge at Obidos, the most downstream lying discharge gauge. The idea is that the discharge dynamics at the basin outlet integrates all hydrological

processes in the basin and determines the exports of water and matter to the coast. Nevertheless, the discharges from major sub-basins are evaluated as well.

For the fluxes of C along the terrestrial-aquatic continuum, we build on the default calibration of vegetation processes in ORCHIDEE and on the calibration of soil C processes in ORCHIDEE-SOM (Camino Serrano, 2015), and based on that we tried to reproduce observed DOC exports from the soil to the river network by $F_{RO}$ and $F_{DR}$, before evaluating the model

performance with regard to reproducing observed DOC concentrations in the river (Table 2). The main parameters controlling the DOC concentration in $F_{RO}$ and $F_{DR}$ relative to DOC concentrations in the soil solution are $S_{fast+slow,H2O,ref}$ and $red_{DOC,base}$. As empirical data for calibration and validation are limited, we started with parameter values taken from the literature or based on assumptions. The parameter $red_{DOC,base}$ was set to a value of 0.2 following Braun (2002). The $S_{fast+slow,H2O,ref}$ was set to 160 mm, which is about the $90^{th}$ percentile of $S_{fast,H2O} + S_{slow,H2O}$ within the Amazon basin. The decomposition rates for labile and

refractory DOC within the inland water network, $k_{DOClab} = 0.3$ day$^{-1}$ and $k_{DOCref} = 0.01$ day$^{-1}$, were also taken from the literature (see section 2.2.5). Nevertheless, we made sure that the simulated DOC concentrations in $F_{RO}$ and $F_{DR}$ are comparable to values

reported in the literature, and that deviations between simulated and observed DOC concentrations in the rivers are minimal. In that context, we performed a sensitivity analyses with regard to model performance for changes in $S_{fast+slow,H2O,ref}$, $red_{DOC,base}$, $k_{DOClab}$ and $k_{DOCref}$.

## 3 Model results and discussion

### 3.1 Evaluation of simulated seasonal flooding and river discharge

The upgraded river routing scheme allows us to reproduce seasonal inundation in the Amazon basin (Fig. 8). The improvement using ORCHILEAK instead of the trunk version of ORCHIDEE is in particular visible for the central Amazon basin (Fig. 8a, see Fig. 4 for location). However, compared to the observed inundation reported by Richey et al. (2002), our simulation underestimates the total areal extent of inundation. This is not surprising as our forcing data derived from space borne

microwave remote sensing (Prigent et al., 2007) excludes flooded forests with dense canopies covering free water surfaces and does not capture small water bodies. In contrast, the observed inundation from Richey et al. (2002) was derived from airborne radar imagery, which is able to detect flooded areas in more detail and at higher resolution (Hess et al., 2003). Nevertheless, the simulated spatial pattern inundation throughout the Amazon basin correlates well with the high resolution airborne observations (Hess et al., 2015) (Fig. 9) The observed inundation data for the Roraima and Llanos de Moxos wetlands

(Hamilton et al. 2002; Hamilton et al. 2011) were derived from space borne microwave imagery, and are thus, in terms of spatial resolution and detail, more directly comparable to our forcing data. Therefore, the good match between observed and simulated inundation in these regions highlights the good performance of our new flooding module in ORCHILEAK (Fig. 8b). Nevertheless, it is important to keep in mind that while the overall seasonality of inundation is well reproduced in all regions, the total inundated area across the Amazon basin is likely underestimated because of our choice of forcing data.

After recalibrating the outflow velocity from the floodplains and reducing the amount of water redirected to swamps ($\tau_{flood}$ = 1.4, $f_{swamp}$ =0.1), the simulated discharges are in general quite close to those simulated by Guimberteau et al. (2012) (Fig. 10, Table 3). In the southern tributaries of the Amazon basin (Rio Jurua, Purus, Madeira, Tapajos, Xingu), we overestimate the discharge during high-flow periods (Feb. to April) while for the rest of the year our simulation is well in line with observations. This might be due to a bias in the meteorological forcing data, which could give too much weight to very rainy spots during

the interpolation process, or to an underestimation of simulated evapotranspiration compared to flux tower measurements (Guimberteau et al., 2012). For the northern tributaries (Rio Japura and Rio Negro), such an overestimation during high flows is not visible. Along the main stem (Amazon at Obidos, Rio Solimoes at Sau Paulo de Olivenca), the seasonality is reproduced very well except for Rio Solimoes at Manacapuru where the simulated discharge peak occurs one month too early, due to backwater effects by Rio Negro and Rio Madeira (Meade et al., 1991), process which are not accounted for in ORCHILEAK

nor in the trunk version of ORCHIDEE.

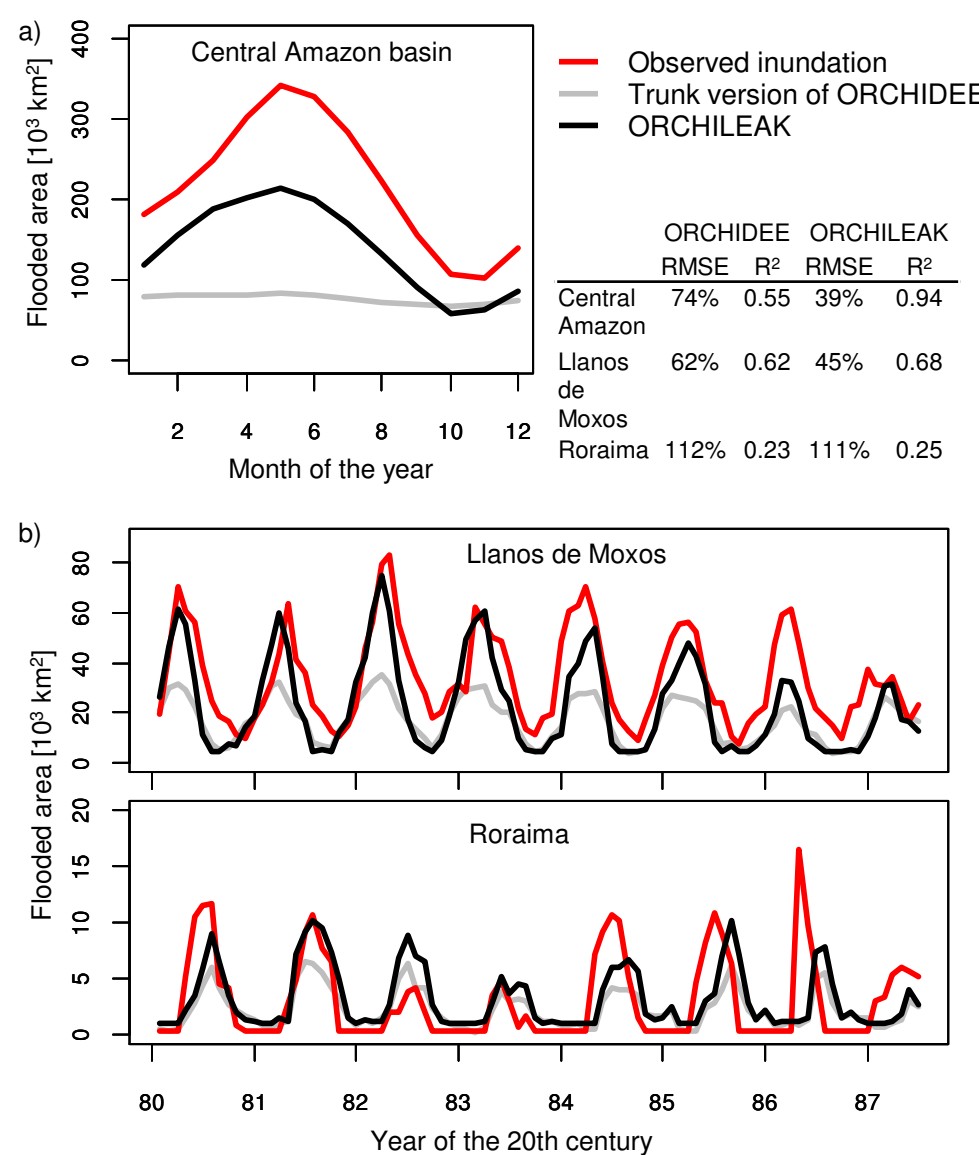

**Figure 8.** Simulated versus observed flooded area in the Amazon basin. a) Central Amazon basin. Observed data from Richey et al. (2002) after Hess et al. (2003). Inundation corresponds to the sum of water surfaces of main channel, tributaries and floodplains recorded during the period October 1995 to September 1996. b) Llanos de Moxos and Roraima floodplains over the period January 1980 to September 1987. Observed data from Hamilton et al. (2011). RMSE is expressed as relative to the mean observed value per area.

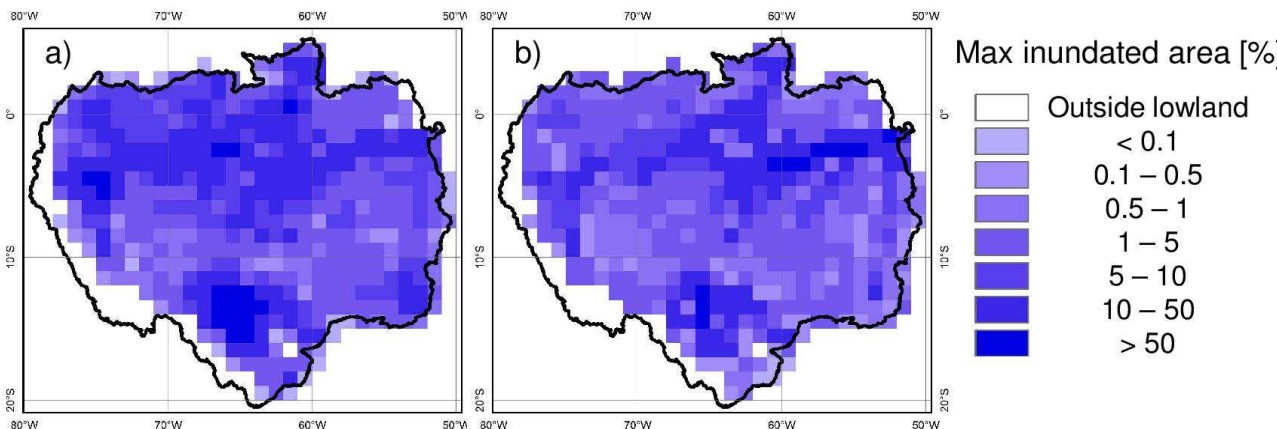

**Figure 9**: Maximum inundated proportion in the lowland (< 500 m altitude) Amazon basin during the years 1995/1996: a) observed (after Hess et al., 2015) vs. b) simulated in ORCHILEAK, y=0.04+0.90*x, $R^2$= 0.56, RMSE=11%.

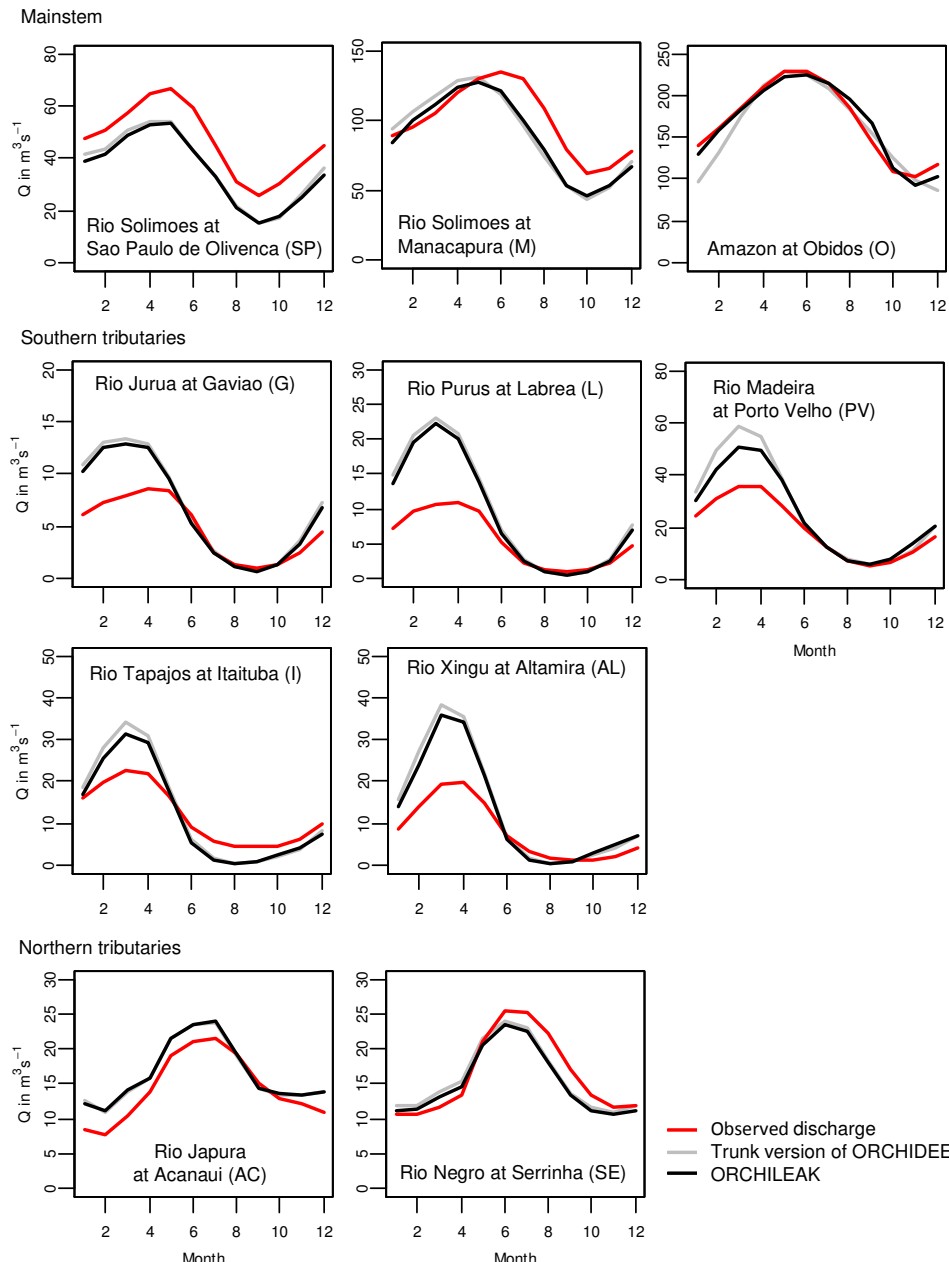

**Figure 10.** Simulated versus observed monthly discharge in the Amazon River and its major tributaries. The simulated discharge represents the average over the simulation period 1980-2000. For the stations at Rio Negro, Rio Purus, Rio Tabajos as well as for the Amazon at Obidos, observed discharges are derived from ORE-HYBAM gauging data for the same period. For the other stations, long-term average monthly

5    discharges from GRDC data set have been used, which cover a longer period: Amazon at Sao Paolo de Olivenca (1973-2010), Rio Madeira at Porto Velho (1967-2007), Rio Japura at Acanaui (1973-1997), Rio Jurua at Gaviao (1972-2010), Rio Xingu at Atamira (1971-2008). See Table 3.

**Table 3:** Performance of discharge simulation in trunk version of ORCHIDEE (parametrization by Guimberteau et al., 2012) vs. ORCHILEAK. In addition, it is shown how ORCHILEAK would perform with the $\tau_{flood} = 2.5$ and $f_{swamp} = 0.2$ used in Guimberteau et al., 2012). As performance measures, RMSE and Nash-Sutcliffe efficiency (NSE) are reported[*]. In most cases, only the performance in reproducing the seasonality is reported, as it's presented in Figure 10. For 4 stream gauges with time-series data, the performance in reproducing these time-series is additionally reported.

| | ORCHIDEE $\tau_{flood}=2.5$ $f_{swamp}=0.2$ | | ORCHILEAK $\tau_{flood}=2.5$ $f_{swamp}=0.2$ | | ORCHILEAK $\tau_{flood}=1.4$ $f_{swamp}=0.1$ | |
|---|---|---|---|---|---|---|
| | RMSE | NSE | RMSE | NSE | RMSE | NSE |
| *Seasonality* | | | | | | |
| **Amazon at O** | **11%** | **0.83** | **20%** | **0.42** | **6%** | **0.95** |
| Rio Solimoes at SP | 23% | 0.32 | 25% | 0.21 | 25% | 0.20 |
| Rio Solimoes at M | 19% | 0.42 | 14% | 0.67 | 17% | 0.55 |
| Rio Jurua at G | 65% | -0.20 | 59% | 0.00 | 59% | 0.00 |
| Rio Purus at L | 113% | -1.68 | 105% | -1.28 | 104% | -1.24 |
| Rio Madeira at PV | 57% | -0.05 | 38% | 0.54 | 40% | 0.48 |
| Rio Tapajos at I | 47% | 0.35 | 38% | 0.58 | 38% | 0.57 |
| Rio Xingu at AL | 108% | -0.64 | 87% | -0.07 | 93% | -0.22 |
| Rio Japura at AC | 17% | 0.71 | 16% | 0.75 | 17% | 0.71 |
| Rio Negro at SE | 13% | 0.86 | 11% | 0.89 | 13% | 0.85 |
| *Time series* | | | | | | |
| **Amazon at O** | **15%** | **0.74** | **23%** | **0.37** | **12%** | **0.84** |
| Rio Purus at L | 122% | -1.95 | 113% | -1.55 | 112% | -1.51 |
| Rio Tapajos at I | 61% | 0.01 | 53% | 0.25 | 54% | 0.24 |
| Rio Negro at SE | 18% | 0.81 | 17% | 0.83 | 18% | 0.80 |

[*] NSE can take values between 1 and - ∞. An NSE = 1 would mean a perfect fit between observed and simulated values. A NSE = 0 means that using the mean observed value as constant simulated value would lead to as much deviation between observed and predicted values as using the actually simulated values. If NSE is negative, there is more deviation between simulated and observed values as between the observed values and their mean.

## 3.2 Evaluation of simulated C fluxes along the terrestrial-aquatic continuum

### 3.2.1 Vegetation, litter and soil carbon

The Amazon basin is largely dominated by tropical rain forest. Other notable plant functional types (PFTs) in the study area are rain green forest, i.e. deciduous tropical forest with litter fall during the dry period, and tropical C3 and C4 grasslands (Table 4). C4 cropland contributes with an areal proportion of 1%, mainly in the form of sugar cane plantations. All other PFTs have an areal proportion smaller than 1%. Over the simulation period (1980-200), the land use forcings give a slight increase in C4 grasslands and croplands at the expense of tropical rain forest (Table 4).

Table 4a summarizes yearly-mean NPP per PFT reported in the literature and simulated with ORCHILEAK, using the default settings for vegetation simulation. Overall, simulated values are in good agreement with those reported in the literature, especially for the dominant PFTs (rain forests). Values for C3 grassland are compared to a study in the Andes, as most C3 grassland in the Amazon basin is found in high altitudes at the western rim of the study area. For C4 grassland, a rather wide range of NPP has been reported, with highest values for grass dominated wetland systems which are important for the C biogeochemistry in the Amazon floodplains (Melack et al., 2009). In that specific area, the average annual NPP for this PFT is simulated at around 2900 g C m$^2$ yr$^{-1}$, i.e. still at the lower end of the reported value range for C4 wetland grasses. In the southernmost part of our study areas, the average simulated NPP for simulated C4 grassland goes below 1500 g C m$^{-2}$yr$^{-1}$. Fig. 11 shows the spatial heterogeneity in simulated average NPP 1980 to 2000. The spatial pattern reflects the relatively low NPP of rain forest compared to tropical grasses. Within the Amazon basin, the tropical grasses in the lower Amazon floodplains and in the Llanos de Mojos show the highest average NPP. The simulated soil carbon stocks in the Amazon basin are in good agreement with the Harmonized Worlds Soil database (Table 4b).

**Table 4a.** Yearly-mean simulated NPP in the Amazon basin (period 1980-2000) reported for the five dominant Plant Functional Types (PFT).

| PFT | Areal proportion | | NPP [gC m$^{-2}$ yr$^{-1}$] | | |
|---|---|---|---|---|---|
| | 1980 | 2000 | simulated | literature | |
| Tropical rain forest | 83.1% | 81.6% | 1,086 | 1,250 | Saugier et al. (2001) |
| Rain-green forest | 3.1% | 2.9% | 1,001 | 1,200 | Martinez-Yrizar et al. (1996) |
| C3 grass-land | 4.1% | 4.0% | 835 | 460 - 1530 (Andean grass lands) | Oliveras et al. (2014) |
| C4 grass-land | 6.9% | 8.0% | 2,202 | 100-500 (low rainfall) 500-2,000 (high rainfall) 2,500-7,000 (wetland) | Long et al. (1991) |
| C4 crop-land | 1.0% | 1.6% | 2,566 | 3,000-5,500 (sugar cane plantation) | Long et al. (1991) |

**Table 4b.** Simulated and observed mean soil organic carbon (SOC) stocks in the Amazon basin. Values are reported for the top 30 cm, the top 100cm and the whole 200cm profile used in the simulation.

| Depth | Soil carbon stocks [kg C m$^{-2}$] | | |
|---|---|---|---|
| | This study | HWSD[*] | Literature[**] |
| 30 cm | 5.2±1.7 | 6.4±5.6 | 4.0-4.8 |
| 100 cm | 7.4±2.3 | 11.2±9.5 | 7.9-9.0 |
| 200 cm | 8.3±2.6 | | |

[*] Derived from the Harmonized World Soil Database (FAO/IIASA/ISRIC/ISS-CAS/JRC, 2009).

10 [**] After literature review in Ceddia et al. (2015)

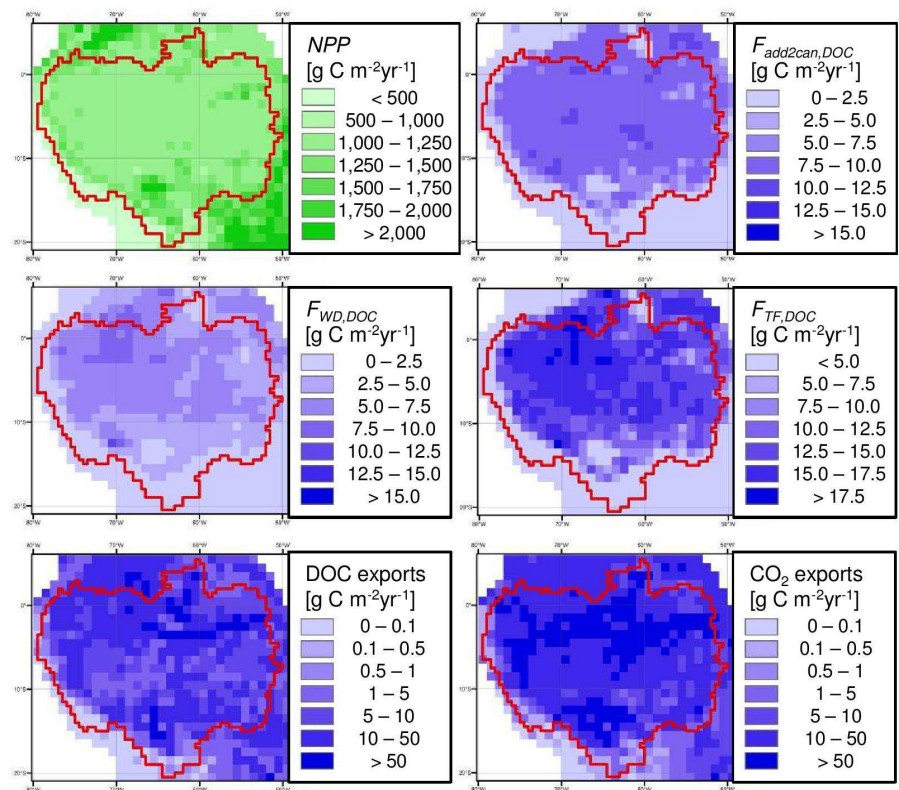

**Figure 11.** Averages of simulated net primary production (*NPP*), dry deposition of soluble organic C onto the canopy and leaching of DOC from leaves ($F_{add2can,DOC}$), wet deposition of DOC (DOC in rain, $F_{WD,DOC}$), throughfall DOC flux ($F_{TF,DOC}$), as well as total DOC and CO$_2$ exports into the inland water network ($F_{RO}+F_{DR}+F_{soil2flood}+F_{soil2river}$) over the simulation period 1980-2000.

**3.2.2 DOC in precipitation and throughfall**

Figure 11 shows the spatial patterns in simulated averages of DOC production in canopy (sum of dry deposition of soluble organic C and leaching of DOC from leaves, $F_{add2can,DOC}$), wet deposition of DOC (DOC in rain, $F_{WD,DOC}$), and throughfall DOC flux ($F_{TF,DOC}$). In most parts, $F_{add2can,DOC}$ contributes more to $F_{TF,DOC}$ than $F_{WD,DOC}$. The patterns in $F_{add2can,DOC}$ is mainly controlled by the distribution of tropical rain forest and rain-green tropical forests, because, due to limitations in calibration

data, we do not simulate this flux for grass lands nor crop lands. $F_{WD,DOC}$ follows the patterns of precipitation, as we use fixed DOC concentrations for this flux. Simulated average values for $F_{TF,DOC}$ range from 0 g C m$^{-2}$yr$^{-1}$ to about 20 g C m$^{-2}$yr$^{-1}$, with the highest fluxes to be found where dense rain forests coincide with highest average precipitation, like in the NW Amazon basin.

Our simple representation of throughfall DOC fluxes is able to reproduce the yearly-mean and seasonal variations in

throughfall DOC concentrations observed by Johnson et al. (2006) in Southern Amazonia (Fig. 12). Although the throughfall DOC was calibrated only for this study area, it reproduces the observed yearly mean fluxes in NW and Central Amazonia

(Filoso et al., 1999; Tobón et al., 2004) in a satisfying way as well (Fig. 12a, Table 5). Interestingly, the annual throughfall DOC fluxes do not differ much among these very different regions of the Amazon. In particular, the average annual precipitation differs substantially from 3400 mm yr$^{-1}$ in the NW part of the basin (locations 1, 2, 3 correspond to points TF1 TF2, and TF3 in Fig. 4) to only about 2000-2200 mm yr$^{-1}$ at the other two locations in the central and southern part of Amazonia (see Table 5). Similar throughfall flux has also been reported for tropical rainforest in Indonesia (12.6 to 16.4 g C m$^{-2}$yr$^{-1}$, (Fujii et al., 2011)) as well as for primary, sub-tropical rain forests in Puerto Rico (13.2 g C m$^{-2}$yr$^{-1}$, (Heartsill-Scalley et al., 2007)) and Taiwan (18.9 g C m$^{-2}$yr$^{-1}$ (Liu and Sheu, 2003)).

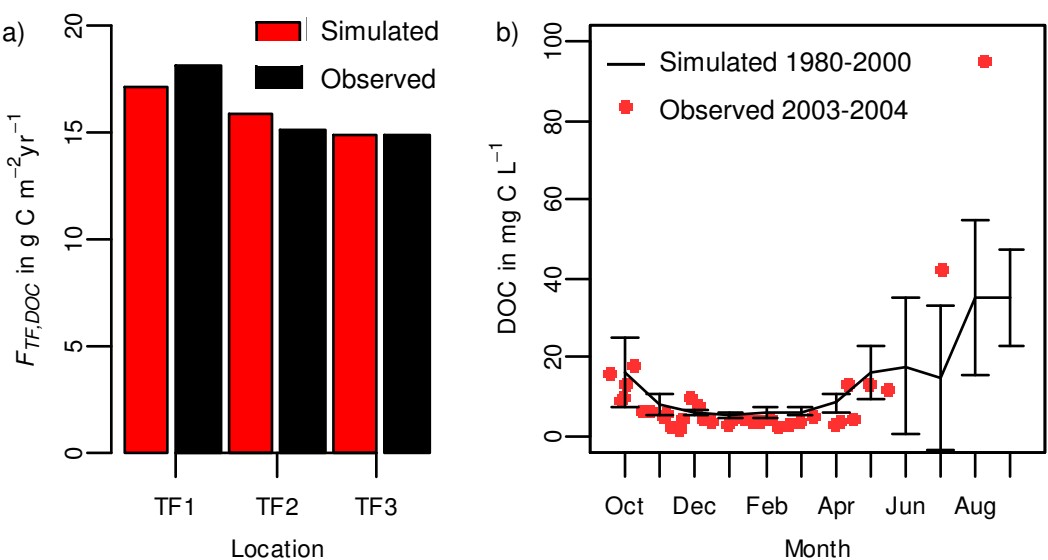

**Figure 12.** Simulated versus observed DOC in throughfall ($F_{TF}$). a) Yearly-mean throughfall DOC flux vs. literature values for the following three locations: 1) NW Amazonia (TF1 in Fig. 4) - Tobon et al., (2004); 2) Lower Rio Negro (TF2 in Fig. 4) – Filoso et al., (1999); 3) S Amazonia (TF3 in Fig. 4) – Johnson et al., (2006). b) Seasonality in throughfall DOC concentrations for the site in S Amazonia (TF3, Johnson et al., 2006). Note the sharp concentration increase during dry season from May to September. As the sampling period is outside of our simulation period, we compare the observed concentration with simulated average DOC concentrations over the entire run (1980-2000).

**Table 5.** Simulated versus observed DOC concentrations (conc.), water and DOC fluxes in precipitation (rain) and throughfall (TF).

| | | | Literature | | | | | | Ref | Simulated | | | |
| | | | DOC conc. | | Water-Flux | | DOC-Flux | | | DOC conc. | Water-Flux | | DOC-Flux |
| | | | [mg C L$^{-1}$] | | [mm yr$^{-1}$] | | [g C m$^{-2}$yr$^{-1}$] | | | [mg C L$^{-1}$] | [mm yr$^{-1}$] | | [g C m$^{-2}$yr$^{-1}$] |
| | Lon | Lat | Rain | TF | Rain | TF | Rain | TF | | TF | Rain | TF | TF |
| Pena Roja, sedimentary plain | 72°06'W | 0°45'S | 3.9 | 5.5 | 3400 | 2689* | | 14.8 | 1 | 5.8 | 3374 | 2951 | 18.2 |
| high terrace | " | " | " | 6.7 | " | 2841* | | 19.0 | 1 | " | " | " | " |
| low terrace | " | " | " | 6.7 | " | 2603* | | 17.4 | 1 | " | " | " | " |
| floodplain | " | " | " | 6.3 | " | 2783* | | 17.6 | 1 | " | " | " | " |
| Lower Rio Negro | 60°47'W | 2°41'S | 1.3 | 9.7 | 2083 | 1635 | | 15.9 | 2 | 8.4 | 2172 | 1671 | 15.1 |
| rainy season | " | " | 1.4 | 8.6 | | | | | 2 | 6.9 | | | |
| dry season | " | " | 1.3 | 11.3 | | | | | 2 | 10.4 | | | |
| Southern Amazonia | 58°28'W | 10°28'S | | 6.2 | 2200 | | | 14.9 | 3 | 8.8 | 2055 | 1690 | 14.9 |
| Lake Calado, Central Amazon | 60°34' W | 3°15' S | 1.9 | | | | 4.8 | | 4 | 2.4 | 2319 | 2319 | 5.6 |

* Calculated from the (flux-weighted) average concentration and throughfall DOC flux

1) Tobón et al., 2004      based on samples taken from January 1995 and August 1997, simulation results for 1995-1997
2) Filosos et al., 1999      based on samples March to December 1991, rainy season is from December to May,
here, rainy season: March to May + December 1991, dry season: June to November 1991

3) Johnson et al, 2006      based on samples taken during 2003/2004, here simulation results for 1980-2000 (no newer forcing file available)
4) Williams et al., 1997      July 1989 to June 1990, here throughfall is assumed to equal rain, as there is no vegetation on the lake

### 3.2.3 Exports of DOC from soils to headwaters and floodplains

Comparing our simulation results to observed export fluxes through the terrestrial-aquatic interface is rather difficult, because studies with robust data are rare and the coarse resolution of our simulation in combination with the global soil forcing data may not reproduce the soil-hydrology at the plot scale. Nevertheless, we attempted such comparison for three headwater catchments located far apart in the Amazon basin (Table 6, RO1-3, Fig. 4). All three case studies have more recent sampling times than our simulation period, and we thus compared observations with simulation results averaged over the 1980-2000 period. The first basin used for comparison is a small black-, head-water catchment in the lower Rio Negro basin (RO1 in Table 6) (Waterloo et al., 2006). While our forcing data agree with the reported annual precipitation in the region, ORCHILEAK underestimates the contribution of surface runoff to total runoff by a factor of two. Nevertheless, the simulated DOC concentrations in $F_{DR}$ and $F_{RO}$ agree well with the observed values (Table 6). We can also compare to reported concentrations in headwater catchments, which are not represented by $S_{river}$ due to the coarse resolution of the routing scheme, but which can roughly be estimated from the concentration associated to the summed flux of $F_{fast\ out}$ and $F_{slow\ out}$. Here, we underestimate the DOC concentration in the headwaters by a factor of about two, which is consistent with the underestimation of $F_{RO}$ contributions with high DOC concentrations. In the second case for comparison, a small headwater in the Peruvian mountains (RO2, Table 6, Fig. 4), our simulated headwater DOC concentrations are close to observed values. The third case study RO3 (Johnson et al., 2006, 2008) is for two neighbouring headwater catchments in S-Amazon, and was also used for calibrating the throughfall DOC component. At this location, we have again good agreement for the annual precipitation, but, for the grid cell corresponding to the sampling location, we overestimate the contribution of $F_{RO}$ to total runoff, due to the contribution of swamps. Thus, we also compare observations to the simulation results for a neighbouring grid cell without swamps (Simulation b, Table 6). Here, the simulated contribution of $F_{RO}$ is closer to the observations. The simulated DOC concentration in $F_{RO}$ is about the same for both cells and lies between the values observed for the two headwater catchments. The simulated headwater DOC concentration agrees well with the observed values for the second cell, for which the simulated $F_{RO}$ contribution is more in agreement with the observation. For the first grid cell, for which the contribution of $F_{RO}$ is overestimated, the headwater DOC concentration is overestimated accordingly. From the sensitivity analysis in Table 7, we see that the simulated DOC concentrations in overland flow, or surface runoff, are sensitive to the parameters $S_{fast+slow,H2O,ref}$ and $red_{DOC,base}$ which determine the DOC concentration in $F_{RO}$ relative to the DOC concentration in the top 4.5 cm of the soil column (eqs. 47, 53-55). The parameter $red_{DOC,base}$ has in addition an influence on DOC concentrations in drainage, as it controls the advection of DOC relative to water fluxes within the soil column (eqs. 51-54). The decomposition rate of labile DOC ($k_{DOC,lab}$) exerts a moderate control on the simulated DOC concentrations in the headwaters (here the combined outflows from $S_{fast}$ and $S_{slow}$), while the decomposition rate of refractory DOC ($k_{DOC,ref}$) does not have a significant effect due to the rather short residence time in $S_{fast}$ and $S_{slow}$.

**Table 6.** Observed and simulated DOC concentrations in overland flow (=$F_{RO}$) and headwater streams (=$F_{fast\ out}$+$F_{slow\ out}$).

| Source | Rain mm yr$^{-1}$ | Runoff mm yr$^{-1}$ | Surface runoff mm yr$^{-1}$ | Component | DOC conc. mg C L$^{-1}$ |
|---|---|---|---|---|---|
| *RO1 - Igarapé Asu rainforest catchment, 6.8 km², Lower Rio Negro Basin (60°12'W,2°36'S)* | | | | | |
| Waterloo et al. | 2442 | | 1071 | Overland flow, avg. | 16.6 |
| | | | | ~, range | 8-27 |
| | | | | Drainage | <5 |
| | | | | Stream | 9.5 - 15.4 |
| Simulation | 2412 | 1480 | 526 | Overland flow, avg. | 20.4 |
| (60.5°W,2.5°S) | | | | ~, range | 6.1-37.0 |
| | | | | ~, 5th-95th percentile | 9.8-29.6 |
| | | | | Drainage | 4.0 |
| | | | | Stream | 5.9 |
| *RO2 - Upland Peruvian headwater catchment* | | | | | |
| Saunders et al. (2006) | 1800 | | | Stream (May to Sept) | 3.1 |
| Simulation | 1434 | | | Stream (May to Sept) | 2.6 |
| (75.5°W,10.5°S) | | | | | |
| *RO3 - Southern Amazonia (58°28'W,10°28'S)* | | | | | |
| Johnson et al. (2006a,2006b,2008) | 2200 | | 2.5%[*] | Overland flow | 10.7 |
| | " | | 3.2%[*] | | 25.2 |
| | " | | | Stream, 1st order | 2.3 |
| | " | | | Stream, 2nd order | 3.7 |
| Simulation a | 2055 | 862 | 68%[*] | Overland flow | 16.1 |
| (58.5°W,10.5°S) | " | " | " | Stream | 6.1 |
| Simulation b | 2090 | 959 | 4.9%[*] | Overland flow | 16.0 |
| (59.5°W,10.5°S) | " | " | " | Stream | 2.8 |

[*] The surface runoff is reported as percentage of total runoff, in the literature, and, for comparison, also for simulated values.

**Table 7**: Sensitivity of simulated average DOC concentrations (mg C L$^{-1}$) in surface runoff, drainage and headwater streams to changes in key parameters in calibration.

| | Final set-up | $S_{fast+slow,H2O,ref}$ | | $red_{DOC,base}$ | | $k_{doc,lab}$ | | $k_{doc,ref}$ | |
|---|---|---|---|---|---|---|---|---|---|
| | | -50% | 50% | -50% | 50% | -50% | 50% | -50% | 50% |
| *RO1* | | | | | | | | | |
| Overland flow, avg. | 20.4 | 27.9 | 17.0 | 18.0 | 22.6 | 20.4 | 20.4 | 20.4 | 20.4 |
| Drainage, avg. | 4.0 | 3.9 | 4.0 | 3.2 | 4.8 | 4.0 | 4.0 | 4.0 | 4.0 |
| Stream, avg. | 5.9 | 7.2 | 5.3 | 5.3 | 6.5 | 6.4 | 5.5 | 5.8 | 5.8 |
| *RO2 (May to September)* | | | | | | | | | |
| Stream, avg. | 2.6 | 3.5 | 2.2 | 1.7 | 3.3 | 2.6 | 2.6 | 2.6 | 2.6 |
| *RO3, simulation b* | | | | | | | | | |
| Overland flow, avg. | 16.0 | 22.6 | 13.1 | 10.4 | 20.6 | 16.0 | 16.0 | 16.0 | 16.0 |
| Stream, avg. | 2.8 | 3.1 | 2.7 | 1.8 | 3.9 | 3.0 | 2.6 | 2.8 | 2.8 |

With an arithmetic mean of about 21 g C m$^{-2}$ yr$^{-1}$, the simulated total DOC inputs to the inland water network of the Amazon are significant (Table 8, Fig. 11), and about 5 times larger than the lateral DOC export from the Amazon basin at Obidos (4.6 g C m$^{-2}$yr$^{-1}$, Moreira-Turcq et al. 2003). More than half of the inputs are delivered by surface runoff ($F_{RO}$) (Table 8). More specifically, the total DOC input associated to $F_{RO}$ is more than 3 times higher than that originating from drainage ($F_{DR}$) although the simulated $F_{RO}$ contributes only to 44% of the total runoff. This result can be explained by the much higher basin-scale average DOC concentration in $F_{RO}$ than in $F_{DR}$ (see Table 8). The simulated DOC inputs from $F_{RO}$ can reach very high values (Table 8) in the presence of swamps, where a constant fraction of river water is redirected to the soil column, leading to a very high runoff from the topsoil that can be several times higher than the precipitation flux. Note that a substantial part of this DOC export from swamps is fed by the DOC from the infiltrating river water. Thus the very high basin-scale DOC input associated to $F_{RO}$ of 362 g C m$^{-2}$yr$^{-1}$ (Table 8) is reduced to 71 g C m$^{-2}$yr$^{-1}$ when swamp areas are excluded from the analysis. The simulated return flow of river water into the soil column in swamps ($F_{up2swamp}$) averages 2.1 g C m$^{-2}$yr$^{-1}$ throughout the Amazon basin. The simulated infiltration of DOC on floodplains reaches a similar value of 2.4 g C m$^{-2}$yr$^{-1}$ ($F_{flood2soil}$). Subtracting these fluxes from the inputs, we obtain an average net-input from the soil-vegetation system into the inland water network of about 16.5 g C m$^{-2}$yr$^{-1}$. Although the maximum floodable area in the Amazon basin does not exceed 6.4% according to our forcing files (Fig. 6), the simulated DOC input from submerged litter amounts to one third of total DOC inputs to the inland waters.

As explained in the method section, a "poor soils" forcing was implemented to represent coarsely textured, acidic and nutrient depleted soils in which DOC decomposition is reduced and vertical advection is more effective. For nine grid cells in the Amazon basin where the areal proportion of "poor soils" is higher than 75% (Fig. 6), the simulated DOC export is dominated by such soils. Here, the DOC export flux associated to $F_{DR}$ averages at 22.7 g C m$^{-2}$yr$^{-1}$, i.e. nearly nine times the basin average

value. The average DOC concentration in drainage (21.6 mg C L$^{-1}$) is more than six times the basin average. For the two grid cells having 100% "poor soils", the average DOC concentration reaches 24.7 mg C L$^{-1}$, which is however still substantially lower than the value of 36 g C L$^{-1}$ reported for groundwater seeping through the Podzols of the Rio Negro basin (McClain et al., 1997).

**Table 8.** Statistical distributions of simulated export fluxes and concentrations within the Amazon basin.

| | Mean | Min | 1$^{th}$ perc. | 5$^{th}$ perc. | Median | 95$^{th}$ perc. | 99$^{th}$ perc. | Max |
|---|---|---|---|---|---|---|---|---|
| DOC fluxes [g C m$^{-2}$yr$^{-1}$] | | | | | | | | |
| Surface runoff | 10.6 | 0.0 | 0.0 | 0.1 | 3.7 | 39.6 | 169.9 | 361.6 |
| Drainage | 3.7 | 0.0 | 0.0 | 0.0 | 2.1 | 11.8 | 30.8 | 45.6 |
| Floodplain | 6.6 | 0.0 | 0.0 | 0.0 | 0.4 | 33.3 | 73.7 | 115.3 |
| Labile proportion [%] | | | | | | | | |
| Surface runoff | 35.9 | 7.7 | 15.9 | 23.0 | 33.4 | 61.4 | 66.2 | 68.7 |
| Drainage | 3.8 | 0.8 | 1.1 | 1.9 | 3.8 | 4.4 | 4.9 | 10.1 |
| Floodplain | 61.2 | 59.4 | 60.1 | 60.6 | 61.1 | 62.2 | 65.3 | 66.4 |
| DOC concentration [mg L$^{-1}$] | | | | | | | | |
| Surface runoff | 13.5 | 0.2 | 0.6 | 4.3 | 13.3 | 24.3 | 36.9 | 43.2 |
| Drainage | 3.2 | 0.0 | 0.1 | 1.5 | 2.6 | 6.1 | 17.9 | 26.4 |
| CO$_2$ fluxes [g C m$^{-2}$yr$^{-1}$] | | | | | | | | |
| Surface runoff | 1.6 | 0.0 | 0.0 | 0.0 | 0.6 | 3.4 | 25.8 | 72.5 |
| Drainage | 20.0 | 0.0 | 0.0 | 0.8 | 16.0 | 51.8 | 63.4 | 125.3 |
| Floodplain | 37.9 | 0.0 | 0.0 | 0.0 | 2.6 | 174.1 | 419.3 | 491.0 |

10 **3.2.4 Transport and decomposition of DOC in the river network**

To evaluate the simulated DOC concentrations and fluxes, we used data from the CAMREX (Carbon in the AMazon River EXperiment) program (Richey et al., 2008), during which 13 cruises were performed over the period 1982-1991, the Ore-Hybam sampling network (Cochonneau et al., 2006), which was designed to capture the land-ocean matter transfer through the Amazon river network from the Andes down to Obidos with regular sampling campaigns, and the data from the study of

15 Moreira-Turcq (2003). Comparing observed vs. simulated DOC concentrations, we were able to reproduce the average concentrations at least in the main stem of Rio Solimoes/Amazon River and in the Rio Negro (Fig. 13). However, apart for the Rio Negro, we generally underestimate the seasonal variability of DOC concentrations. For Obidos, the most downstream sampling location for which we have data, the mean simulated DOC concentration deviates by only -2% from the observed ones (Table 9). For the whole set of observed data, the deviation of simulated from observed average concentrations is -1%

(Table 9, 'Final set-up'). For Rio Jurua, concentrations are generally underestimated, while they are overestimated and Rio Japuru. These discrepancies could likely result from the coarseness of the river routing scheme, soil and wetland forcing files, thereby limiting our ability to reproduce the contributions of a specific flow path ($F_{RO}$ high in DOC vs. $F_{DR}$ low in DOC) to stream flow and additional inputs from riparian wetlands. The simulated DOC concentrations are sensitive to the parameters controlling DOC export with surface runoff from the top-soil, $F_{fast+slow,H2O}$ and $red_{DOC,base}$, and the decomposition rate of labile DOC, $k_{DOC,lab}$, but not to the decomposition rate of refractory DOC, $k_{DOC,ref}$, which is very low and doesn't contribute much to in-stream respiration (Table 9).

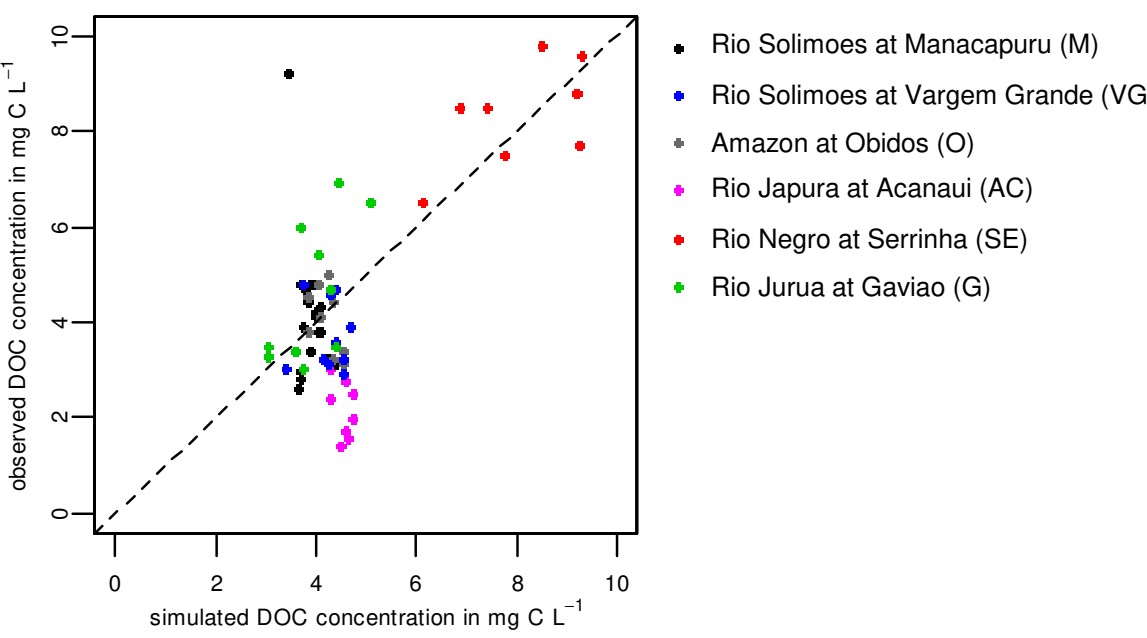

**Figure 13:** Observed versus simulated DOC concentrations ($R^2$=0.45, RMSE = 1.45 mg C L$^{-1}$). For simulated values, each point represents the average during the year and month for which field data are available. The dashed line represents the 1:1 line.

**Table 9**: Sensitivity analysis on the performance of simulating DOC concentrations. As performance measures, root mean squared errors (RMSE) and mean signed deviation (MSD), both relative to the mean observed concentration, are reported per sampling location, and for the whole set of observed DOC concentrations.

| | Amazon at O | | R. Solimoes at M | | R. Solimoes at SP | | R. Japura at AC | | R. Negro at SE | | R. Jurua at G | | All together | |
|---|---|---|---|---|---|---|---|---|---|---|---|---|---|---|
| | RMSE | MSD | RMSE | MSD | RMSE | MSD | RMSE | MSD | RMSE | MSD | RMSE | MSD | RMSE | MSD |
| Final set-up | 18% | -2% | 37% | -9% | 24% | 10% | 89% | 83% | 14% | -7% | 29% | -16% | 33% | -1% |
| $F_{fast+slow,H2O}$ | | | | | | | | | | | | | | |
| -50% | 22% | 8% | 37% | 2% | 32% | 23% | 107% | 101% | 12% | 0% | 25% | -1% | 34% | 10% |
| +50% | 19% | -7% | 39% | -14% | 22% | 4% | 80% | 73% | 15% | -10% | 34% | -24% | 33% | -6% |
| $red_{DOC,base}$ | | | | | | | | | | | | | | |
| -50% | 24% | -16% | 45% | -25% | 24% | -11% | 55% | 45% | 17% | -13% | 37% | -27% | 36% | -16% |
| +50% | 21% | 10% | 37% | 8% | 37% | 30% | 125% | 121% | 12% | -1% | 23% | -5% | 37% | 14% |
| $k_{doc,lab}$ | | | | | | | | | | | | | | |
| -50% | 27% | 21% | 37% | 6% | 32% | 23% | 107% | 102% | 13% | 5% | 25% | -8% | 35% | 13% |
| +50% | 27% | -19% | 42% | -20% | 22% | -1% | 73% | 66% | 20% | -17% | 34% | -23% | 36% | -12% |
| $k_{doc,ref}$ | | | | | | | | | | | | | | |
| -50% | 18% | 3% | 37% | -4% | 27% | 15% | 96% | 90% | 12% | -1% | 27% | -13% | 33% | 4% |
| +50% | 19% | -3% | 38% | -10% | 23% | 9% | 86% | 80% | 14% | -8% | 30% | -17% | 33% | -2% |

The simulated DOC fluxes (Figs. 14, 15) follow mainly the dynamics in simulated discharge (Fig. 10), while the simulated DOC concentrations are less variable. In Fig. 14, we compare our simulations to data from the CAMREX project. We restrict our validation to the period 1982 to 1986, during which sampling frequency was highest (9 of the 13 cruises in that first half of the total period). In Figure 15, we collate various data sources (CAMREX, the Ore-Hybam sampling network (Cochonneau

et al., 2006), and the data from Moreira-Turcq (2003) to validate the simulated seasonality in DOC fluxes at the sampling location Manacapuru (Rio Solimoes) and Porto Velho (Rio Madeira). Overall, just as for discharge, the simulation reproduces the observed mean and seasonal variability in DOC fluxes quite well (Figs. 14 and 15). We find very good agreement for the Rio Solimoes at Sao Paulo de Olivenca, which drains the Andes in the Western part of the Amazon Basin, the Rio Negro as the major black water tributary, and Rio Jurua (Fig. 14, see Fig. 4 for locations). For Rio Solimoes at Manacapuru, the simulated

peak in DOC fluxes occurs one month too early (Fig. 15), consistent with the simulation of discharge (Fig. 10). This slight time lag can be attributed to backwater effects from the two main tributaries, Rio Negro and Rio Madeira, which are not accounted for in our simulation (see section 3.1). For Rio Japura, we overestimate the DOC fluxes although the simulated discharge agrees quite well with observations (Fig. 10), because we generally overestimate the DOC concentrations (Fig. 13, Table 9). For the Rio Madeira (Fig. 15), we have only observed DOC fluxes for years (2003-2006) beyond our simulation

period (1980-2000). Comparing the mean monthly fluxes for the respective periods, we observe that simulated fluxes are generally overestimated, particularly during high flow periods, a result which is consistent with the overestimation of river flow (section 3.1).

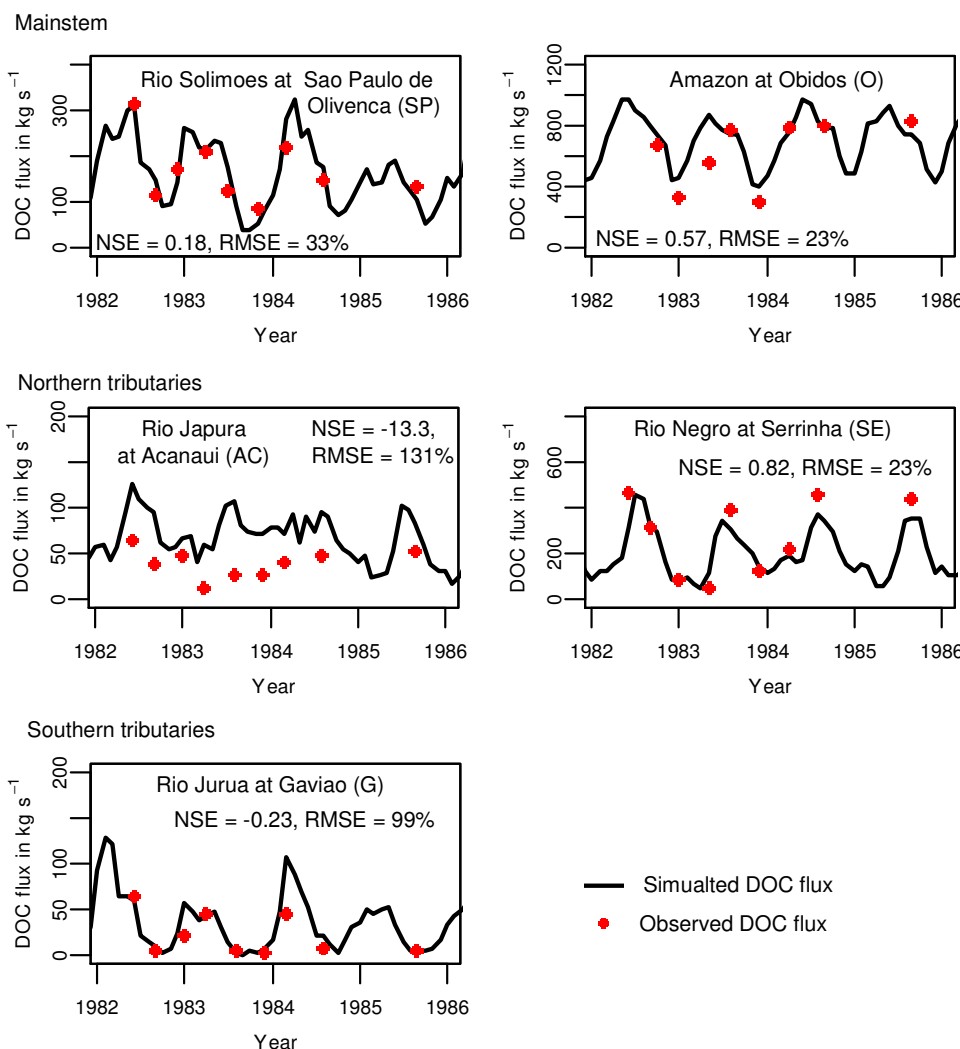

**Figure 14.** Simulated versus observed DOC fluxes in the Amazon main stem and its major tributaries. Observed data are taken from the CAMREX data set (Richey et al., 2008).

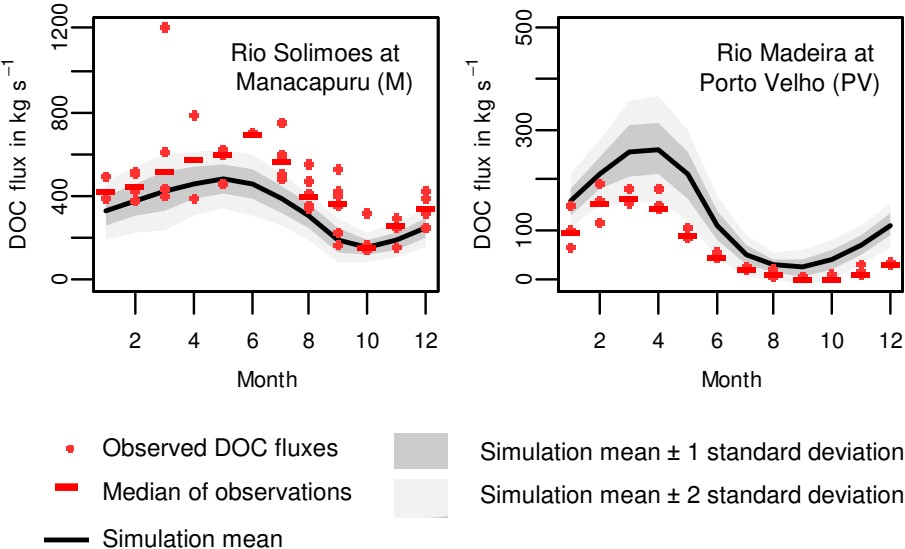

**Figure 15.** Seasonality in DOC fluxes in rivers at two sampling locations with more than 10 samples: Rio Solimoes at Manacapuru (RMSE = 29.4%, NSE = 0.17) and Rio Madeira at Porto Velho (RMSE = 89%, NSE = -0.06). Simulated data report the mean of simulated values per month over the simulation period 1980-2000, including standard deviations of monthly means over the same period. Observed data are from Moreira-Turcq et al. (2003), Cochonneau et al. (2006) and Richey et al. (2008). For the observed data, we report median values (instead of the mean, which is more sensitive to single outliers).

Combining the fluxes at Obidos with that of Rio Tapajos, which is entering the Amazon just below Obidos, the integrated, yearly DOC export fluxes during our simulation period is in the range 19-27 Tg DOC yr$^{-1}$, with a mean value of 23.4 Tg C yr$^{-1}$. Our estimate is very close to that of 22.4 Tg DOC yr$^{-1}$ (710 kg C s$^{-1}$) calculated by Richey et al. (1990) and slightly lower than the 27 Tg DOC yr$^{-1}$ (856 kg C s$^{-1}$) estimated by Moreira-Turcq et al. (2003). This mean simulated annual DOC export flux corresponds to a flux of about 4 g C m$^{-2}$yr$^{-1}$ if normalized to the whole catchment area, a value which is 80% lower than the simulated net input flux of DOC from precipitation, vegetation and the soil system (see section 3.3.1). The Amazon basin can be sub-divided into three major sub basins: 1) The Rio Solimoes, i.e. the Amazon mainstem down to Manacapuru: 2) The Rio Negro, and 3) The Rio Madeira. Our simulation results show that Rio Solimoes contributes about half (10.7±1.4 Tg DOC yr$^{-1}$) of the total DOC export flux at Obidos, while the remainder is largely contributed by the Rio Negro (7.0±1.2 Tg DOC yr$^{-1}$) and Rio Madeira (5.7±0.7 Tg DOC yr$^{-1}$).

### 3.2.5 Transport and evasion of CO$_2$

The simulated total inputs of CO$_2$ to the inland waters is significantly higher than that of DOC (Table 8). However, for inputs via $F_{RO}$ only, the CO$_2$ load is one order of magnitude lower than that of DOC. This is compensated by the inputs via $F_{DR}$, where

the simulated $CO_2$ exports are more than 5 times higher than that of DOC. Overall, $F_{DR}$ is responsible for about 90% of the $CO_2$ exports from non-flooded soils to inland waters, in agreement with the relative $CO_2$ concentrations set for the two export pathways (see section 2). Similarly, the $CO_2$ inputs from root respiration and heterotrophic respiration in the flooded soils gives an average flux of 39.5 g C m$^{-2}$yr$^{-1}$, nearly twice as large as the input from non-flooded soils. Abril et al. (2014) estimate

the C inputs ($CO_2$ + DOC) to the water column per floodable area to 1100 ± 455 g C m$^{-2}$ yr$^{-1}$ for the Central Amazon basin. Relating our simulated $F_{soil2flood}$ to $\%flood_{max}$, we obtain a similar average flux rate 1036 g C m$^{-2}$yr$^{-1}$ within the Central Amazon basin.

The spatial pattern in our simulated $CO_2$ evasion (Fig. 16) correlates, naturally, strongly with $\%flood_{max}$ (Fig. 6), because floodplains represent the largest contribution to the total inland water surface area. Thus, highest average fluxes are found in

the central Amazon floodplain and the Llanos de Moxos. As we use constant gas exchange velocities and do not account for in-river autotrophic production by algae, our simulated $CO_2$ evasion cannot reproduce short-term variation in evasion fluxes. However, our average $CO_2$ evasion rate per water surface area are in good agreement with average observed evasion rates from several large rivers of the Amazon basin (Fig. 17). In addition, the simulated $CO_2$ evasion can be compared to the values reported by Richey et al. (2002). For the central Amazon basin (cf. Fig. 4), our simulation results give an average $CO_2$ evasion

of 229 Tg C yr$^{-1}$, which is close to Richey et al.'s (2002) estimate of 210 ± 60 Tg C yr$^{-1}$. In addition, the simulation reproduces well the observed seasonal variations in $CO_2$ fluxes (Fig. 18). According to our results, floodplains contribute half (51%) of the yearly-mean $CO_2$ evasion, rivers contribute another 39% while the remainder (10%) evades from the fast reservoir. The latter can be regarded as small headwaters without inputs of $CO_2$ rich groundwater, which, in our model, do not exchange $CO_2$ with the atmosphere until they enter the river reservoir.

The fact that we simulate a total $CO_2$ evasion similar to the one reported by Richey et al. (2002) is somewhat surprising taken that our mean water surface area is substantially lower (see section 3.1). In other words, we simulate a higher $CO_2$ evasion rate per water surface area than estimated by Richey et al. (2002). These authors used relatively low gas exchange velocities $k_{600}$ of 1.2 to 2.3 m day$^{-1}$ to calculate $CO_2$ evasion from rivers, while we applied a significantly higher value of 3.5 m day$^{-1}$, following more recent observations (Alin et al., 2011; Rasera et al., 2013). Note that in our physically based model approach,

the total $CO_2$ evasion is not very sensitive to the gas exchange velocity, but rather to the simulated $CO_2$ sources. Reducing or increasing the gas exchange velocities $k_{river,600}$ and $k_{swamp,600}$ by 50% leads to a change in simulated total $CO_2$ evasion of only -4% and 1%, respectively. On the contrary, in a data driven approach to calculate $CO_2$ from observed river p$CO_2$ values, the calculated $CO_2$ evasion will change linearly with changes in the gas exchange velocity. Rasera et al. (2013) finds higher gas exchange rates than Richey et al. (2002) and thus suggests that the total $CO_2$ evasion must be considerably higher. As the

results summarized in Fig. 16 suggest, our $CO_2$ evasion rates per water surface area are comparable to those of Rasera et al. (2013). Assuming that we underestimate the average flooded area, we conclude that we likely underestimated the $CO_2$ inputs from flooded soils and vegetation and the $CO_2$ evasion from the water surface to the atmosphere. In the future, improved floodplain forgings and simulations at higher spatial resolution might help to overcome these underestimations.

Although our estimates of $CO_2$ evasion from inland waters of the central Amazon basin is slightly higher than those of Richey et al. (2002), the same conclusion does not hold when assessing the $CO_2$ budget for the whole Amazon basin. The upscaling of Richey et al. (2002) led to a total $CO_2$ evasion estimate of 470 Tg C yr$^{-1}$ while our simulation, which explicitly accounts for spatial heterogeneities across the basin leads to a total $CO_2$ evasion of only about 379±46 Tg C yr$^{-1}$.

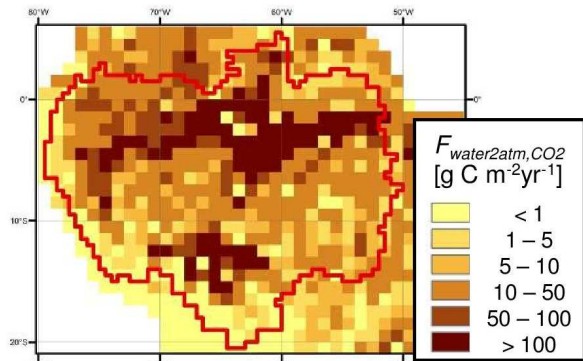

**Figure 16.** Simulated average $CO_2$ evasion from rivers, floodplains, and headwaters (summed up as $F_{water2atm}$) for the period 1980-2000. The evasion flux is reported relative to the total area of each grid cell.

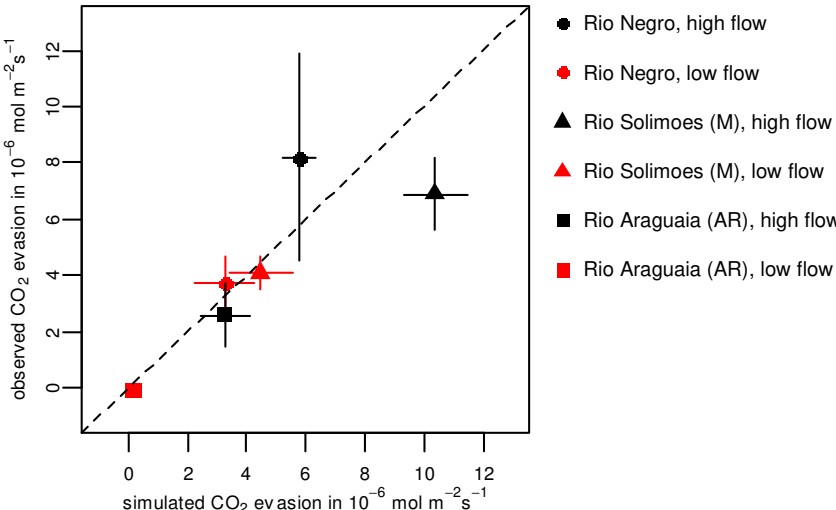

10  **Figure 17.** Observed versus simulated $CO_2$ evasion rates per water surface area ($R^2$=0.64, RMSE = 1.7 µmol $CO_2$ m$^{-2}$s$^{-1}$). Observed data are from Rasera et al., 2013. Reported are means and standard deviation of the observed values between 2006 and 2010. Note that due to the coarse resolution of our model, only data from the largest rivers (catchment area> 100,000 km²) are taken into account. The simulated values refer to the average evasion rate during low (monthly avg. discharge < yearly avg. discharge) and high flow periods (monthly avg. discharge > yearly avg. discharge) across the whole simulation period (1980-2000). The whiskers represent the standard deviations from the inter-
15  annual variations.

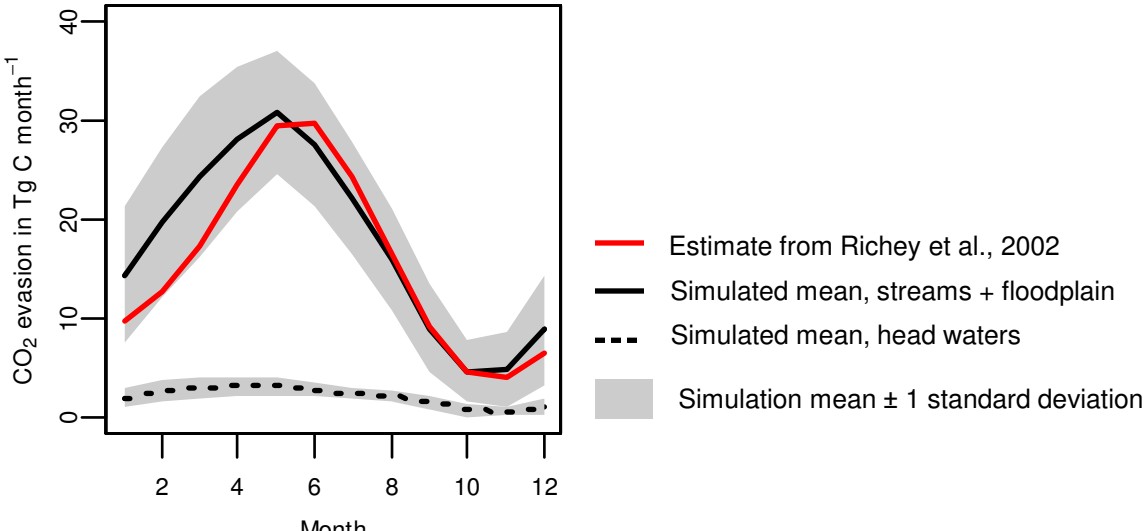

**Figure 18.** Seasonality in $CO_2$ evasion from inland waters (rivers plus floodplains, including swamps) within the central Amazon basin (see map in Fig. 4). The simulation result report the mean monthly $CO_2$ evasion during the simulation period 1980-2000 as well as the standard deviation of monthly mean simulated values during the same period. The $CO_2$ evasion from headwaters is here represented by the $CO_2$ evasion from $S_{fast}$. Simulation results are compared with the observation-based estimate by Richey et al., 2002, given here as the sum of the evasion from the Amazon main channel, the tributaries, and the floodplains. $R^2=0.85$, RMSE=23%)..

### 3.3 Synthesis of simulation results

Figure 19 summarizes the simulated fluxes of dissolved C, i.e. the sum of DOC and $CO_2$, through the river network of the Amazon basin. The total simulated export of carbon from the basin amounts to 413.9±50.0 Tg C yr$^{-1}$, to which lateral exports to the coast contribute only 8.3%, while the remainder is contributed by $CO_2$ evasion from the inland water surface. 57% of the total dissolved carbon inputs is contributed by flooded soils and litter. Surface runoff and drainage contribute 14% and 28%, respectively. It is interesting that the flux of carbon via throughfall onto the topsoil is as high as the lateral exports of dissolved C from the topsoil, although it is not necessarily its source. According to our simulations, about 8% of the dissolved C mobilized into the water column are reinfiltrating into the soil column in swamps ($F_{up2swamp}$) or on floodplains ($F_{flood2soil}$).

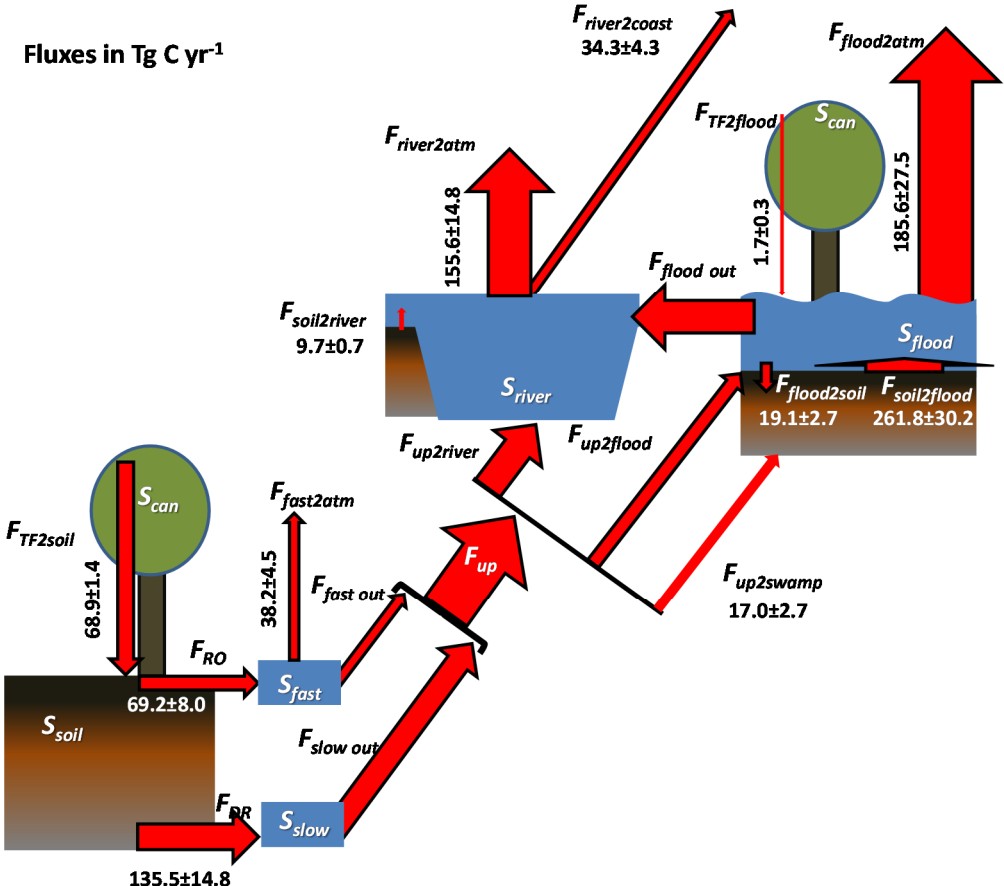

Fig. 19: Simulated fluxes of dissolved carbon (DOC + CO$_2$) through the inland water network of the Amazon basin. Numbers are average annual fluxes ± standard deviations over the simulation period 1980-2000.

### 3.4 Simplification of biogeochemical processes in the river network

5    The representation of biogeochemical transformation processes between different C species in the water column of the inland water network is rather simplistic. In the light of the limited empirical basis for calibration and validation on the one side, as well as the rather uncertain boundary conditions provided by the forcing data and structural model uncertainties to represent terrestrial biogeochemical processes for tropical forests on the other side, a more detailed representation of in-river processes is, for the time being, hardly achievable. Moreover, the validation supports the idea that ORCHILEAK represents the dominant

10  aquatic C cycle processes at the scale of the major sub-basins in a rather satisfactory way. In the following, we shortly discuss the main limitations and future perspective towards improving the simulation of in-stream biogeochemical processes.

One of the major future steps would be the implementation of particulate organic C (POC) fluxes in ORCHILEAK. Of the TOC fluxes at Obidos, the most downstream sampling location on the Amazon main stem, POC contributes less than one

fourth of the total flux (Moreira-Turcq et al., 2003; Ward et al., 2015), and was reported to further decrease to only about 10% downstream to the river mouth (Ward et al., 2015). The decomposition of this POC, which is mainly derived from floodplain litter, has been reported to contribute substantially to the in-river $CO_2$ production in the lower part of the Amazon (Ward et al., 2013). Our simulation results also highlighted the substantial contribution of submerged leaf litter to the $CO_2$ evasion.

However, in our simulation, POC is not transported downstream with the water flow, i.e. it is assumed to decompose locally, and only the DOC and dissolved $CO_2$ produced from this decomposition are transferred laterally. The representation of POC transport would induce a downstream shift in the simulated DOC and $CO_2$ production from POC. The lack of this representation might have induced a bias in the simulated longitudinal pattern of DOC concentrations, $pCO_2$ and $CO_2$ evasion with an overestimation of upstream values compared to downstream values. With the limited availability of evaluation data

and the rather simplified representation of POC and DOC decomposition in the model, it is impossible to conclude whether the lack of representation of POC transport explains part of the discrepancy between observed and simulated DOC concentrations (Fig. 13), or whether a too low DOC decomposition rate compensates for the bias. Mayorga et al. (2005) found that there must be a small, rapidly cycling pool of young organic matter from terrestrial vegetation close to the river that sustains a high $CO_2$ concentrations of a young $^{14}C$ age, while the majority of the transported POC is substantially older. The

actual effect of POC transport shifting $CO_2$ evasion downstream is thus likely rather limited. Nevertheless, a more complete representation of fluvial POC and DOC exports would be highly beneficial to constrain dynamic boundary conditions for an ocean biogeochemical model of the Amazon plume. The application of ORCHILEAK to rivers with substantial soil erosion driven POC exports will require the implementation of soil erosion and sediment transport modules (Naipal et al., 2015, 2016). The next major simplification in ORCHILEAK is the exclusion of autochthonous sources of TOC. In most parts of the Amazon

River system, in-river autotrophic production is inhibited by the high water turbidity due to sediment fluxes from the Andes and, thus, most of the exported TOC is of allochthonous sources (Moreira-Turcq et al., 2003). For the application to more eutrophic, heavily dammed rivers, autotrophic production plays a non-negligible role in the aquatic organic C cycle (Maavara et al. accepted). However, the simulation of in-river autotrophic production requires the synchronous simulation of potentially limiting nutrients, nitrogen (N) and phosphorous (P), as well as of the light conditions as another limiting factor of algae growth

(Billen et al., 1994). Taken the recent efforts in coupling the terrestrial C-N-P cycles in ESMs (e.g. Goll et al., 2012), the simulation of nutrient lateral transfers along the land-aquatic continuum seems a realistic target in the coming years. The implementation of dams into a river routing scheme (Lehner et al., 2011; Maavara et al., 2017; Zarfl et al., 2014) could also support this development.

For the decomposition of DOC in-transit, we considered here two pools of DOC with different, water temperature dependent

decomposition rates. So far, our approach does not distinguish between heterotrophic respiration of DOC and photo-oxidation, which would make the simulation of the DOC fate more complex. For heterotrophic respiration, inclusion of priming effects of more labile organic carbon on the decomposition of more refractory fractions could also be implemented (Guenet et al., 2014; Ward et al., 2016). Here, in particular, the labile pools produced by autotrophic processes could be of importance.

Moreover, the production and decomposition of organic C, N and P would need to be coupled if the effect of the C:N:P ratios of organic matter on its degradability is to be accounted for. In addition, particularly if POC is concerned, a representation of the heterotrophs in the ecosystem could be useful, including the "shredders" responsible for the physical breakdown of POC (Yoshimura et al., 2010) and "grazers" that feed on algae (Billen et al., 1994). Finally, photo-oxidation plays an important

role in the breakdown of chromatic dissolved organic matter (CDOM), which is usually highly resistant to heterotrophic degradation. This process is likely important in black water systems like e.g. the Rio Negro (Amon and Benner, 1996). If this process was to be simulated, one would need to distinguish CDOM as a distinct species and precise information on light-penetration depth and river-channel geometry would be required. For further developments in the modelling of DOC and POC decomposition in transit, a stronger empirical basis is needed, in particular for tropical river systems like the Amazon.

**4 Conclusion and Outlook**

ORCHILEAK reproduces observed DOC and $CO_2$ fluxes in the Amazon basin, and their seasonal to interannual variability, at least at the scale of the major sub-basins. As highlighted in the introduction, we consider that the explicit simulation of lateral export of soil and litter material to river headstreams and further down to the tropical ocean using an approach consistent with existing representations of terrestrial ecosystem carbon and water budgets, is a major step forward in physically based,

integrated modelling approaches of the global C cycle. Currently, the empirical basis for calibration and validation of these lateral fluxes and their fate within the aquatic system is still limited for the Amazon basin. Nevertheless, the simulated terrestrial inputs are within the ranges reported in the literature, and the basin-scale export fluxes agree well with observed fluxes. An improved representation of spatial heterogeneities and peculiar environments such as black-water systems will require even higher spatial resolution (0.25° or less), improved regional soil, wetland, and climate forcings as well as

observations with higher spatial and temporal coverage for calibration and validation.

In this study, ORCHILEAK was applied to the Amazon using upgraded regional wetland and climate forcing files. In order to apply ORCHILEAK to other river systems, similar forcings will have to be constructed using the methodology described in Guimberteau et al. (2012) and in this study. In the future, ORCHILEAK is intended for global scale applications. Before this objective can be reached, however, the new model branch will have to be tested at regional scale in other river basins pertaining

to different climate zones and ecosystem types. Adaption of the parameterization and, if required, implementation of additional key processes will need to be considered. The latter will, for instance, be important in high latitude rivers under the influence of permafrost, an ecosystem subject to distinct physical and biogeochemical processes currently not included in ORCHILEAK. ORCHILEAK will in future be augmented with additional transported species, in particular POC and nutrients. The simulated export fluxes to the coast will also provide useful time-dependent boundary conditions for ocean biogeochemistry models.

Finally, ORCHILEAK will be useful to better assess the terrestrial C sink in ESM simulations by taking into account the permanent leakage of C from the plant-soil system. In the long run, our new model could also help better constrain terrestrial

C cycle-climate feedbacks, future atmospheric $CO_2$ levels and temperature for different scenarios of anthropogenic $CO_2$ emissions.

**5 Code availability**

ORCHILEAK is derived from ORCHIDEE with the modifications presented in Sect. 2. A description of the general code of
5    ORCHIDEE can be found here: http://forge.ipsl.jussieu.fr/orchidee/browser#tags/ORCHIDEE_1_9_6/ORCHIDEE. For ORCHIDEE, the main part of the code was written by Krinner et al. (2005). For the general description of the basic river routing scheme see d'Orgeval et al. (2008). For the upgraded soil C module see Camino Serrano (2015). In the supplement information of this MS, you find the source-code of ORCHILEAK and a short instruction how to install the code. For more general information about how to install ORCHIDEE and its different branches, please consult:
10    http://forge.ipsl.jussieu.fr/orchidee/wiki/Documentation/UserGuide.

**Table A.1.** Abbreviations used in the text.

| | Abbreviation | Meaning | Restrictions |
|---|---|---|---|
| | | *acronyms used for fluxes* | |
| Main symbol | $F$ | Flux (given in g C $dt^{-1}$ or kg $H_2O$ $dt^{-1}$) | |
| 1st subscript | *add2can* | Addition of DOC by leaf leachates and dry deposition | only DOC |
| (path) | *can2ground* | Flux from canopy to ground | DOC, $H_2O$ |
| | *dec terr* | Produced from decomposition in/on non-flooded soils | DOC, $CO_2$ |
| | *dec flood* | Produced from decomposition in/on flooded soils | DOC, $CO_2$ |
| | *DR* | Export from soil column by drainage | |
| | *fast out* | Outflow from the fast reservoir | |
| | *fast2atm* | $CO_2$ evasion from fast reservoir (headwaters) | only $CO_2$ |
| | *flood out* | Outflow from the flood reservoir | |
| | *flood2atm* | $CO_2$ evasion from water surface on floodplain | $CO_2$, $H_2O$ |
| | *flood2soil* | Infiltration into flooded top soil | |
| | *river out* | Outflow from the river reservoir | |
| | *river2atm* | $CO_2$ evasion from river surface | only $CO_2$ |
| | *RO* | Export from soil column by surface runoff | |
| | *slow out* | Outflow from the slow reservoir | |
| | *soil adv* | Advection of DOC between soil layers | |
| | *soil2flood* | Inputs from decomposition of SOC, litter to water column of $S_{flood}$ | DOC, $CO_2$ |
| | *soil2river* | Inputs from decomposition of SOC, litter to water column of $S_{river}$ | DOC, $CO_2$ |
| | *TF* | The total flux of DOC and water to the ground | DOC, $H_2O$ |

|  | Abbreviation | Meaning | Restrictions |
|---|---|---|---|
|  | *up* |  | Flux entering the basin from upstream |  |
|  | *up lim* |  | $F_{up,H2O}$ to be exceeded before flooding starts | only $H_2O$ |
|  | *up2flood* |  | Flux from upstream to flood reservoir |  |
|  | *up2river* |  | Flux from upstream to stream reservoir |  |
|  | *up2swamp* |  | Flux from upstream to swamp, infiltrates into sub soil |  |
|  | *WD* |  | Wet deposition | DOC, $H_2O$ |
|  | *WD2can* |  | Wet deposition onto canopy | DOC, $H_2O$ |
|  | *WD2ground* |  | Wet deposition onto ground | DOC, $H_2O$ |
| 2nd subscript |  | *H2O* | Water |  |
| (transported |  | *DOC* | Total DOC |  |
| species) |  |  |  |  |
|  |  | *DOClab* | Labile DOC |  |
|  |  | *DOCref* | Refractory DOC |  |
|  |  | *DOC pool* | Distinctively for each of the following three DOC pools |  |
|  |  | *DOC active* | Active DOC pool (soil C module) |  |
|  |  | *DOC slow* | Slow DOC pool (soil C module) |  |
|  |  | *DOC passive* | Passive DOC pool (soil C module) |  |
|  |  | *SOC pool* | Distinctively for each of the following three soil organic carbon (SOC) pools |  |
|  |  | *SOC active* | Active SOC (soil C module) |  |
|  |  | *SOC slow* | Slow SOC pool (soil C module) |  |
|  |  | *SOC passive* | Passive SOC pool (soil C module) |  |
|  |  | *Litter pool* | Distinctively for each of the two litter pools |  |
|  |  | *Litter met* | Metabolic litter |  |
|  |  | *Litter str* | Structural litter |  |
|  |  | *CO2* | Dissolved carbon dioxide |  |

| | Abbreviation | Meaning | Restrictions |
|---|---|---|---|
| 3nd subscript | $i$ | Basin (subunit of grid cell $x$) | |
| | $i\text{-}1$ | Basins lying upstream of basin $i$ | |
| | $grid\ x$ | Grid cell containing basin $i$ | |
| 4th subscript | $v$ | Plant function type. If nof indicated, same flux for all PFTs | |
| 5th subscript | $l$ | Soil layer (1 to 11) | |
| 6th subscript | $t$ | Time-step | |
| | | *Acronyms used for storages* | |
| Main symbol | $S$ | Storage given in g C or kg $H_2O$ | |
| 1st subscript | $can$ | Canopy | DOC, $H_2O$ |
| (ecosystem | $fast$ | Fast reservoir | |
| compartment) | $flood$ | Flood reservoir | |
| | $river$ | River reservoir | |
| | $slow$ | Slow reservoir | |
| | $soil$ | Soil column | |
| 2nd to 5th subscript | | *Same as for fluxes* | |
| | | *other acronyms (subscripts i/x, v, l, and t correspond to 3rd to 6th subscript described above)* | |
| | $A_{flood}$ | Water surface area of $S_{flood}$ [m²] | |
| | $A_{river}$ | River surface area [m²] | |
| | $A_{river\ small}$ | Area of rivers with a width≤100m [m²] | |
| | $A_{river\ large}$ | Area of rivers with a width>100m [m²] | |
| | $A_{river\ basic}$ | River surface area [m²] at low water stage [m²] | |
| | $A_{river\ act}$ | Actual $A_{river}$ [m²] that can be larger than $A_{river\ basic}$ | |
| | $A_{total}$ | Area of the grid cell or basin (dep. on subscript) [m²] | |
| | $b$ | Parameter describing shape of floodplain (see text) | |
| | $CUE$ | Carbon use efficiency (fraction of organic C that is transformed to another form of organic C) | |
| | $dt$ | Time step used for soil C and vertical fluxes (=30 min) | |

| Abbreviation | Meaning | Restrictions |
|---|---|---|
| *floodcri* | Constant [m] (default 2m) used in TRUNK version in simulation of actual flood extent, in ORCHILEAK replace by *floodh$_{95th}$* | |
| *floodh* | Water level over floodplain [m] | |
| *floodh$_{95th}$* | 95$^{th}$ percentile of *floodh$_{i,t}$* over simulation period [m] | |
| *f$_{swamp}$* | Fraction of $F_{up}$ that is diverted to the bottom soil layer | |
| $K_{CO2}$ | Solubility constant of $CO_2$ [mol L$^{-1}$ atm$^{-1}$] | |
| $k_{DOClab}$ | Decomposition rate of labile DOC at $T_{water}$= 28°C [day$^{-1}$] | |
| $k_{DOCref}$ | Decomposition rate of refractory DOC at $T_{water}$= 28°C [day-1] | |
| $k_{flood}$ | Gas exchange velocity for $CO_2$ [m day$^{-1}$] from floodplains, mix of the $k_{river}$ and $k_{swamp}$ | |
| $k_{river}$ | Gas exchange velocity for $CO_2$ [m day$^{-1}$] from open water | |
| $k_{swamp}$ | Gas exchange velocity for $CO_2$ [m day$^{-1}$] from flooded forests | |
| $k_{*,600}$ | Gas exchange velocity [m day$^{-1}$] for $CO_2$ in *=river,swamp or flood at 20°C | |
| $k_{SOC\,pool}$ | Decomposition rate of the active, slow or passive SOC pool | |
| $k_{litter\,pool}$ | Decomposition rate of the metabolic or structural litter pool | |
| $k_{DOC\,pool}$ | Decomposition rate of the active, slow or passive DOC pool | |
| *leaf biomass$_{i,v,t}$* | Biomass allocated to leaves [g C m$^{-2}$] | |
| $pCO_{2\,atm}$ | Atmospheric partial pressure of $CO_2$ [atm] | |
| $pCO_{2\,fast}$ | Aquatic partial pressure of $CO_2$ in $S_{fast}$ [atm] | |
| $pCO_{2\,river}$ | Aquatic partial pressure of $CO_2$ in river [atm] | |
| $pCO_{2\,flood}$ | Aquatic partial pressure of $CO_2$ in floodplain [atm] | |
| *red$_{RO}$* | Combined reduction factor for exports with runoff | |

**Table A.1.** continued

| Abbreviation | Meaning | Restrictions |
|---|---|---|
| $red_{DOC}$ | Reduction factor for vertical, advective DOC fluxes and lateral DOC export from soil column (set to 0.2) | |
| $red_{connect}$ | Reduction factor for exports with runoff depending on extends of saturated soils around headwaters | |
| $SC$ | Schmidt-number | |
| $S_{fast+slow,H2O,ref}$ | Reference storage of water [mm] in $S_{fast}$ and $S_{slow}$, at which $red_{connect} = 1.0$ (set to 160 mm) | |
| $T_{ground}$ | Mean daily air temperature near the surface [°C] | |
| $T_{water}$ | Mean daily water temperature [°C] | |
| $Topo$ | Topographic index of the grid cell, taken from forcing | |
| $w$ | Mass fraction of a solute per solvent (water) | |
| $RO$ | ..in runoff | |
| $DR$ | ..in drainage | |
| $\tau$ | Factor which translates $Topo$ into a water residence time in the fast, slow, river, flood reservoir | |
| $\%flood$ | Temporally changing, actually flooded proportion | |
| $\%flood_{max}$ | Maximum floodable proportion | |
| $\%flood_{pot}$ | Potentially flooded fraction depending of water storage | |
| $\%flood_{river}$ | Temporally changing flooded fraction close to river | |
| $\%flood_{total}$ | $\%flood_{river} + \%flood$ | |
| $\%lignin$ | Lignin content (mass fraction) in the structural litter | |
| $\%poorsoils$ | Areal proportion of Podzols+Arenosols+Blackwater swamps | |
| $\%swamp$ | Area proportion of swamps in grid box | |

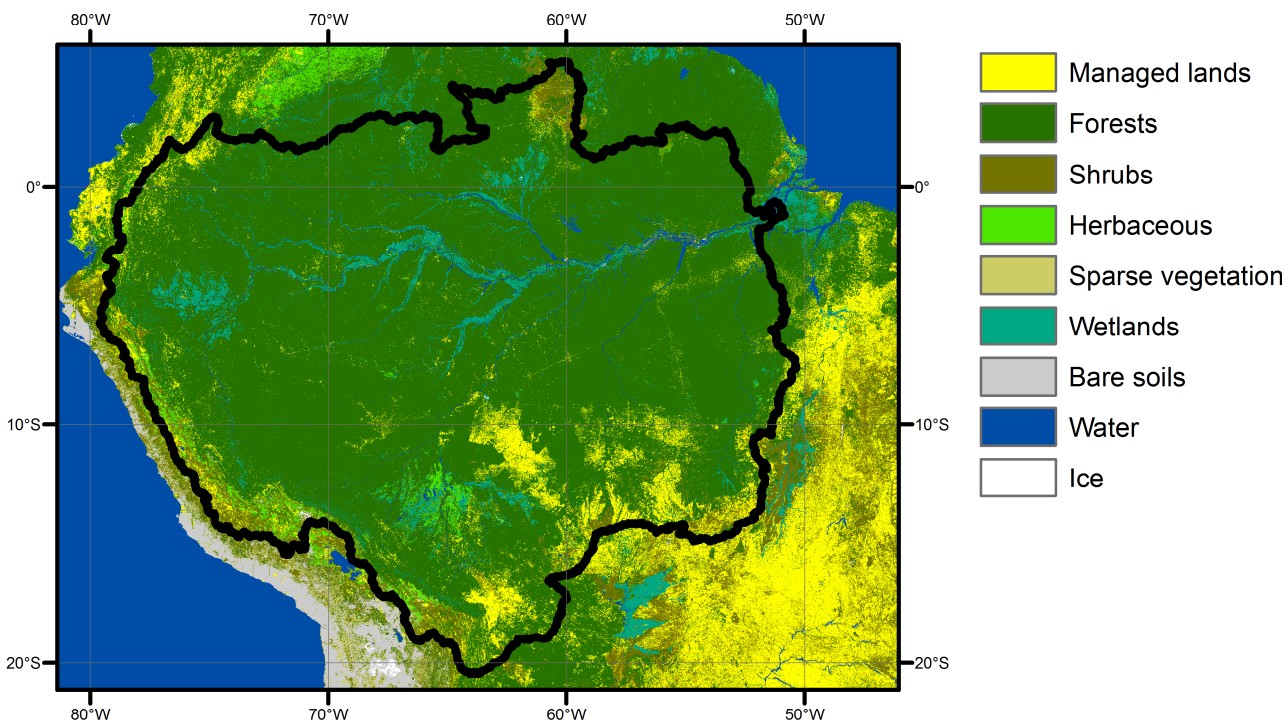

Fig. A.1: Land cover composition of the study area as representative for the years 2005/2006 derived from GLOBCOVER data (Arino et al., 2008). The black outline represents the Amazon watershed.

**Acknowledgements** Funding was provided from the French National Research Agency ("Investissement d'Avenir", n°ANR-10-LABX-0018), from the Université Libre de Bruxelles, through the Bureau des relations internationales (BRIC), the European Union's Horizon 2020 research and innovation program under grant agreement no.703813 for the Marie Sklodowska-Curie European Individual Fellowship "C-Leak" and the European Union's Horizon 2020 research and innovation program under Marie Sklodowska-Curie grant agreement no. 643052 (C-CASCADES project). MC and MG acknowledge funding from the European Research Council Synergy grant ERC-2013-SyG-610028 IMBALANCE-P. We are particularly grateful to Camille Risi who shared her version of the river routing code with tracer transport equations at the early stage of the project, as well as Laurent Bopp and Marion Gehlen for discussions regarding the river's carbonate system.

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
