# Peer review of "ORCHILEAK (revision 3875): A new model branch to simulate carbon transfers along the terrestrial-aquatic continuum of the Amazon basin"

_Geoscientific Model Development, 2017_

## Short Comment (SC1) · 7 Apr 2017

Dear authors,

in my role as Executive editor of GMD, I would like to bring to your attention our Editorial version 1.1:

http://www.geosci-model-dev.net/8/3487/2015/gmd-8-3487-2015.html

This highlights some requirements of papers published in GMD, which is also available on the GMD website in the 'Manuscript Types' section:

http://www.geoscientific-model-development.net/submission/manuscript_types.html

[Figure]

In particular, please note that for your paper, the following requirement has not been met in the Discussions paper:

- "The main paper must give the model name and version number (or other unique identifier) in the title."

Please add a version number for ORCHILEAK in the title upon your revised submission to GMD.

Yours,

Astrid Kerkweg

---

## Referee Comment (RC1) · Anonymous Referee #1 · 10 May 2017

General comments

It has been recognized that the global significance of the inland freshwater carbon cycle. However, it has been completely overlooked owing to the inadequate data and improper models. The new model branch, ORCHILEAK evolved from ORCHIDEE, not only improves our understanding of carbon transformations but also provides a fundamental for the assessment of the impacts of climate change. The main purpose is clear and it is worth developing to quantify the lateral exports of carbon off the terrestrial ecosystems. The results showed that QRCHILEAK could successfully simulate the fluvial transport of DOC and $CO_2$ evasion in Amazon basin. However, I have several specific comments from the perspectives on the model development and verification.

Specific comments

1. What was the time step to execute this model? Several spatial and temporal resolutions were mentioned, e.g. at a 30 min time step and a 1º (or 0.5º) resolution for the hydrology model and at a 6 min to daily time steps and a 1º (or 0.5º) resolution for the C fluxes. I was wondering how the model was executed in practice. How did different time steps work together in one model? Besides, did the choices of time steps operate in coordination with the spatial resolutions?

2. I would suggest a table showing the input details essential for the model execution and the outputs, perhaps including their spatial and temporal resolutions.

3. There is not any information regarding the calibration and validation, e.g. the performance measures in the calibration and validation.

4. P21, L8. What is the performance in terms of the spatial pattern of flood area?

5. Fig. 9. As mentioned above, were there any performance measures showing how good the simulations were? What were the results for monthly time series?

6. Table 1b. I was wondering how the SOC was simulated. What was the initial condition in the model? Did the simulated SOC change with time? How much did it change?

7. Fig. 14. It is found the simulated DOC basically varied within a small range

around 4 mgL$^{-1}$ except the simulations at M, implying the simulated seasonal variability of DOC fluxes (in Fig. 12) were mainly attributed to the discharge not DOC concentration. I speculate the monthly time series for Q would mimic the DOC fluxes. Are these persuasive results for a DOC model? Besides, I wouldn't say the simulations reproduce well DOC concentration (L1, P33).

8. CH4 evasion was negligible. How about the influence of DIC?

9. Is it possible to give a diagram illustrating the numbers of each C flux for the study watershed, just like Fig. 3 but with numbers on each arrow?

10. Fig. 4. What do the yellow color stand for? Where are the discharge gauges? Do you think landuse map is helpful?

11. Table 4. What do the stars indicate? Why is the surface runoff represented by % for the RO3?

---

## Referee Comment (RC2) · Anonymous Referee #2 · 17 Jul 2017

Overall the authors present compelling work to address a major deficiency in earth system models. The authors demonstrate successful simulation of CO2 and DOC lateral transport and CO2 evasion in the Amazon using ORCHILEAK. Previously, no ESM models existed that simulate the lateral transport of CO2 and DOC from surface water sources. The ORCHILEAK enables scientists to attribute DOC and evasion sources. Additionally, the model has the ability to quantify the CO2 evasion in relation to terrestrial net primary production. The authors partition these processes into flooded and non-flooded lands with regard to soil carbon and DOC throughfall and subsequent lateral transport and evasion. Below I provide general and specific comments for the authors to consider to improve the current form of the manuscript.

[Figure]

The authors clearly describe the work that was instrumental to their contribution, and present sufficient references to support their model advancement. The authors provide a succinct abstract summarizing the modeling advancement, results, and potential future application. The authors explicitly show where code area available to replicate this project and provide this in the supplement with good instructions. Including a table of all forcing and evaluation datasets and sources would aid efforts of anyone attempting to build upon the authors work. The equations and formulae are correctly defined, but providing all the equations within the appendix hinders the reader's ability to distinguish which equations are novel to the manuscript. The manuscript would benefit by including a table describing each variable in the appendix, not in a different file in the supplement, or by including the equations in line throughout the document with variable description.

1. The general methods are clearly outlined, however ambiguity exists in the temporal resolution of the model implementation. Specifically, many temporal resolutions are mentioned: i.e. 6 min, 30 min, and daily. Clarification on how these different components of the model interact would enhance clarity. Adding a flow chart or including this within Figure 3 would add clarity.

2. There is limited discussion on calibration. After reading the manuscript, the majority of the parameters seem to be taken from the literature. Please describe the calibration process referenced in Sections 2 and 3.

3. The authors mention this briefly on page 37 L16-21 and on 38 L2-4 how the lack of representation of POC transport might shift in stream DOC and $CO_2$ production downstream. Please expand upon how the lack of representation impacts the current model evaluation including the impact on aggregate downstream DOC, POC, and $CO_2$ evasion. How would this impact the evaluation results presented in Figure 12 and 13? Does this mean the current form of the the model over/under-compensates for the lack of mobile POC? Which parameters would be impacted?

[Figure]

4. Figure 14 displays the performance of modeled DOC concentration relative to observations. Removing the data from Rio Negro from this plot reveals that the model only produces ranges of DOC from 3-5 mg C/L while observations are double that range [1-7 mg C/L]. How do you reconcile the low variability of the model relative to the observations? This seems in contrast to the results of the simulation data presented in Table 4.

5. The authors note how $CO_2$ evasion is comparable to data from Richey et al., 2002, but also disclose that the inundated fraction is greatly underestimated for the central Amazon. The central Amazon shows that the model has the highest evasion rates there [Fig. 15]. The concluding remarks on Page 35, Lines 10-17 should explicitly address how the match in $CO_2$ evasion and mismatch in inundation are related. Specifically, how does the underestimation of inundated extent impact the assertion that 51% of $CO_2$ evasion is attributed to the floodplain?

Specific comments: Figure 3 caption refers to Table 1, I believe the authors intend to reference Table S1.

---

## Author Comment (AC1) · 14 Aug 2017

Interactive comment on "ORCHILEAK: A new model branch to simulate carbon transfers along the terrestrial-aquatic continuum of the Amazon basin" By Ronny Lauerwald et al. A. Kerkweg, kerkweg@uni-bonn.de,

Dear authors, in my role as Executive editor of GMD, I would like to bring to your attention our Editorial version 1.1: http://www.geosci-model-dev.net/8/3487/2015/gmd-8-3487-2015.html This highlights some requirements of papers published in GMD, which is also available on the GMD website in the 'Manuscript Types' section: http://www.geoscientific-model-development.net/submission/manuscript_types.html In

particular, please note that for your paper, the following requirement has not been met in the Discussions paper: • "The main paper must give the model name and version number (or other unique identifier) in the title." Please add a version number for ORCHILEAK in the title upon your revised submission to GMD. Yours, Astrid Kerkweg

We changed the title accordingly to "ORCHILEAK (revision 3875): A new model branch to simulate carbon transfers along the terrestrial-aquatic continuum of the Amazon basin"

---

## Author Comment (AC2) · 14 Aug 2017

Reviewer #1

General comments
It has been recognized that the global significance of the inland freshwater carbon cycle. However, it has been completely overlooked owing to the inadequate data and improper models. The new model branch, ORCHILEAK evolved from ORCHIDEE, not only improves our understanding of carbon transformations but also provides a fundamental for the assessment of the impacts of climate change. The main purpose is clear and it is worth developing to quantify the lateral exports of carbon off the terrestrial ecosystems. The results showed that QRCHILEAK could successfully simulate the fluvial transport of DOC and CO2 evasion in Amazon basin. However, I have several specific comments from the perspectives on the model development and verification.

Specific comments
1. What was the time step to execute this model? Several spatial and temporal resolutions were mentioned, e.g. at a 30 min time step and a 1o (or 0.5o) resolution for the hydrology model and at a 6 min to daily time steps and a 1o (or 0.5o) resolution for the C fluxes. I was wondering how the model was executed in practice. How did different time steps work together in one model? Besides, did the choices of time steps operate in coordination with the spatial resolutions?

In theory, the length of all time-steps can be changed. Here, for reason of simplicity, we want to stick to the standard setting at which the model was run. Practically, the whole model runs at a 30-minute time step. However, the lateral routing of water and dissolved C is executed only every 48$^{th}$ time-step, i.e. when one day is over. At each 30 min time-step, the routing module of ORCHILEAK aggregates the runoff, discharge and C inputs to the river or floodplains simulated in the soil C module. When the end of the day is reached, the lateral flows of water and carbon are simulated for that whole day using those aggregated water and C inputs. Similarly, the flows of C and water back to the soil column (Fflood2soil, Fup2swamp), which are simulated in the routing module at this daily time step, are used as inputs to the soil hydrology and soil carbon modules at each of the 48 30-minute time steps over the following day of simulation, simply by evenly distributing the daily fluxes over the 48 30-minute time steps.

The 6-minute time step is used as an iteration (240 iterations per day) to approximate the continuous interplay of $CO_2$ inputs to the water column and $CO_2$ evasion from the water column using a sufficiently short time-step. $CO_2$ inputs are increasing the water-atmosphere $pCO_2$ gradient, while the $CO_2$ evasion flux also controls the water-atmosphere $pCO_2$ gradient by continuously decreasing it. All the inputs of $CO_2$ to the water column which have been aggregated to the daily time step of the routing module, be it from decomposition of DOC in the water column, the decomposition of submerged litter or from the respiration in inundated soils, are thus split over the 240 6-minute time-steps of the day. The $pCO_2$ and the $CO_2$ evasion are calculated for each of those 6-minute time-steps, and the $CO_2$ storage in the water column is updated accordingly. The $CO_2$ evasion is then simply aggregated over the 240 iteration time-steps to obtain the daily values.

We followed the advice of Reviewer 2 and included the information on different temporal resolutions into Figure 3. A colour code, explained with a legend, now indicates the temporal resolution of each flux.

We also added a more detailed description for the choice of the different time steps at the end of subsection 2.2.1:

"Like the cycling of water and C in vegetation and soils, the allochthonous inputs of DOC from $S_{can}$ and $S_{soil}$ into the inland water network ($F_{RO}$, $F_{DR}$, $F_{soil2flood}$, $F_{soil2river}$, see Fig 3) are computed at a temporal resolution of 30 minutes and at the spatial resolution of the grid cell. The lateral transfer between the $S_{fast}$, $S_{slow}$, $S_{river}$ and $S_{flood}$ and the transformation of C within those storage reservoirs are only simulated at a daily time step and at the spatial resolution of the basin. Therefore, to simulate the lateral transfers, the allochthonous DOC and $CO_2$ inputs are first aggregated over 48 30-minute time steps until one full day is over. The fluxes from the water column back into the soil column ($F_{flood2soil}$,$F_{up2swamp}$ in Fig. 3) are simulated at the daily time-step of the routing module, but are used as inputs in the soil carbon module, which runs at a 30 minute temporal resolution. This is achieved by downscaling the daily fluxes uniformly over the 48 30 minute time-steps of the following day of simulation. The evasion of $CO_2$ from river and floodplain water surface ($F_{river2atm}$, $F_{flood2atm}$) is also simulated at the daily time-step of the routing module, but to approximate the continuous interplay of $CO_2$ inputs and $CO_2$ evasion controlling the water-air gradient in $CO_2$ partial pressures ($pCO_2$) a much shorter time-step of 6 minutes is used, and the $CO_2$ inputs to the water column are thus uniformly distributed over the 240 6-minutes time-step contained in each day."

2. I would suggest a table showing the input details essential for the model execution and the outputs, perhaps including their spatial and temporal resolutions.
We agree with the reviewer suggestion and have added a new table 1 in the revised manuscript.

3. There is not any information regarding the calibration and validation, e.g. the performance measures in the calibration and validation.

We thank the reviewer for this important point. We have now added performance measures (Correlation coefficients, Nash-Sutcliffe efficiency and/or root mean squared errors) to all figures reporting observed vs. simulated values. In addition, to highlight the improvements in the new flooding scheme, we also compare these performance measures for the TRUNK version of ORCHIDEE and the recalibrated version implemented in ORCHILEAK. Similarly, we compare performance measures for the simulated discharges obtained with 1) the TRUNK version using the parameterization of Guimberteau et al. (2012); 2) the ORCHILEAK version using the modified floodplain forcing and the setup of Guimberteau et al. (2012), 3) the ORCHILEAK version with recalibrated discharge. For that purpose, we added a new table to section 3.1 of the revised manuscript summarizing the results of the recalibration.
The calibration of DOC exports from the soil to the river and the subsequent DOC decomposition was difficult because of data limitation, in particular for the exports through the soil-water interface. Thus, we took values from the literature whenever available but sometimes had to make assumptions for some parameters (e.g. $F_{fast+slow,H2O}$). However, we always tried to secure that our simulated DOC exports from the soils to the water (orig. Table 3, now Table 6) were within reasonable ranges. Similarly, we verified that our simulated DOC concentrations in the river were close to observed DOC concentrations. With our final parameterisation, the simulated average DOC concentrations are close to observed average DOC concentrations (see original Figure 14), for the entire set of observed concentrations over all stations (mean deviation of -2% relative to mean observed value) and at Obidos (mean deviation of -1% relative to mean observed value) in particular, the is the sampling location the most far downstream. In the revised version of our manuscript, we now report results from a sensitivity analysis (new Tables 7 and 9) for the most important parameters controlling DOC exports to the water column and decomposition within the water column ($F_{fast+slow,H2O}$, $red_{DOC,base}$, $k_{doc,lab}$, $k_{doc,ref}$).

To better accommodate these modifications into the manuscript, we restructured parts of the results and discussion sections. The parts of sections 3.2.3 (mobilization from soils to the river) and 3.3 (fluxes in and from the river to the atmosphere) dealing with DOC are now merged into a new section 3.3, while the parts dealing with $CO_2$ are merged into a new section 3.4 (the old section 3.4 becoming now section 3.5). In the new section 3.3, following the logic of reporting first the parameters calibration, we start with the comparison between observed and simulated DOC concentration in overland flow, drainage, and headwaters (now Table 6), and then report the results from the sensitivity analysis on DOC exports through the soil-water interface (Table 7). Next, we compare the observed and simulated DOC concentrations in the river (original Fig. 14), including the sensitivity analysis, and finish with the fluxes of DOC in the river (original Figs. 12 and 13, now Figs. 14 and 15).

For the new section 3.4, dealing with $CO_2$ evasion, we report at the end the results from the sensitivity analysis, highlighting the weak sensitivity of simulated $CO_2$ evasion to the gas exchange velocity: changing the gas exchange velocities by +/-50% only lead to a change in total $FCO_2$ from the Central Amazon basin of only +1% and -4%, respectively. Thus, the choice of a "right" gas exchange velocity does not seem important.

4. P21, L8. What is the performance in terms of the spatial pattern of flood area?

We compared our simulated maximum inundation during the period 1995/1996 to the high-resolution airborne remote sensing data by Hess et al. (2015), which we aggregated to the 1-degree resolution of our simulation. The correlation in spatial patterns is satisfying ($R^2$=0.56, y=0.04+0.90*x (with y = observed data, x = our simulation)). We have added a figure to the revised manuscript (Figure 9), which compares our simulation to the aggregated observations by Hess et al., 2015.

5. Fig. 9. As mentioned above, were there any performance measures showing how good the simulations were? What were the results for monthly time series?

We have calculated the Nash-Sutcliff efficiencies, correlation coefficients and RMSEs for each time series and added them to all the relevant graphs (Figs. 8, 10, 13, 14, 15).

6. Table 1b. I was wondering how the SOC was simulated. What was the initial condition in the model? Did the simulated SOC change with time? How much did it change?

The soil carbon storage was produced by running the entire model for a simulation period of 1000 years, using the atmospheric $CO_2$ concentration and land cover of the year 1980 and looping over the climate forcing files for the years 1980 to 2000 (the whole of 21 forcing data produced by Guimberteau et al., (2012). These 5000 years simulation was started with an initially empty soil C storage, which then grew and reached a steady state, i.e. apart from minor fluctuation over the 21 year periods of the climate forcing, no significant trend in soil carbon storage occurred anymore. During this initialisation period, the soil carbon stock grew from 0 Pg C to 48 Pg C. Over the subsequent 21-year simulation period reported in this paper, the soil C storage in the Amazon basin fluctuated between 48.6 to 48.2 Pg C, with no trend.

In the revised manuscript, we clarified this point in the method section:
"To obtain initial soil carbon pools which are in steady-state with the model set-up for the 1980-2000 period, the model was first run for 5000 years, looping over the full set of climate forgings and using the land use and an atmospheric $pCO_2$ as representative for the year 1980. The terrestrial C pools

simulated during this initialization phase were subsequently used for the simulation over the period 1980-2000 with changing land cover and increasing atmospheric pCO₂."

7. Fig. 14. It is found the simulated DOC basically varied within a small range around 4 mgL-1 except the simulations at M, implying the simulated seasonal variability of DOC fluxes (in Fig. 12) were mainly attributed to the discharge not DOC concentration. I speculate the monthly time series for Q would mimic the DOC fluxes. Are these persuasive results for a DOC model? Besides, I wouldn't say the simulations reproduce well DOC concentration (L1, P33).

In Fig. 14, we made two little mistakes by reporting wrong simulated values for Obidos and Gaviao. With the corrected time-series, there is now little bit more variation in simulated values. However, the reviewer's observation is still correct and the simulated DOC concentrations show a significantly lower variability compared to the observed values, and DOC fluxes therefore mainly follow the variations in simulated discharge.
We clarified this observation in the revised manuscript:
"Comparing observed vs. simulated DOC concentrations, we were able to reproduce the average concentrations at least in the main stem of Rio Solimoes/Amazon River and in the Rio Negro. However, apart for the Rio Negro, we generally underestimate the seasonal variability of DOC concentrations."

8. CH4 evasion was negligible. How about the influence of DIC?

We agree with the reviewer that this is an important point. According to Richey et al., (1990), the fluvial exports of DIC (at Obidos) amount to 35 Tg C yr$^{-1}$, which is significantly higher than the export of DOC of 22.4 Tg C yr$^{-1}$ after the same source. The same authors report that throughout the Amazon main stem and its major tributaries, 60-90% of the DIC is in the form of $HCO_3^-$ derived from chemical rock weathering, while the remainder is mainly free dissolved $CO_2$, concentrations of $CO_3^{2-}$ being negligible at the pH values of 6.5 to 7.2 typically reported for the Amazon basin. Thus, the fluvial export of free dissolved $CO_2$ would lie between 4 and 14 Tg C yr$^{-1}$. According to our simulation, the average fluvial flux of dissolved $CO_2$ at Obidos amounts to 8.3 Tg C yr$^{-1}$, and lies thus within the range reported by Richey et al. 1990. Rock weathering and the related $HCO_3^-$ fluxes are not represented in ORCHILEAK.
Dissolved $CO_3^{-2}$ in the river water could exert a buffering effect on $CO_2$ produced and transported in the water column by the reaction $H_2O+CO_2+CO_3^{2-} => 2\ HCO_3^-$. However, as the $CO_3^{2-}$ concentrations are negligible in the Amazon, this buffering effect is negligible. For an application of ORCHILEAK to high pH rivers, like for example the Rhine, a full representation of the carbonate system including weathering related fluxes of DIC would be of higher importance to reproduce riverine $CO_2$ transport and evasion.
In the revised manuscript, we added two sentences to the introduction to include this important point:

"Further, we ignore the fluxes of carbonate alkalinity as at average pH values of 6.5 to 7.2 typical of the Amazon basin (Richey et al., 1990) the concentrations of $CO_3^{2-}$ are negligible and, thus, the carbonate-buffering of $CO_2$ is limited."

9. Is it possible to give a diagram illustrating the numbers of each C flux for the study watershed, just like Fig. 3 but with numbers on each arrow?

We agree with the reviewer and produced a new figure according to his suggestion along with a short subsection "Synthesis **of simulation results" (now subsection 3.3).**

**"3.3 Synthesis of simulation results**

Figure 19 summarizes the simulated fluxes of dissolved C, i.e. the sum of DOC and $CO_2$, through the river network of the Amazon basin. The total simulated export of carbon from the basin amounts to 413.9±50.0 Tg C yr$^{-1}$, to which lateral exports to the coast contribute only 8.3%, while the remainder is contributed by $CO_2$ evasion from the inland water surface. 57% of the total dissolved carbon inputs is contributed by flooded soils and litter. Surface runoff and drainage contribute 14% and 28%, respectively. It is interesting that the flux of carbon via throughfall onto the topsoil is as high as the lateral exports of dissolved C from the topsoil, although it is not necessarily its source. According to our simulations, about 8% of the dissolved C mobilized into the water column are reinfiltrating into the soil column in swamps ($F_{up2swamp}$) or on floodplains ($F_{flood2soil}$).

[Figure]

Fig. 19: Simulated fluxes of dissolved carbon (DOC + $CO_2$) through the inland water network of the Amazon basin. Numbers are average annual fluxes ± standard deviations over the simulation period 1980-2000."

10. Fig. 4. What do the yellow color stand for? Where are the discharge gauges? Do you think landuse map is helpful?

The yellow colour represents the continental area that lies outside of the contributing areas of river sampling locations used in this study. This is now clarified in the figure caption. The river sampling location reported in this map do indeed include discharge gauges. In the revised manuscript, we added a colour code to distinguish locations that are only sampling locations of river water quality, locations that are only used as discharge gauges, and locations that serve for both (river water sampling and discharge gauge).

We added a land use map to the appendix, Fig A.1.

11. Table 4. What do the stars indicate? Why is the surface runoff represented by % for the RO3?

The surface runoff is given in % of the total runoff (surface runoff + drainage). We took those percentages directly from Johnson et al. (2006,2008), which reported surface runoff in that way, without absolute values. For comparison, we thus reported the simulated surface runoff the same way. A footnote, marked by *, should have been included to clarify this point It is now included in the revised manuscript.

Note that in Waterloo et al. (2006) only the absolute value for surface runoff was reported, i.e. values for drainage or total runoff were missing. Therefore, we do not know the percentage of surface runoff at the sampling locations reported in this study. For comparison, we report also the absolute value of our simulated surface runoff, but also that of total runoff allowing the reader to deduce the contribution of simulated surface runoff to the total runoff.

---

## Author Comment (AC3) · 14 Aug 2017

Overall the authors present compelling work to address a major deficiency in earth system models. The authors demonstrate successful simulation of CO2 and DOC lateral transport and CO2 evasion in the Amazon using ORCHILEAK. Previously, no ESM models existed that simulate the lateral transport of CO2 and DOC from surface water sources. The ORCHILEAK enables scientists to attribute DOC and evasion sources. Additionally, the model has the ability to quantify the CO2 evation in relation to terrestrial net primary production. The authors partition these processes into flooded and non-flooded lands with regard to soil carbon and DOC throughfall and subsequent lateral transport and evasion. Below I provide general and specific comments for the authors to consider to improve the current form of the manuscript.

The authors clearly describe the work that was instrumental to their contribution, and present sufficient references to support their model advancement. The authors provide a succinct abstract summarizing the modeling advancement, results, and potential future application. The authors explicitly show where code area available to replicate this project and provide this in the supplement with good instructions. Including a table of all forcing and evaluation datasets and sources would aid efforts of anyone attempting to build upon the authors work.

We agree and added a table of forcing files necessary to run ORCHILEAK over the Amazon basin (now Table 1) as well as a table of evaluation datasets used in this study (now Table 2) in the main text, method section.

The equations and formulae are correctly defined,
but providing all the equations within the appendix hinders the reader's ability to distinguish which equations are novel to the manuscript. The manuscript would benefit by including a table describing each variable in the appendix, not in a different file in the supplement, or by including the equations in line throughout the document with variable description.

We agree. We now include the equations in the text and have inserted the table with the variable explanations in the appendix (Table A.1).

1. The general methods are clearly outlined, however ambiguity exists in the temporal resolution of the model implementation. Specifically, many temporal resolutions are mentioned: i.e. 6 min, 30 min, and daily. Clarification on how these different components of the model interact would enhance clarity. Adding a flow chart or including this within Figure 3 would add clarity.

We followed the advice of Reviewer 2 and included the information on different temporal resolutions into Figure 3. A colour code, explained with a legend, now indicates the temporal resolution of each flux.

We also added a more clear description of the use of the different time steps at the end of subsection 2.2.1:

"Like the cycling of water and C in vegetation and soils, the allochthonous inputs of DOC from $S_{can}$ and $S_{soil}$ into the inland water network ($F_{RO}$, $F_{DR}$, $F_{soil2flood}$, $F_{soil2river}$, see Fig 3) are computed at a temporal resolution of 30 minutes and at the spatial resolution of the grid cell. The lateral transfer between the $S_{fast}$, $S_{slow}$, $S_{river}$ and $S_{flood}$ and the transformation of C within those storage reservoirs are only simulated at a daily time step and at the spatial resolution of the basin. Therefore, to simulate the lateral transfers, the allochthonous DOC and $CO_2$ inputs are first aggregated over 48 30-minute time steps until one full day is over. The fluxes from the water column back into the soil column ($F_{flood2soil}$, $F_{up2swamp}$ in Fig. 3) are simulated at the daily time-step of the routing module, but are used as inputs in the soil carbon module, which runs at a 30 minute temporal resolution. This is achieved by downscaling the daily fluxes uniformly over the 48 30 minute time-steps of the following day of simulation. The evasion of $CO_2$ from river and floodplain water surface ($F_{river2atm}$, $F_{flood2atm}$) is also simulated at the daily time-step of the routing module, but to approximate the continuous interplay of $CO_2$ inputs and $CO_2$ evasion controlling the water-air gradient in $CO_2$ partial pressures ($pCO_2$) a much shorter time-step of 6 minutes is used, and the $CO_2$ inputs to the water column are thus uniformly distributed over the 240 6-minutes time-step contained in each day."

2. There is limited discussion on calibration. After reading the manuscript, the majority of the parameters seem to be taken from the literature. Please describe the calibration process referenced in Sections 2 and 3.

The majority of parameters are indeed taken from the literature. The calibration process mainly focussed on the change in the flooding scheme, and the necessary recalibration of discharge afterwards. Most parameters controlling the flux of DOC from vegetation and soils to the water column and the decomposition of DOC within the water column were taken from the literature or are based on assumptions. Nevertheless, we secured that our choice of model parameters led to a good agreement with observed values. In the revised manuscript, we now report sensitivity analyses on the most important parameters, showing the sensitivity of the model results to a change in parameter values. Generally, we now include performance measures in all graphs on simulated discharges, inundation, DOC concentrations and fluxes as well as $CO_2$ evasion rates, which are more objective.

For this purpose, we restructured parts of the results and discussion section (see our response to comment #3 of reviewer #1) to better present the results of our calibration and sensitivity analyses. In section 2.3, we also now describe more clearly the general calibration procedure.

3. The authors mention this briefly on page 37 L16-21 and on 38 L2-4 how the lack of representation of POC transport might shift in stream DOC and CO2 production downstream. Please expand upon how the lack of representation impacts the current model evaluation including the impact on aggregate downstream DOC, POC, and CO2 evasion. How would this impact the evaluation results presented in Figure 12 and 13? Does this mean the current form of the the model over/under-compensates for the lack of mobile POC? Which parameters would be impacted?

We agree that this is an important point. We expanded the discussion on the absence of representation of POC transport in section 3.4, which now reads:
"...
One of the major future steps would be the implementation of particulate organic C (POC) fluxes in ORCHILEAK. Of the TOC fluxes at Obidos, the most downstream sampling location on the Amazon main stem, POC contributes less than one fourth of the total flux (Moreira-Turcq et al., 2003; Ward et al., 2015), and was reported to further decrease to only about 10% downstream to the river mouth (Ward et al., 2015). The decomposition of this POC, which is mainly derived from floodplain litter, has been

reported to contribute substantially to the in-river $CO_2$ production in the lower part of the Amazon (Ward et al., 2013). Our simulation results also highlighted the substantial contribution of submerged leaf litter to the $CO_2$ evasion. However, in our simulation, POC is not transported downstream with the water flow, i.e. it is assumed to decompose locally, and only the DOC and dissolved $CO_2$ produced from this decomposition are transferred laterally. The representation of POC transport would induce a downstream shift in the simulated DOC and $CO_2$ production from POC. The lack of this representation might have induced a bias in the simulated longitudinal pattern of DOC concentrations, $pCO_2$ and $CO_2$ evasion with an overestimation of upstream values compared to downstream values. With the limited availability of evaluation data and the rather simplified representation of POC and DOC decomposition in the model, it is impossible to conclude whether the lack of representation of POC transport explains part of the discrepancy between observed and simulated DOC concentrations (Fig. 14), or whether a too low DOC decomposition rate compensates for the bias. Mayorga et al. (2005) found that there must be a small, rapidly cycling pool of young organic matter from terrestrial vegetation close to the river that sustains a high $CO_2$ concentrations of a young [14]C age, while the majority of the transported POC is substantially older. The actual effect of POC transport shifting $CO_2$ evasion downstream is thus likely rather limited. Nevertheless, a more complete representation of fluvial POC and DOC exports would be highly beneficial to constrain dynamic boundary conditions for an ocean biogeochemical model of the Amazon plume. The application of ORCHILEAK to rivers with substantial soil erosion driven POC exports will require the implementation of soil erosion and sediment transport modules (Naipal et al., 2015, 2016).

…."

4. Figure 14 displays the performance of modeled DOC concentration relative to observations. Removing the data from Rio Negro from this plot reveals that the model only produces ranges of DOC from 3-5 mg C/L while observations are double that range [1-7 mg C/L]. How do you reconcile the low variability of the model relative to the observations? This seems in contrast to the results of the simulation data presented in Table 4.

We agree that the variability in simulated DOC concentrations is significantly lower than the observed variation in DOC concentrations. Only in case of the Rio Negro, we see flushing effects with high DOC concentrations at high discharges.

We see indeed high variation in DOC concentration in the overland flow (or "surface runoff", as it is termed in our study) (Table 6). Two distinct phenomena have to be taken into account when evaluating their effect.

First, the streams and rivers are fed by the sum of both, surface runoff and drainage, and drainage shows a comparatively low variability in DOC concentrations. For instance for sampling point RO3 (simulation b, Table 4 [now Table 6]), we simulate DOC concentration ranges (5th and 95th percentile) of 1.8 to 37.7 mg C L-1 for the surface runoff and of 2.2 to 4.8 mg C L-1 for drainage. However, according to our simulations, surface runoff contributes only 4.9% to total runoff (see Table 4), and thus DOC concentrations in total runoff (5th and 95th percentile) range only from 1.9 to 4.9 mg C L-1. In addition, we find that in our simulations the variability in DOC concentrations is highest at low total runoff, and decrease with increasing total runoff, see the following graph (Fig. R.1). When at this location, the total runoff rate is higher than on average (2.62 mm/day) (these events make in sum about 90% of the total runoff over the simulation period), the DOC concentration ranges (5th and 95th percentile) only from 2.9 to 4.3.

[Figure]

Figure R.1: Simulated DOC concentration vs. total runoff (from monthly values) for sampling location RO3, simulation b (see Table 4, which is now Table 6).

The second aspect that we have to take into account is the proportion of labile vs. refractory DOC. Note that in our simulation, labile DOC has a half-life of only 2 days. This implies that the majority of labile DOC is already decomposed before it reaches a downstream river sampling location. From table 3 (now table 5), we see that the labile proportion of DOC in the surface runoff is quite important, but also highly variable. In the drainage, the labile proportion is on the contrary generally not significant in our simulations (Table 3, now table 5). Thus, for RO3, where total runoff is dominated by drainage, the variation in labile proportions is not that important. For sampling location RO1 (see Table 4, now table 6), however, surface runoff contributes to about one third of total runoff according to our simulation. Simulated DOC concentrations vary (5th and 95th percentile) over 9.8 to 29.6 mg C L$^{-1}$ in the surface runoff and from 5.4 to 13.3 mg C L$^{-1}$ in the total runoff. The simulated concentrations of refractory DOC in total runoff vary only from 4.8 to 9.0 mg C L$^{-1}$. In our simulations, there is a tendency for a higher proportion of labile DOC at high total DOC concentrations and at high total runoff (see Fig. R.2 a and b). This pattern is in agreement with what has been observed in the field (McLaughlin and Kaplan, 2013). Thus, although we see a strong positive trend for higher total DOC concentrations at higher total runoff (Fig. R.2c), this trend is much weaker for the refractory DOC (Fig. R.2d). We also see that refractory DOC concentrations vary more strongly at low total runoff. When at this location, the total runoff is higher than on average (the average is 4.05 mm/day, the days with higher runoff contribute about 79% of the total runoff over the simulation period) the refractory DOC concentration ranges (5th and 95th percentile) only from 5.5 to 8.5 mg C L$^{-1}$.

[Figure]

Figure R.2: DOC dynamics at location RO1 (see Table 6 (which was originally 4))

In summary, the simulated DOC concentrations in total runoff vary less than in surface runoff. The variations are much lower for refractory DOC which is transported over longer distances and which represents the bulk of DOC passing to downstream river sampling locations. The variations are in addition even lower during episodes of high total runoff that contribute most to the bulk of river discharge. For these three reasons, the simulated DOC variations in the river are much lower than in surface runoff and in the headwaters.

The higher variability in observed vs. modelled DOC concentrations in the river (Figure 14) is likely due to a number of processes not included in ORCHILEAK, which control short-term fluctuations in DOC production and consumption. Note, for instance, that algae blooms can contribute to short-term peaks in DOC concentrations (Moreira-Turcq et al., 2003), and algae derived organic matter can also exert a priming effect which increases the decomposition rates of allochthonous organic carbon in the river (Ward et al., 2016).

5. The authors note how CO2 evasion is comparable to data from Richey et al., 2002, but also disclose that the inundated fraction is greatly underestimated for the central Amazon. The central Amazon shows that the model has the highest evasion rates there [Fig. 15]. The concluding remarks on Page 35, Lines 10-17 should explicitly address how the match in CO2 evasion and mismatch in inundation are related. Specifically, how does the underestimation of inundated extent impact the assertion that 51% of CO2 evasion is attributed to the floodplain?

We agree and have modified the revised manuscript to include this important point. We have added a short paragraph in the new section 3.4 of the revised manuscript:

"The fact that we simulate a total $CO_2$ evasion similar to the one reported by Richey et al. (2002) is somewhat surprising taken that that our mean water surface area is substantially lower (see section 3.1). In other words, we simulate a higher $CO_2$ evasion rate per water surface area than estimated by Richey et al. (2002). These authors used relatively lower gas exchange velocities $k_{600}$ of 1.2 to 2.3 m day$^{-1}$ to calculate $CO_2$ evasion from rivers, while we applied a significantly higher value of 3.5 m day$^{-1}$, following more recent observations (Alin et al., 2011; Rasera et al., 2013). Note that in our physically based model approach, the total $CO_2$ evasion is not so sensitive to the gas exchange velocity, but rather to the simulated $CO_2$ sources. Reducing or increasing the gas exchange velocities $k_{river,600}$ and $k_{swamp,600}$ by 50% lead to a change in simulated total $CO_2$ evasion of only -4% and 1%, respectively. On the contrary, in a data driven approach to calculate $CO_2$ from observed river $pCO_2$ values, the calculated $CO_2$ evasion will change linearly with changes in the gas exchange velocity. Rasera et al. (2013) finds higher gas exchange rates than Richey et al. (2002) and thus suggests that the total $CO_2$ evasion must be considerably higher. As the results summarized in Fig. 16 suggest, our $CO_2$ evasion rates per water surface area are comparable to those of Rasera et al. (2013). Assuming that we underestimate the average flooded area, we conclude that we likely underestimated the $CO_2$ inputs from flooded soils and vegetation and the $CO_2$ evasion from the water surface to the atmosphere. In the future, improved floodplain forgings and simulations at higher spatial resolution might help to overcome these underestimations."

Specific comments: Figure 3 caption refers to Table 1, I believe the authors intend to reference Table S1.

Reviewer #2 is right. It should be table S1, which has now become table A1 in the appendix. This has been corrected in the figure caption.